# Purkinje cell ablation and Purkinje cell-specific deletion of *Tsc1* in the developing cerebellum strengthen cerebellothalamic synapses

Hiroshi Nishiyama, Naoko Nishiyama and Boris V. Zemelman

*Center for Learning and Memory, Department of Neuroscience, The University of Texas at Austin, Austin, TX, USA*

Handling Editors: Katalin Toth & Nathan Schoppa

The peer review history is available in the Supporting Information section of this article (https://doi.org/10.1113/JP285887#support-information-section).

**Abstract figure legend** Cerebellothalamic fibres are excitatory projections originating from the cerebellar nuclei, transmitting cerebellar signals to the forebrain. Using whole-cell patch-clamp recordings and optogenetic stimulation, we demonstrated that Purkinje cell ablation and Purkinje cell-specific deletion of *Tsc1* (an autism-associated gene) in the developing cerebellar cortex strengthen cerebellothalamic synapses.

**Hiroshi Nishiyama** is an Associate Professor in the Center for Learning and Memory, Department of Neuroscience at the University of Texas at Austin. He completed his PhD at Kyoto University and postdoctoral training at the Johns Hopkins University. His lab primarily uses two-photon microscopy and electrophysiology to study how neuronal circuits in the cerebellum are refined during development, reorganized in adulthood, and affect the formation and function of other brain regions.

This article was first published as a preprint. Nishiyama H, Nishiyama N, Zemelman BV. 2023. Loss of Purkinje cells in the developing cerebellum strengthens the cerebellothalamic synapses. bioRxiv. https://doi.org/10.1101/2023.11.01.564864

The Journal of Physiology

**Abstract** Cerebellar damage early in life often causes long-lasting motor, social and cognitive impairments, suggesting the roles of the cerebellum in developing a broad spectrum of behaviours. This recent finding has promoted research on how cerebellar damage affects the development of the cerebral cortex, the brain region responsible for higher-order control of all behaviours. However, the cerebral cortex is not directly connected to the cerebellum. The thalamus is a major direct target of the cerebellar nuclei, conveying cerebellar signals to the cerebral cortex. Despite its crucial position in cerebello–cerebral interaction, thalamic susceptibility to cerebellar damage remains largely unclear. Here, we studied the consequences of early cerebellar perturbation on thalamic development. Whole-cell patch-clamp recordings showed that the synaptic organization of the cerebellothalamic circuit is similar to that of the primary sensory thalamus, in which aberrant sensory activity alters synaptic circuit formation. The ablation of Purkinje cells in the developing cerebellum strengthened cerebellothalamic synapses and enhanced thalamic suprathreshold activities. Purkinje-cell specific deletion of tuberous sclerosis complex subunit 1 (*Tsc1*), an autism-associated gene for which the protein product negatively regulates the mammalian target of rapamycin, also strengthened cerebellothalamic synapses. However, this strengthening occurred only in homozygous deletion, whereas both homozygous and hemizygous deletion are known to cause autism-like behaviours. These results suggest that, although the cerebellothalamic projection is vulnerable to disturbances in the developing cerebellar cortex, other changes may also drive the behavioural consequences of early cerebellar perturbation.

(Received 2 November 2023; accepted after revision 22 October 2024; first published online 18 November 2024)

**Corresponding author** H. Nishiyama: Center for Learning and Memory, Department of Neuroscience, The University of Texas at Austin, 1 University Station, Stop C7000, Austin, TX 78712, USA.    Email: hnishiyama@mail.clm.utexas.edu

## Key points

- Cerebellar damage early in life often causes motor, social and cognitive impairments, suggesting the roles of the cerebellum in developing a broad spectrum of behaviours.
- Recent studies focus on how the developing cerebellum affects the formation and function of the cerebral cortex, the higher-order centre for all behaviours. However, the cerebellum does not directly connect to the cerebral cortex.
- Here, we studied the consequences of early cerebellar perturbation on the thalamus because it is a direct postsynaptic target of the cerebellum, sending cerebellar signals to the cerebral cortex.
- Loss of cerebellar Purkinje cells, which are commonly associated with various neurological disorders, strengthened cerebellothalamic synapses, suggesting the vulnerability of the thalamus to substantial disturbance in the developing cerebellum.
- Purkinje cell-specific loss of tuberous sclerosis complex-1, a negative regulator of mammalian target of rapamycin, is an established mouse model of autism. This mouse model also showed strengthened cerebellothalamic synapses.

## Introduction

Cerebellar abnormality occurs in a variety of neurological disorders, such as autism spectrum disorder (ASD), Down syndrome, schizophrenia and spinocerebellar ataxia (Bailey et al., 1998; D'Mello & Stoodley, 2015; Moberget et al., 2018; Pinter et al., 2001; Tamura et al., 2017; Wang et al., 2014). The link between the cerebellum and cognitive impairments in these disorders previously seemed perplexing because the cerebellum was traditionally considered a part of the motor system.

However, recent studies show that the cerebellum is also involved in various non-motor functions, such as cognitive processing, social behaviour, fear and emotion, through its interaction with other brain regions (Badura et al., 2018; Chao et al., 2023; Fastenrath et al., 2022; Koziol et al., 2014; Palesi et al., 2015; Vaaga et al., 2020).

Still, the emerging roles of the cerebellum in non-motor behaviours do not fully explain how the cerebellum contributes to the pathogenesis of the above neurological disorders. Although cerebellar injury at birth has one of the highest known risk ratios for ASD, damage

in the mature cerebellum is not (Wang et al., 2014). Chemogenetic perturbation of cerebellar circuits causes severe social and cognitive impairments when it occurs in the juvenile period but not in adulthood (Badura et al., 2018). These findings show that early cerebellar damage or dysfunction causes more profound neurological symptoms than later ones, suggesting that the cerebellum is necessary to develop functionally precise brain circuits.

Neuronal activity plays a crucial role in the maturation and refinement of synaptic circuits (Katz & Shatz, 1996; Lichtman & Colman, 2000). Therefore, aberrant activities of the immature cerebellum as a result of injury or genetic mutation potentially interfere with the development of its downstream targets (Wang et al., 2014). In this regard, recent studies focused on the cerebral cortex, the higher-order centre for all behaviours, showing that early cerebellar damage in human patients results in reduced cortical volume (Limperopoulos et al., 2010, 2014; Stoodley & Limperopoulos, 2016). However, the cerebellum is not directly connected to the cerebral cortex. How aberrant cerebellar activities impair cortical development is largely unclear.

In the present study, we focused on the thalamus for the following reasons. First, the cerebellum sends dense, mono-synaptic excitatory projections to the thalamus, directly impacting the synaptic activities in the thalamus (Bosch-Bouju et al., 2013; Chan-Palay, 1977; Evarts & Thach, 1969). Second, the thalamus is the heart of cerebello–cerebral interaction. It relays cerebellar activity to the motor and non-motor areas of the cerebral cortex (D'Angelo & Casali, 2013; Koziol et al., 2014); hence, if thalamic circuits are not formed properly, functional interaction between the cerebellum and cerebral cortex must be disturbed. Third, neuronal activities are necessary for the maturation of the primary sensory thalamus (Hooks & Chen, 2006; Takeuchi et al., 2014; Wang & Zhang, 2008), suggesting that normal cerebellar activities are also required to build functional cerebellothalamic circuits.

To study how cerebellar perturbation affects thalamic development, we compared neuronal function in two mouse lines. We created a transgenic mouse line in which cerebellar Purkinje cells are ablated from the second postnatal week. This mouse line represents a considerable perturbation because the ablation of Purkinje cells removes their potent tonic inhibition on the deep cerebellar nuclei, substantially altering the cerebellar output activities to its downstream targets. We also used a previously reported mouse model of autism caused by Purkinje cell-specific deletion of the *Tsc1* gene from the second postnatal week to study the effect of a moderate perturbation (Tsai et al., 2012).

We performed whole-cell patch-clamp recordings in thalamic slices obtained from these mice and characterized the synaptic and intrinsic properties of the cerebellothalamic circuit. Purkinje cell ablation and the homozygous deletion of the *Tsc1* gene both strengthened the synaptic inputs from the cerebellum to the thalamus. However, only Purkinje cell ablation significantly enhanced synaptically-evoked thalamic action potentials, presumably because it slightly hyperpolarized the thalamic resting membrane potentials. These results suggest that damage in the developing cerebellum potentially alters the cerebellothalamic circuit, depending on the type and severity of the damage.

## Methods

### Ethical approval

All animal procedures were performed in accordance with the University of Texas at Austin Institutional Animal Care and Use Committee guidelines (AUP-2020-00208, AUP-2023-00169). The animals had access to food and water *ad libitum*.

### Animals

The knock-in mouse line, in which the Cre-inducible diphtheria toxin fragment A (DTA) gene is inserted downstream of the neuron-specific enolase promoter ($Eno2^{fsDTA/+}$; RBRC10744; Riken BioResource Centre, Japan), the transgenic mouse line, in which Purkinje cells-specific protein-2 promoter drives Cre expression ($Pcp2^{Cre/+}$; JAX:004146; The Jackson Laboratory, Bar Harbor, ME, USA) and the floxed mutant mouse line, in which exons 17 and 18 of the *Tsc1* gene are flanked by loxP sites ($Tsc1^{flox/+}$; JAX:005680, The Jackson Laboratory) were used in this study. These mouse lines were maintained in the C57BL/6J background.

To ablate Purkinje cells, we crossed $Pcp2^{Cre/+}$ mice with $Eno2^{fsDTA/+}$ mice and obtained the double transgenic mice ($Pcp2^{Cre/+}$ $Eno2^{fsDTA/+}$). The single transgenic littermates (either $Pcp2^{Cre/+}$ or $Eno2^{fsDTA/+}$) were used as a control group. To delete the *Tsc1* gene in Purkinje cells hemizygously, we crossed $Pcp2^{Cre/+}$ mice with $Tsc1^{flox/+}$ mice and obtained the double transgenic mice ($Pcp2^{Cre/+}$ $Tsc1^{flox/+}$). The single transgenic littermates (either $Pcp2^{Cre/+}$ or $Tsc1^{flox/+}$) were used as a control group. For homozygous deletion, we crossed $Pcp2^{Cre/+}$ $Tsc1^{flox/+}$ double transgenic mice with $Tsc1^{flox/flox}$ mice and obtained the double transgenic mice with homozygous floxed *Tsc1* allele ($Pcp2^{Cre/+}$ $Tsc1^{flox/flox}$). The single transgenic littermates ($Tsc1^{flox/flox}$) were used as a control group. The control groups for the hemizygous and homozygous deletion were combined in the analysis.

## Virus preparation

An adeno-associated virus (AAV) encoding an enhanced human synapsin promoter, channelrhodopsin2-tdTomato, woodchuck hepatitis virus posttranscriptional regulatory element (i.e. WPRE) and SV40 polyadenylation signal was assembled using a modified helper-free system (Stratagene, San Diego, CA, USA) as a serotype 2/1 (rep/cap genes) AAV and then harvested and purified over sequential cesium chloride gradients as previously described (Grieger et al., 2006). The hybrid synapsin promoter comprised a cytomegalovirus enhancer (Niwa et al., 1991), a neural-restrictive silencer element (Mori et al., 1992) and a human synapsin promoter (Kügler et al., 2001). The channerhodopsin2 fusion protein included C-terminal Golgi and endoplasmic reticulum export signals (Hofherr et al., 2005; Ma et al., 2001) to aid membrane expression. Virus titers were typically $>10^{13}$ genomes per ml.

## Animal injection

The injection was performed on neonatal male and female mice around the postnatal day 7 (P7) before substantial cerebellar atrophy occurred in the Purkinje cell ablation mice. AAV was injected into the deep cerebellar nuclei of the control mice, Purkinje cell-specific *Tsc1* deletion mice and Purkinje cell ablation mice.

Neonatal mice were anaesthetized with 5% isoflurane, and the dose was reduced to 1.5–2% following the induction. Carprofen (10 mg kg$^{-1}$) was s.c. administrated as an analgesic. The scalp was cut and the left side of the interparietal bone was exposed. Lidocaine (2%) was topically applied to the exposed skull and adjacent muscle surfaces. Posterior to the lambdoid suture, another suture runs mediolaterally between the interparietal and occipital bones. We used the midpoint of this suture as a landmark and located the injection site 1.5 mm left and 1.2 mm anterior from the landmark. A small craniotomy was made on the injection site with a dental drill and a drill bit (tip diameter of 0.8 mm), and a glass pipette (tip diameter of 20–40 μm) containing the virus solution was inserted vertically to the depth of 2.2 mm from the surface. Approximately 200–300 nL of virus was injected over 4–6 min using a Nanoject II injector (Drummond Scientific, Broomall, PA, USA). This injection co-ordinates primarily label the interposed nucleus of the deep cerebellar nuclei, but the virus also spreads into the lateral and medial nucleus. The pipette was left in place for 5 min before being withdrawn, and the scalp was sutured. The mice were returned to their mother after recovering from anaesthesia. Carprofen (10 mg kg$^{-1}$) was administrated every 24 h for 2 days post-surgery as an analgesic. The mice were kept alive for 1.5–3 months until being used for electrophysiological recordings.

## Histochemistry

The control and Purkinje cell ablation mice (1 month old) were anaesthetized with an intraperitoneal injection of ketamine/xylazine (100/10 mg kg$^{-1}$) and intra-cardially perfused with 4% paraformaldehyde in phosphate-buffered saline (PBS). The brains were extracted post-perfusion and immersed in the same fixative at 4°C overnight. After washing the tissues with PBS, sagittal sections of the cerebellum (50 μm thickness) and coronal sections of the forebrain (80 μm thickness) were cut using Microm HM650V vibration microtome (Thermo Fisher Scientific, Waltham, MA, USA). The sections were washed with PBS containing 0.3% Triton X-100 (PBST).

Then, the cerebellar sections were blocked at room temperature for at least 1 h in the blocking solution containing PBST and 5% normal goat serum. Following the blocking step, the sections were incubated with anti-calbindin-D28k mouse monoclonal antibody (1:2000 dilution by the blocking solution; C9848; MilliporeSigma, Burlington, MA, USA) at 4°C overnight, then washed with PBST and incubated with goat anti-mouse IgG (H+L) Alexa Fluor 594 (1:500 dilution by the blocking solution; A-11032; Thermo Fisher Scientific) at room temperature for at least 2 h. Sections were then washed with PBST and mounted with 4′,6-diamidino-2-phenylindole (DAPI)-containing mounting media (DAPI Fluoromount-G, 0100-20; Southern Biotech, Birmingham, AL, USA).

The forebrain sections were incubated with Blue Neuro-Trace Fluorescent Nissl Stains (1:100 dilution by PBS; N-21479; Thermo Fisher Scientific) at room temperature for 2–3 h. Sections were then washed with PBST and mounted with mounting media (Fluoromount-G, 0100-01; Southern Biotech).

## Image acquisition

The wide-field and confocal fluorescence images were acquired with a FV1000 confocal microscope system (Olympus, Tokyo, Japan). For the wide-field images, the stained sections were excited by a mercury arc lamp and fluorescence emission was collected via a 4× objective lens (0.1 NA). DAPI and Blue NeuroTrace Fluorescent Nissl (Thermo Fisher Scientific) were excited using a bandpass filter [325–375 nm (violet)] and fluorescence emission was filtered with a bandpass filter [435–485 nm (blue)] before being captured by a CCD camera. A long-pass filter 400 nm was used as a dichroic mirror to separate the violet excitation and blue emission. Alexa Fluor 594 was excited using a bandpass filter [510–560 nm (green)] and fluorescence emission was filtered with a bandpass filter [575–645 nm (red)]. A long-pass filter 565 nm was used

as a dichroic mirror to separate the green excitation and red emission.

For the confocal images, the immunostained cerebellar sections were excited by 405 nm (for DAPI) and 543 nm (for Alexa Fluor 594) lasers. The DAPI and Alexa Fluor 594 fluorescence emissions were separated by a long-pass filter 490 nm and then filtered by bandpass filters 430–470 nm and 560–660 nm, respectively. Z-stack images were acquired with a $40\times$ objective lens (0.75 NA) at 2 μm step size with the $x$–$y$ resolution of 0.8 μm per pixel.

### Purkinje cell number quantification

The control and Purkinje cell ablation mice (post-natal days around 10, 15, 20 and 30) were deeply anaesthetized with isoflurane and decapitated. The brains were extracted, briefly washed in PBS and fixed in 4% paraformaldehyde in PBS at 4°C overnight. The 20 μm thick parasagittal slices of the cerebellum were cut and stained with anti-calbindin-D28k mouse mono-clonal antibody and Blue NeuroTrace Fluorescent Nissl Stains as described in the Histochemistry section. The wide-field images were obtained as described in the Image acquisition section.

For the control mice (all ages) and 10-day-old Purkinje cell ablation mice, the length of the molecular layer and the linear density of Purkinje cells were manually measured for each slice using ImageJ (NIH, Bethesda, MD, USA). The total number of Purkinje cells in the slice was estimated as the product of the molecular layer length and the linear density. For elder Purkinje cell ablation mice in which most Purkinje cells were already degenerated, the remaining Purkinje cells in each slice were manually counted. The total number of Purkinje cells in the control mice was averaged in each age group. Then, the number of Purkinje cells in each mouse in each age group was normalized to the average value of the control mice in the age group.

### Electrophysiology: overall procedures

Acute brain slices were cut near physiological temperatures, as recommended in a previous study (Huang & Uusisaari, 2013). The cutting and holding solutions were prepared by referring to the methods optimized for adult brains (Ting et al., 2014, 2018), although we used sucrose to replace NaCl and simplified the procedure.

To record from the ventrolateral thalamus (VL), mice (1.5–3 months old, 5–11 weeks post-injection) were deeply anaesthetized with isoflurane. The brain was quickly removed after decapitation and submerged in a warmed cutting solution (32-34°C) containing (in mM): 150 sucrose, 30 $NaHCO_3$, 20 HEPES, 2.5 KCl, 1.2 $NaH_2PO_4$, 0.5 $CaCl_2$, 10 $MgCl_2$, 5 Na-ascorbate, 3 Na-pyruvate and 25 dextrose. Coronal forebrain slices (220 μm thick) containing the VL were cut in the solution using a 7000smz-2 vibration microtome (Campden Instruments, Loughborough, UK), recovered in a warmed holding solution (32–34°C) containing (in mM): 90 NaCl, 20 sucrose, 30 $NaHCO_3$, 20 HEPES, 2.5 KCl, 1.2 $NaH_2PO_4$, 2 $CaCl_2$, 2 $MgCl_2$, 5 Na-ascorbate, 3 Na-pyruvate and 25 dextrose. The slices were kept at the warmed temperature for 30 min and then stored at room temperature until being used for recording.

To record from the deep cerebellar nuclei (DCN), parasagittal cerebellar slices (220 μm thick) were prepared as described above from 2-week-old and 1.5-3-month-old mice that did not receive virus injection.

Recordings were performed at 32°C in a submerged chamber perfused with artificial cerebrospinal fluid containing (in mM): 124 NaCl, 2.5 KCl, 1.25, $NaH_2PO_4$, 25 $NaHCO_3$, 2 $CaCl_2$, 2 $MgCl_2$ and 20 dextrose, supplemented with gabazine (5 μM; HB0901; Hello Bio, Bristol, UK) to block $GABA_A$ receptor-mediated inhibitory synaptic transmission. To block $\alpha$-amino-3-hydroxy-5-methyl-4-isoxazolepropionic acid receptors (AMPA-R) and *N*-methyl D-aspartate receptors (NMDA-R)-mediated excitatory synaptic transmission, the AMPA-R antagonist 2,3-dioxo-6-nitro-1,2,3,4-tetrahydrobenzo[*f*]quinoxaline-7-sulfonamide (NBQX) (10 μM; HB0443; Hello Bio) and NMDA-R antagonist DL-2-amino-5-phosphonopentanoic acid (DL-APV) (100 μM; HB0252; Hello Bio) were added to artificial cerebrospinal fluid in some experiments. All the solutions were bubbled with 95% $O_2$ and 5% $CO_2$, and the osmolality was adjusted to $\sim310 \pm 10$ mOsm kg$^{-1}$.

Patch-clamp recordings were made using a Multiclamp 700B amplifier (Molecular Devices, San Jose, CA, USA) and AxoGraph data acquisition software (https://axograph.com). The signals were sampled at 20 kHz with an ITC-18 AD/DA board (Heka, Reutlingen, Germany.). The pipette solution for cell-attached recordings was 150 mM NaCl. The intracellular solution for whole-cell voltage-clamp recordings contained (in mM): 35 CsF, 100 CsCl, 10 EGTA, 10 HEPES and 4 QX314 ($\sim290$ mOsm kg$^{-1}$, pH adjusted to $\sim7.3$ with CsOH). The intracellular solution for whole-cell current-clamp recordings contained (in mM): 130 K-MeSO$_3$, 4 KCl, 10 HEPES, 8 Na$_2$-phosphocreatine, 0.2 EGTA, 4 Mg-ATP and 0.4 Na-GTP ($\sim290$ mOsm kg$^{-1}$, pH adjusted to $\sim7.3$ with KOH). The pipette resistance was around $3 \pm 0.5$ MΩ when filled with the solutions. The AxoGraph recording data files were read with Python (https://www.python.org) using the Axographio package, and all the measurements, event detection, threshold detection and curve fitting were performed with Python.

## DCN recordings

Large DCN neurons with a soma diameter of ~20–30 μm were identified with a Fluor 60X water immersion objective lens (1.0 NA) (Nikon, Tokyo, Japan) under differential interference contrast mode. Most of these neurons spontaneously discharged action potentials (APs), which could be recorded even before a giga-ohm seal was achieved. These spontaneous APs were recorded under cell-attached mode for 10 s, and the firing rate was calculated.

After the recording, the pipette was replaced with a new one containing the intracellular solution for whole-cell current-clamp recordings. When possible, the whole-cell recordings were made from the same DCN neurons recorded under cell-attached mode. Neurons were initially voltage-clamped at –70 mV, and the input resistance was calculated from the amplitude of the steady-state current in response to a –5 mV, 50 ms hyperpolarizing voltage step. Then, current-clamp recordings were performed with bridge balance, compensating the series resistance. The depolarizing current step (800 ms, 0–400 pA with 50 pA increment) was injected every 20 s, and the AP frequency during the current injection was measured for each current step.

## VL recordings and optogenetics

Cerebellothalamic axon terminals expressing channelrhodopsin2 (ChR2) were identified in the VL using tdTomato fluorescence excited by 560 nm wavelength LED light (BLS-LCS-0560-03-22; Mightex, Toronto, ON, Canada). Individual postsynaptic VL neurons were visualized with the same 60X objective lens used for DCN recordings. After whole-cell recording was established, ChR2 was excited by a brief (0.1–0.2 ms) pulse of 470 nm wavelength LED light (BLS-LCS-0470-03-22; Mightex). The duration, power and timing of the pulse were controlled by BLS-series software (BLS-SA02-US; Mightex). The power under the objective lens was measured using an optical power meter (PM10; Thorlabs, Newton, NJ, USA) and a sensor (D10MM; Thorlabs). The synaptic and intrinsic properties of VL neurons were quantified as described below.

## Voltage-clamp recordings in the VL

VL neurons were voltage-clamped at –70 mV and ChR2-expressing cerebellothalamic axons were stimulated every 20 s by a 470 nm light pulse. The light power density was gradually increased from 0 to ~40 mW mm$^{-2}$. This stimulation protocol yielded one or several discrete steps of excitatory postsynaptic currents (EPSCs). The peak amplitude and charge transfer of each EPSC step were quantified and plotted against the light power density. The charge transfer *vs.* light power density plots and the corresponding EPSC traces were used to count the number of discrete EPSC steps.

After the above recording session, the same VL neurons remained held at –70 mV and stimulated by a pair of light pulses with the maximum power. The paired light pulses were given every 20 s by changing the paired-pulse interval from 25 to 200 ms. This recording was repeated two to four times, and the EPSC traces for each paired-pulse interval were averaged. Finally, the holding potential was changed from –70 mV to +40 mV, and the same neurons were stimulated by the maximum power of light. The light pulse was given every 20 s, three to six times, and the EPSC traces were averaged.

The amplitude, 10–90% rise time and the weighted decay time constant of the first EPSC in the paired-pulse recording were measured. The weighted decay time constant was calculated as described in a previous study (Cathala et al., 2000). Briefly, a double exponential decay curve was fitted to the decay phase of the EPSCs. If the coefficient and time constant of a fast and slow decay component are $C_{fast}$, $C_{slow}$, $\tau_{fast}$ and $\tau_{slow}$, the weighted decay constant was calculated as: $(C_{fast}\tau_{fast} + C_{slow}\tau_{slow})/(C_{fast} + C_{slow})$. The paired-pulse ratio was calculated as the amplitude ratio between the second and the first EPSCs.

The EPSCs recorded at +40 mV include AMPA and NMDA currents. To measure the NMDA currents near the peak yet minimize the AMPA currents as much as possible, we measured the amplitude of EPSCs +40 mV at 13 ms after the optogenetic stimulation. This is ~10 ms after the peak of AMPA currents. Based on the average $C_{fast}$, $C_{slow}$, $\tau_{fast}$ and $\tau_{slow}$ recorded from all the cells, AMPA currents decay ~92% at 10 ms after the peak. On the other hand, this timing is a few milliseconds after the peak of NMDA currents, which do not decay much due to their slow kinetics. The AMPA/NMDA ratio was calculated as the amplitude ratio between the EPSC –70 mV (peak) and the EPSC +40 mV (13 ms after the stimulation).

The series resistance was measured in all recordings by the current response to a 5 mV, 50 ms hyperpolarizing voltage step. The cells showing a series resistance larger than 15 MΩ were excluded. In addition, some cells showed no or very small EPSCs, most probably because their primary synaptic inputs were damaged or not labelled by ChR2. To reduce the contribution of those cells to underestimating the synaptic strength and the number of synaptic inputs, we excluded the cells showing their maximum EPSC amplitude below 1 nA.

Furthermore, a few VL neurons expressed ChR2. These neurons were directly stimulated, showing fundamentally different currents from EPSCs, and were excluded from the analysis. The VL neurons may have been infected following virus diffusion through the superior cerebellar

peduncle to the thalamus. Although trans-synaptic labelling has previously been postulated (Castle et al., 2014; Zingg et al., 2017, 2020), we have seen no evidence of AAV transfer between cells with any of our reagents.

## Current-clamp recordings in the VL

VL neurons were initially voltage-clamped at –70 mV and stimulated with light. Cells were discarded if they were directly stimulated or showed no or small EPSCs. Other cells were switched to current-clamp mode with no holding current. The series resistance was compensated using bridge balance.

Cells were stimulated every 20 s with a single light pulse, and the number of APs evoked by the maximum power of light was counted. The average voltage before the light pulse was used to calculate the resting membrane potential. The input resistance was calculated from the average steady-state voltage deflection in response to a 50 pA, 300 ms hyperpolarizing current injection.

The synaptically evoked APs began to fire on the steep rise of excitatory postsynaptic potentials (EPSPs), making the detection of the first AP threshold unreliable. Therefore, after the above recording session, we injected depolarizing current steps (200 ms, 25–250 pA with a 25 pA increment) into the same cells to quantify AP properties. The AP threshold was defined as the voltage where dV/dt exceeded 10 mV ms$^{-1}$. The AP amplitude was defined as the difference between the peak and threshold. The AP half-width was defined as the duration during which an AP was higher than half of its amplitude. These values were obtained for the first AP in each current step and averaged across all steps that evoked APs.

In addition, the average voltage before the current injection was used to calculate the resting membrane potential. This value was averaged with the corresponding value obtained in the optogenetic stimulation and used as the resting membrane potential of the cells.

## Statistical analysis

All measured values are reported as the mean ± SD. In most analyses, we performed bootstrapping to test the statistical significance and estimate the effect size using the R package, Dabestr (Ho et al., 2019). *P* values indicate whether the difference between groups is statistically significant, whereas effect sizes indicate the magnitude of the difference (i.e. how substantially the distribution differs between the groups) (Cohen, 1990; Meissel & Yao, 2024; Sullivan & Feinn, 2012). Because an appropriate measure of effect size depends on the distribution, we first test the normality of the data distribution by the Shapiro–Wilk test.

For normally distributed data, the standardized mean difference (i.e. the mean difference between two groups divided by their pooled SD) was used as a parametric effect size. This value, Cohen's *d*, is widely used to evaluate the magnitude of the difference (Cohen, 1990; Meissel & Yao, 2024; Sullivan & Feinn, 2012). If Cohen's *d* is 1, the mean difference between the groups is equal to 1 pooled SD.

For non-normally distributed data, Cliff's delta was used as a non-parametric effect size (Cliff, 1993; Meissel & Yao, 2024). Briefly, all individual values in two distributions (e.g. control and test groups) were compared in a pairwise manner. Then, the number of times a value in the test group was larger or smaller than a value in the control group is counted. The difference between the count ($N_{\text{test > control}} - N_{\text{test < control}}$) was divided by the total number of the pairwise comparisons (i.e. the product of the number of samples in each group). The resultant value, Cliff's delta, ranges from –1 (all individual values in the test group are smaller than those in the control group) to 1 (all individual values in the test group are larger than those in the control group), which is a highly robust and powerful non-parametric effect size (Cliff, 1993; Meissel & Yao, 2024). Although the median focuses only on the centre of the distribution, Cliff's delta utilizes all individual values without being distorted by outliers.

We performed bootstrap resampling of the observed data 5000 times. It computes the observed effect size (Cohen's *d* or Cliff's delta) and its resampled distribution with an appropriate confidence interval. The 95% confidence interval was used to compare the control *vs.* Purkinje cell ablation mice. The 97.5% confidence interval was used to perform two comparisons: (1) the control *vs. Tsc1* hemizygous deletion mice and (2) the control *vs. Tsc1* homozygous deletion mice (Bonferroni correction for multiple comparisons). If the confidence interval of the resampled effect size does not include zero, it is equivalent to $P < 0.05$ in null hypothesis significance testing. Furthermore, beyond the statistical significance, the sharpness of the effect size distribution and the proximity of the confidence interval to zero allows for inferring the magnitude and certainty of the difference (Calin-Jageman & Cumming, 2019; Ho et al., 2019).

Despite these advantages, however, effect sizes or bootstrapping have not been used frequently in similar studies. Therefore, we also performed more commonly used statistical tests, such as Welch's *t* test and Wilcoxon rank sum test, and provided the results for comparison.

For the paired-pulse ratio, mixed model analysis of variance (mixed ANOVA) was used to test the difference between the control and cerebellar mutant mice, the interstimulus intervals, and the interaction between these variables. Because Mauchly's test for sphericity revealed a substantial degree of asphericity (epsilon < 0.75) with the interstimulus interval, the Greenhouse–Geisser correction was used to adjust the degrees of freedom of

the *F*-distribution. $P < 0.05$ was considered statistically significant.

For the correlation, Pearson's correlation coefficient was computed between the two variables and $P < 0.05$ was considered a statistically significant correlation. The normality of the data distribution was confirmed by the Shapiro–Wilk test. All the statistical analyses and data visualization were performed with R (https://www.r-project.org/).

## Results

Although the primary goal of this study is to gain basic biological insights into how the cerebellum contributes to thalamic formation, we also sought to understand it in the context of neurological disorders. Therefore, we perturbed the developing cerebellum in a manner that resembles human diseases. One of the most common cerebellar pathologies in neurological conditions is Purkinje cell abnormality (Cook et al., 2021; Gill & Sillitoe, 2019). Purkinje cells spontaneously discharge action potentials and tonically inhibit the DCN, the sole output of the cerebellum (Bell & Grimm, 1969; Latham & Paul, 1971; Palay & Chan-Palay, 1974). Hence, Purkinje cell dysfunction directly impacts the DCN activity, altering the communication between the cerebellum and other brain regions.

Loss of Purkinje cells or their reduced firing rate is commonly associated with various neurological disorders, such as ASD, schizophrenia, bipolar disorder, Huntington's disease and spinocerebellar ataxia (Bailey et al., 1998; Cook et al., 2021; Dougherty et al., 2012, 2013; Kordasiewicz & Gomez, 2007; Maloku et al., 2010; Zoghbi & Orr, 1995). To highlight the effects of Purkinje cell loss, we ablated Purkinje cells selectively via DTA. A knock-in mouse in which the Cre-inducible DTA gene is inserted downstream of the neuron-specific enolase promoter ($Eno2^{fsDTA/+}$ mouse) was crossed with a transgenic mouse in which Purkinje cells-specific protein-2 promoter drives Cre expression ($Pcp2^{Cre/+}$ mouse). Because DTA expression is Cre-dependent, only Cre-expressing cells are ablated by DTA.

Previous studies used the $Eno2^{fsDTA/+}$ mouse successfully for cell type-specific ablation (Imayoshi et al., 2008; Kobayakawa et al., 2007; Kobayashi et al., 2013). The $Pcp2^{Cre/+}$ mouse is a well-established genetic tool for Purkinje cell-specific gene manipulation, in which Cre-mediated recombination begins from postnatal day 6 (P6) (Barski et al., 2000; Sługocka et al., 2017; Tsai et al., 2012). Single transgenic mice carrying either the DTA or the Cre gene were indistinguishable from wild-type mice (Fig. *1A–D*). On the other hand, the double-transgenic mice expressing DTA in Purkinje cells showed cerebellar atrophy as a result of substantial loss of Purkinje cells (Fig. 1*E–H*). The atrophy was specific to the cerebellum; the overall forebrain structure was unaffected (Fig. 1*B* and *F*).

A seemingly normal Purkinje cell layer was observed at P10 in the Purkinje cell ablation mice (DTA mice) (Fig. 1*I*). The continuous linear array of Purkinje cells disappeared by P15 (Fig. 1*J*), and the cell population rapidly declined during this period, from ~80% of the control mice at P9–10 ($79.09 \pm 5.97\%$, $n = 3$ mice) to 8% at P14–15 ($7.81 \pm 2.47\%$, $n = 3$ mice) (Fig. 1*M*). The Purkinje cell number further declined to ~4% of the control mice at P20–21 ($4.04 \pm 1.71\%$, $n = 3$ mice) and 1.5% at P31–32 ($1.53 \pm 0.40\%$, $n = 3$ mice) (Fig. 1*M*). This time course is consistent with the previous study, showing that Cre-mediated recombination is fully established 2–3 weeks after birth in the $Pcp2^{Cre/+}$ mouse (Barski et al., 2000). However, more than 90% of Purkinje cells were lost by the end of the second postnatal week, indicating that the DCN neurons in the DTA mice are no longer under potent inhibition by Purkinje cells during the developmental phase of the cerebellar cortex.

Although DCN neurons discharge APs without synaptic inputs, it remains unknown how chronically removing the tonic synaptic inhibition affects their functional maturation. Therefore, we recorded the spontaneous AP discharge extracellularly from the DCN neurons in fresh slices and compared the AP firing rate between the control and DTA mice (Fig. 2 and Table 1). Patch-clamp recordings were made only from large neurons because the DCN neurons sending the cerebellothalamic axons tend to have a large soma (Raman et al., 2000; Uusisaari & Knöpfel, 2011).

At 2 weeks old, the spontaneous AP firing rate was $20.10 \pm 12.44$ Hz ($n = 36$ cells) in the control mice and $14.64 \pm 10.91$ Hz ($n = 37$ cells) in the DTA mice (Fig. 2*A* and *C*). The difference between the groups (DTA – Control in Fig. 2*C*, bottom) is depicted using the observed effect size (the black circle) and its bootstrap resampling distribution (the grey curve). The observed effect size was modest compared to its distribution range because the 95% confidence interval (the thick vertical line on the grey curve) extended to near the zero line. Although the distribution was substantially biased toward lower AP firing rates in the DTA mice, the 95% confidence interval included zero, indicating that the difference between the groups was not statistically significant (Table 1).

Elder mice showed higher mean firing rates associated with a broader frequency range (Fig. 2*B* and *D*). The mean firing rate was $37.75 \pm 28.21$ Hz ($n = 22$ cells) in the control mice and $32.52 \pm 39.51$ Hz ($n = 27$ cells) in the DTA mice. The difference was not significant because the 95% confidence interval of the estimated effect size included zero (Fig. 2*D*). Although these results do not certainly rule out the possibility that the spontaneous AP

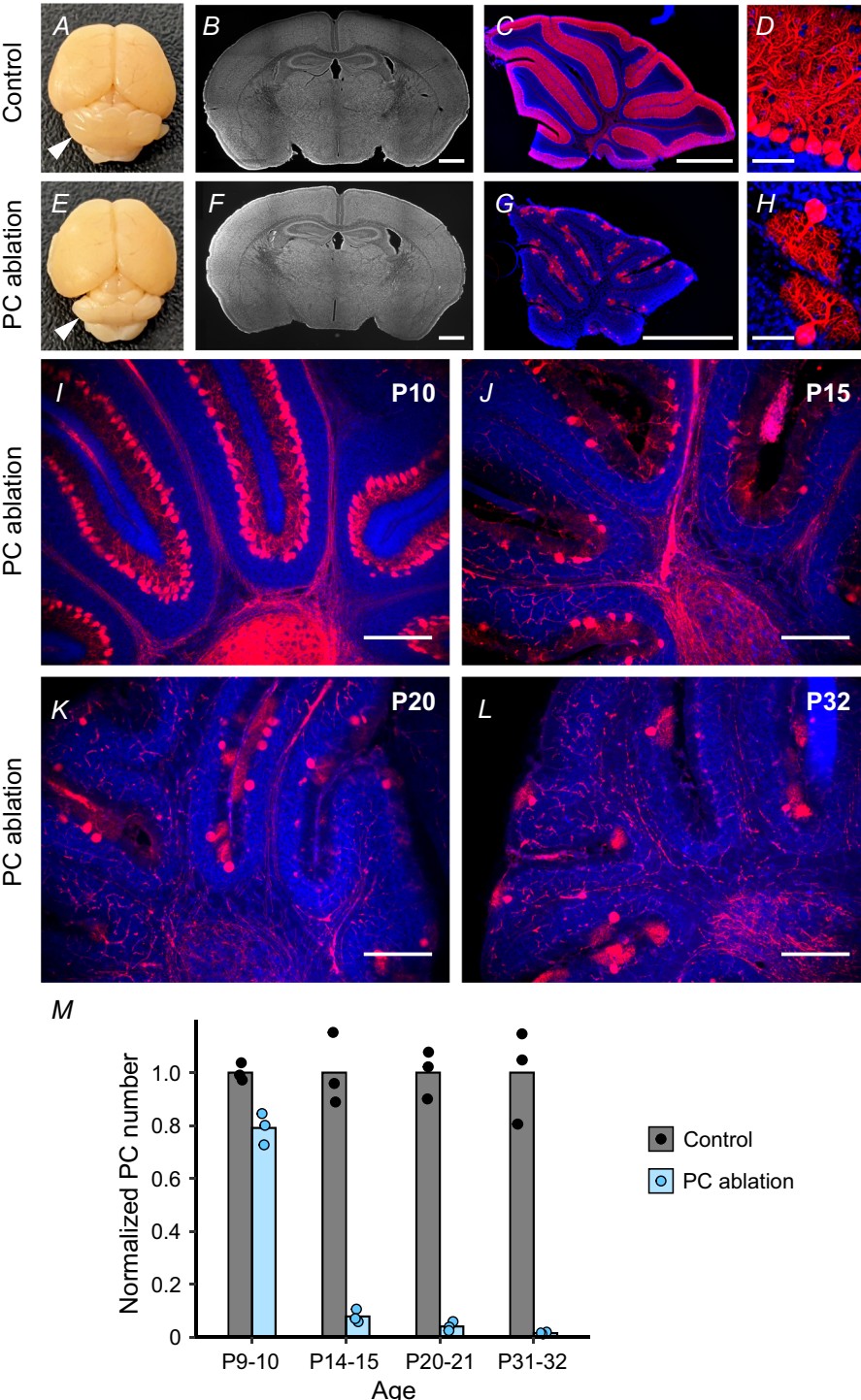

**Figure 1. Selective ablation of Purkinje cells (PCs)**
*A–D*, a control mouse carrying Cre-inducible DTA gene (*Eno2*$^{fsDTA/+}$). This single transgenic mouse does not carry the Cre gene; thus, DTA is not expressed. *E–H*, a double transgenic mouse carrying Cre and DTA genes (*Pcp2*$^{Cre/+}$ *Eno2*$^{fsDTA/+}$). The PC-specific promoter Pcp2 drives Cre expression in this mouse. Therefore, DTA is expressed only in the PCs and ablates them. *A* and *E*, the whole brain. The arrowheads indicate the cerebellum. Note that the cerebellum of the PC ablation mouse is substantially smaller than the control mouse. *B* and *F*, fluorescence Nissl stain of the forebrain. Scale bars = 1 mm. *C* and *G*, immunostaining of PCs with antibodies against a PC marker, calbindin-D28k (red), counterstained with DAPI (blue). Scale bars = 500 µm. *D* and *H*, higher magnification of (*C*) and (*G*), taken by a confocal microscope. Scale bars = 50 µm. *I–J*, PC ablation mice were killed at P10, P15, P20

and P32 and stained with anti-calbindin-D28k antibody (red) and fluorescence Nissl (blue). Scale bars = 200 μm.
*M*, bar graph showing the relative number of PCs in the PC ablation mice (*n* = 3 mice at each age) normalized to the control mice (*n* = 3 mice at each age). [Colour figure can be viewed at wileyonlinelibrary.com]

**Table 1. The summary of DCN recordings in the control and DTA mice**

| 2 weeks old | Control Mean (SD) | DTA Mean (SD) | Effect size (DTA – Control)* Observed [95% CI] |
|---|---|---|---|
| AP frequency (Hz) | 20.10 (12.44), *n* = 36 | 14.64 (10.91), *n* = 37 | −0.261 [−0.498, 0.0143] |
| Input resistance (MΩ) | 166.09 (85.09), *n* = 9 | 184.31 (94.58), *n* = 10 | 0.0222 [−0.533, 0.556] |
| Slope (Hz/(nA × MΩ)) | 1.15 (0.73), *n* = 9 | 1.31 (0.60), *n* = 10 | 0.222 [−0.378, 0.711] |
| **1.5-3 months old** | Control Mean (SD) | DTA Mean (SD) | Effect size (DTA − Control) * Observed [95% CI] |
| AP frequency (Hz) | 37.75 (28.21), *n* = 22 | 32.52 (39.51), *n* = 27 | −0.247 [−0.552, 0.0971] |
| Input resistance (MΩ) | 167.71 (86.62), *n* = 14 | 126.79 (79.20), *n* = 13 | −0.308 [−0.704, 0.176] |
| Slope (Hz/(nA × MΩ)) | 1.56 (0.89), *n* = 14 | 1.84 (0.41), *n* = 13 | 0.363 [−0.143, 0.725] |

*All effect sizes are Cliff's delta.
CI, confidence interval.

firing rate is lower in the DTA mice, losing presynaptic Purkinje cells did not substantially change the ability of DCN neurons to discharge spontaneous APs.

We next examined how DCN neurons respond to membrane depolarization using whole-cell current-clamp recordings (Fig. 2*E–J*). Consistent with previous studies, somatic injection of a depolarizing current step increased their AP firing rate with little to slight spike-frequency adaptation (Fig. 2*E* and *F*) (Beekhof et al., 2021; Huang & Uusisaari, 2013). Furthermore, the AP firing rate linearly increased with the amplitude of the current step. Because the degree of membrane depolarization depends on the amplitude of the injected current (command current: $I_{com}$) and the cell's input resistance ($R_{in}$), the product of $I_{com}$ and $R_{in}$ ($I_{com} \times R_{in}$) more adequately represents the degree of depolarization than $I_{com}$ alone. Therefore, as shown in a previous study (Huang & Uusisaari, 2013), we examined the relationship between the AP frequency and $I_{com} \times R_{in}$ for each cell and plotted the linear regression line (Fig. 2*G* and *H*). The slope of the regression line represents the responsiveness of the DCN neuron to membrane depolarization, which did not significantly differ between the control and DTA mice in either age group (Fig. 2*I* and *J*). These results suggest that losing presynaptic Purkinje cells does not change the responsiveness of DCN neurons to depolarizing stimuli.

### Organization of cerebellothalamic synapses

To study the effects of Purkinje cell loss in thalamic development, we first characterized the physiological properties of cerebellothalamic synapses in the VL under normal circumstances. The VL is a part of the motor thalamus, but the organization of long-range polysynaptic loops (i.e. cerebellum → thalamus → neocortex → pons → cerebellum) is similar regardless of whether the cerebellum connects to the motor or non-motor areas of the neocortex (D'Angelo & Casali, 2013; Kelly & Strick, 2003; Koziol et al., 2014). We injected AAV expressing channelrhodopsin-2-tdTomato (ChR2) into the DCN (Fig. 3*A*, left). The ChR2-expressing cerebellothalamic axons were optogenetically stimulated, and postsynaptic responses in the VL neurons were recorded under whole-cell voltage-clamp mode (Fig. 3*A*, right). All thalamic recordings in this study were conducted using young adult mice (1.5–3 months old).

Consistent with the previous studies, optogenetic stimulation of cerebellothalamic axons evoked a large EPSC, showing paired-pulse depression (Fig. 3*B*) (Gornati et al., 2018; Schäfer et al., 2021). This type of EPSC is characteristic of driver inputs to the primary sensory thalamus, such as retinal inputs to the lateral geniculate nucleus and somatosensory inputs to the ventral posteromedial nucleus (Bosch-Bouju et al., 2013; Reichova & Sherman, 2004; Sherman, 2005). These sensory driver inputs evoke EPSCs primarily through AMPA- and NMDA-type glutamate receptors, and the ratio of the EPSCs recorded at –70 mV *vs.* at +40 mV is routinely used to measure the developmental increase of the AMPA/NMDA ratio (Arsenault & Zhang, 2006; Chen & Regehr, 2000; Hooks & Chen, 2006; Takeuchi et al., 2014; Wang & Zhang, 2008).

To test whether the same approach can be used in cerebellothalamic synapses, we characterized the EPSC components at –70 mV and +40 mV in the VL of the

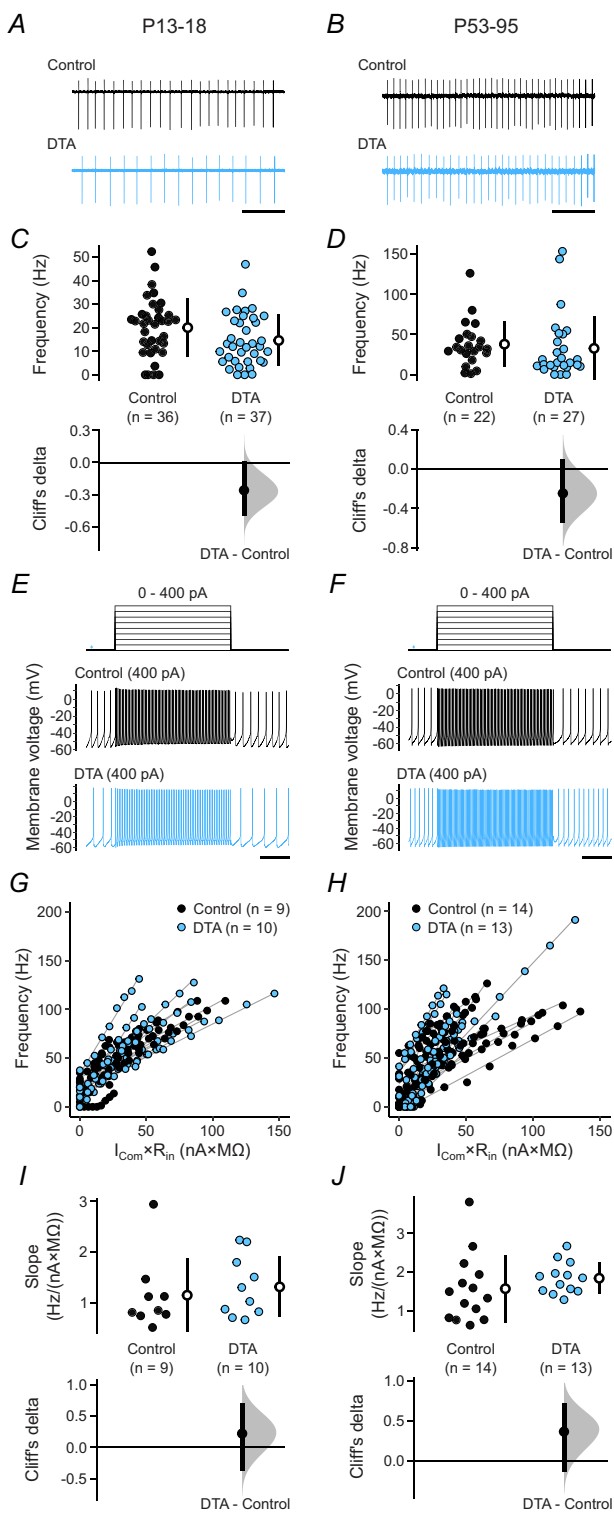

case) was estimated by bootstrap resampling. The top portion shows individual cells (coloured circles), the mean (open circles) and the SD (vertical lines) of each group. The grey curve in the bottom portion shows the resampled distribution of the effect size. The black circles and the thick vertical lines on the grey curves indicate the observed effect size and 95% confidence interval, respectively. The horizontal line indicates zero effect (i.e. no difference between the groups). The statistical significance ($P < 0.05$) is assessed whether the 95% confidence interval includes or excludes the zero line. All the estimation graphics in this study have the same structure. *E* and *F*, under whole-cell current-clamp mode, a depolarizing current step was injected into DCN neurons, and the evoked APs were recorded. Only the recordings with 400 pA current injection were shown, but each neuron received a series of 0–400 pA current with a 50 pA increment (top). Scale bars = 200 ms. *G* and *H*, the responsiveness of the DCN neurons to membrane depolarization was compared between the control and DTA mice, showing juvenile (*G*) and young adult mice (*H*), respectively. Each line represents the linear regression line for each cell, fitted to the relationship between the AP firing rate and $I_{com} \times R_{in}$. For the rationale, see the main text. *I* and *J*, the slope of the regression lines in (*G*) and (*H*) were compared between the control and DTA mice by bootstrap resampling; (*I*) corresponds to (*G*) (juvenile mice) and (*J*) corresponds to (*H*) (young adult mice). [Colour figure can be viewed at wileyonlinelibrary.com]

**Figure 2. Spontaneous and depolarization-evoked action potentials (APs) in the deep cerebellar nuclei (DCN)**
*A* and *B*, cell-attached patch-clamp recordings of spontaneous APs in DCN neurons in juvenile (*A*) and young adult (*B*) mice. Scale bars = 200 ms. *C* and *D*, the firing rate of the spontaneous APs was compared between the control and PC ablation (DTA) mice, showing juvenile (*C*) and young adult mice (*D*), respectively. The difference between the control and DTA mice (effect size, Cliff's delta in this

control mice (Fig. 3*C* and *D*). The AMPA-R antagonist NBQX (10 μM) blocked $94.0 \pm 0.6\%$ ($n = 3$ cells) of EPSCs at –70 mV, and the NMDA-R antagonist DL-APV (100 μM) blocked $92.1 \pm 1.2\%$ ($n = 3$ cells) of EPSCs at +40 mV. Although there was a small remaining current at both holding potentials, these data indicate that the EPSC ratio recorded at –70 mV *vs.* +40 mV in the VL can be used to measure the AMPA/NMDA ratio like the primary sensory thalamus.

As the AMPA/NMDA ratio increases during the maturation of the primary sensory thalamus, neuronal activity-dependent synaptic pruning occurs. In the primary visual and somatosensory thalamus, individual thalamic neurons are initially innervated by many sensory driver inputs, but they are subsequently pruned during the first few postnatal weeks (Arsenault & Zhang, 2006; Chen & Regehr, 2000; Hooks & Chen, 2006; Takeuchi et al., 2014; Wang & Zhang, 2008). Consequently, a mature neuron in these thalamic nuclei receives only a few sensory driver inputs. To examine whether the mature VL neurons exhibit a similar innervation pattern, we recorded AMPA currents (AMAP-EPSCs) in the control mice by gradually increasing the power density of the blue light (Fig. 3*E–M*). The gradual increase of stimulus intensity is a standard electrophysiological technique to quantify the number of presynaptic inputs innervating a postsynaptic neuron.

The gradual increase of light power density, from 0 to ∼40 mW mm$^{-2}$, yielded all-or-none EPSCs in some VL neurons, such that no EPSC was evoked when the light power density was below a threshold, and evoked EPSCs essentially unchanged above the threshold (Fig. 3*E–G*).

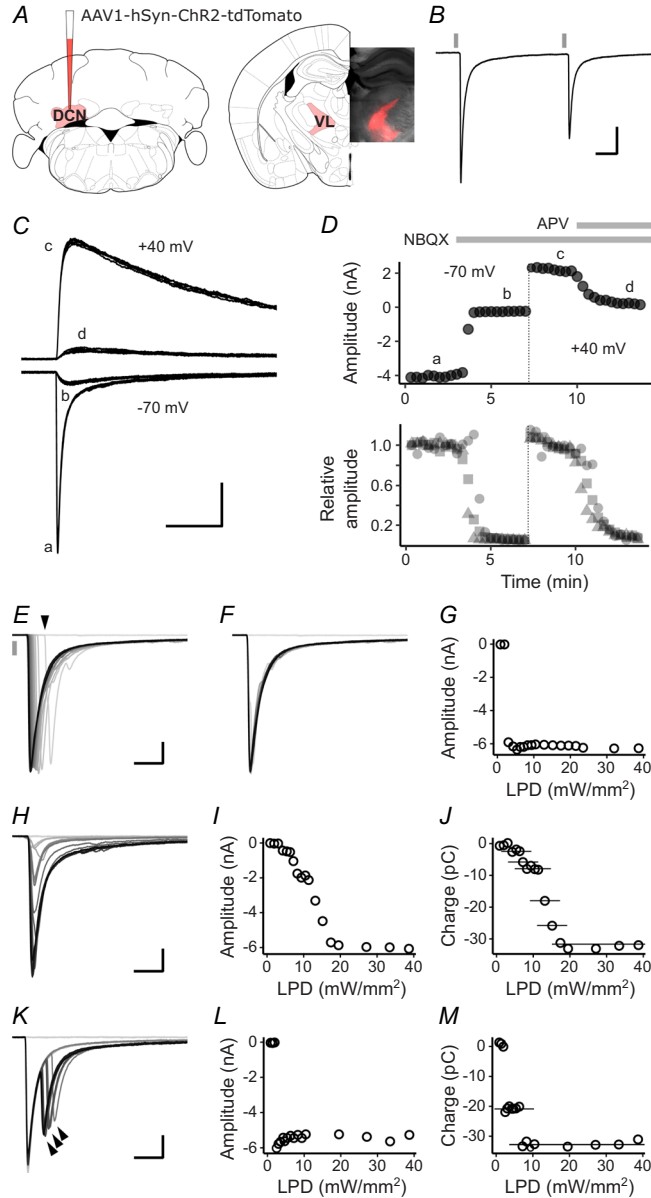

each neuron, represented by a different marker. *E–M*, VL neurons were photostimulated by gradually increased light power density, and the evoked currents were overlaid. The grey rectangles indicate the timing of photostimulation. Each current trace is colour-coded; a darker grey represents a higher light power density. Scale bars = 1 nA and 5 ms. *E*, an example of all-or-none EPSCs. The arrowhead indicates the onset of the EPSC evoked by the minimum light power density above the threshold. Note that the EPSC onsets shifted earlier as the light power density increased. *F*, the EPSC traces in (*E*) are aligned with the EPSC onsets. *G*, a scatter plot showing the relationship between the light power density (LPD) and the EPSC amplitude. *H–M*, representative neurons showing multiple discrete EPSC steps. *H–J* represent one neuron and *K–M* represent another neuron. The EPSC traces are aligned with the EPSC onsets (*H* and *K*). Scatter plots show the relationship between the light power density (LPD) and the EPSC amplitude (*I* and *L*) or charge transfer (*J* and *M*). The horizontal lines in (*J*) and (*M*) indicate the discrete EPSC steps. Note that the neuron in (*K*) initially showed one EPSC, but, as the light power density increased, a delayed EPSC was evoked in addition to the first EPSC. The onset of the delayed EPSC shifted earlier as the light power density further increased (arrowheads). All recordings in this figure were performed using control mice. [Colour figure can be viewed at wileyonlinelibrary.com]

**Figure 3. Recording of cerebellothalamic synapses**
*A*, left: AAV1 encoding ChR2-tdTomato under the control of the human synapsin promoter (hSyn) was injected into the DCN. Right: the ChR2-tdTomato expressing cerebellothalamic axons (red) in the ventrolateral thalamus (VL), shown with the corresponding region of the mouse brain atlas (Paxinos & Franklin, 2001). *B*, whole-cell voltage-clamp recording from a VL neuron at −70 mV. A pair of light pulses (grey rectangles) evoked large EPSCs, showing paired-pulse depression. Scale bars = 1 nA and 20 ms. *C*, EPSCs in another VL neuron. The holding potential was initially −70 mV (bottom), then switched to +40 mV (top). AMPA receptor antagonist NBQX (10 μM) and NMDA receptor antagonist DL-APV (100 μM) were perfused during the recording. Scale bars = 1 nA and 20 ms. *D*, the time course of the recording with antagonist application (grey bars). The vertical dashed lines indicate the timing of the holding potential switch. Top, representing the neuron shown in (*C*). The marks 'a', 'b', 'c' and 'd' indicate the timing when the corresponding EPSCs in (*C*) were recorded. Bottom, combining the recordings from three cells. The EPSC amplitude was normalized to the values before the NBQX application (−70 mV) and DL-APV application (+40 mV) in

The interval from photostimulation to EPSC onset always became shorter as the light power density increased (Fig. 3*E*), presumably because stronger light activated more ChR2, causing faster depolarization of presynaptic terminals. When aligned with the onset, these EPSCs were identical (Fig. 3*F*), indicating that the recorded neuron was innervated by only one ChR2-expressing cerebellothalamic axon.

In other VL neurons, several discrete steps of EPSCs (Fig. 3*H–J*) or additional EPSCs with delayed onset (Fig. 3*K–M*) were evoked as the light power density increased. The stepwise, non-continuous increase of EPSC amplitudes indicates that the recorded neuron was innervated by a relatively small number of ChR2-expressing cerebellothalamic axons; more of them were activated by stronger light (Fig. 3*H–J*). If the kinetics of ChR2-mediated presynaptic depolarization substantially differs among the axons, their EPSC peaks may not align, as shown in Fig. 3*K*. In these cases, the charge transfer of EPSCs represents the number of inputs more precisely than the EPSC amplitude (Fig. 3*L* and *M*). Therefore, we used the number of discrete steps in the EPSC charge transfer as the number of ChR2-expressing cerebellothalamic axons innervating to the recorded neurons (Fig. 3*J* and *M*).

Because we cannot stimulate uninfected neurons using light, we probably underestimated the number of inputs in some cells. However, even if a neuron was innervated by only one or two ChR2-expressing cerebellothalamic axons (Fig. 3*E* and *K*), large EPSCs (∼ −6 nA) could be evoked, which were comparable to a neuron innervated by six ChR2-expressing cerebellothalamic axons (Fig. 3*H*, the most EPSC steps in our data set). Therefore, a

**Table 2. The summary of VL voltage-clamp recordings in the control and DTA mice**

| | Control, $n = 17$ Mean (SD) | DTA, $n = 20$ Mean (SD) | Effect size (DTA – Control) * Observed [95% CI] |
|---|---|---|---|
| Series resistance (MΩ) | 6.19 (1.29) | 6.38 (1.71) | 0.0353 [−0.353, 0.418] |
| # of EPSC steps | 2.76 (1.09) | 3.40 (0.94) | 0.306 [−0.05, 0.619] |
| AMPA amplitude (nA) | 3.45 (1.66) | 5.07 (2.56) | 0.412 [0.00588, 0.682] |
| AMPA charge (pC) | 17.84 (10.35) | 27.77 (12.72) | 0.482 [0.0706, 0.765] |
| NMDA amplitude (nA) | 1.67 (0.91) | 2.56 (1.03) | 0.494 [0.0882. 0.759] |
| AMPA/NMDA ratio | 2.29 (0.89) | 2.02 (0.64) | *−0.348 [−1.02, 0.368]* |
| 10–90% rise time (ms) | 0.47 (0.18) | 0.46 (0.14) | 0.0206 [−0.371, 0.422] |
| Weighted decay constant (ms) | 4.32 (1.61) | 4.64 (0.78) | *0.261 [−0.476, 1.1]* |

*The regular and italic effect sizes represent Cliff's delta and Cohen's *d*, respectively.
CI, confidence interval.

small number of EPSC steps does not mean insufficient stimulation of cerebellothalamic axons. Although it is difficult to quantify the number of inputs per cell accurately, our data suggest that the VL neurons receive a relatively small number of cerebellar inputs, comparable to the primary sensory thalamus receiving the sensory driver inputs.

### Purkinje cell ablation strengthened cerebellothalamic synapses

In the primary visual and somatosensory thalamus, sensory deprivation alters the synaptic strength, the AMPA/NMDA ratio and the pruning of redundant driver inputs (Hooks & Chen, 2006; Takeuchi et al., 2014; Wang & Zhang, 2008). Considering the similar synaptic organization between the VL and these sensory thalamic nuclei, we hypothesized that early cerebellar perturbation alters the synaptic properties in the VL. To test this hypothesis, we compared the Purkinje cell ablation mice with their littermate control, using whole-cell patch-clamp recordings and optogenetic stimulation of cerebellothalamic axons.

We used the single transgenic mice ($Pcp2^{Cre/+}$ or $Eno2^{fsDTA/+}$) as a control group and compared them with their double transgenic littermates ($Pcp2^{Cre/+}$ $Eno2^{fsDTA/+}$), which lose ∼92% of Purkinje cells within the first two postnatal weeks (Fig. 1). The double transgenic mice (DTA mice) are slightly lighter than the control littermates and show moderate ataxia. However, despite the substantial cerebellar atrophy, they were healthy and showed no signs of health decline until we killed them for electrophysiological analysis. This is consistent with previous studies of naturally occurring mouse mutation, Purkinje cell degeneration (*pcd*), which causes the degeneration of 95–99% of Purkinje cells by the end of the fourth postnatal week (Chakrabarti et al., 2009; Mullen et al., 1976).

The electrophysiological properties of cerebellothalamic synapses in the control and DTA mice are shown in Fig. 4 and Table 2. The observed differences between the groups (Cliff's delta or Cohen's *d* effect size) were negligible to modest compared to their distribution ranges for the number of discrete EPSC steps (Fig. 4C), the AMPA/NMDA ratio (Fig. 4H) and the kinetics of AMPA-EPSCs (Fig. 4I and J). Their estimated 95% confidence interval included zero, indicating that these synaptic properties were not significantly affected by Purkinje cell ablation.

The paired-pulse ratio, which is primarily determined by the presynaptic release probability, was not affected either (Fig. 4K and L). Although the ratio was dependent on the interstimulus intervals (mixed ANOVA: $F_{1.23,43.13} = 207.259$, $P < 0.001$), there was no significant difference between the control and DTA mice ($F_{1,35} = 1.073$, $P = 0.307$), nor any significant interaction between the mouse groups and the interstimulus intervals ($F_{1.23,43.13} = 0.750$, $P = 0.418$).

On the other hand, the synaptic strength was higher in the DTA mice than in the control mice. The amplitude of AMPA-EPSCs was $3.45 \pm 1.66$ nA ($n = 17$ cells) in the control mice and $5.07 \pm 2.56$ nA ($n = 20$ cells) in the DTA mice (Fig. 4E). The observed effect size was modest but statistically significant. Because the 95% confidence interval excluded zero marginally, the effect is somewhat uncertain (Table 2). However, the other indicators of the synaptic strength (i.e. the charge transfer of AMPA-EPSCs and the amplitude of NMDA-EPSCs) were also larger in the DTA mice and showed more prominent effects (Fig. 4F and G). These results collectively suggest that losing Purkinje cells in the developing cerebellum strengthens cerebellothalamic synapses without significantly affecting the AMPA/NMDA ratio or presynaptic release probability.

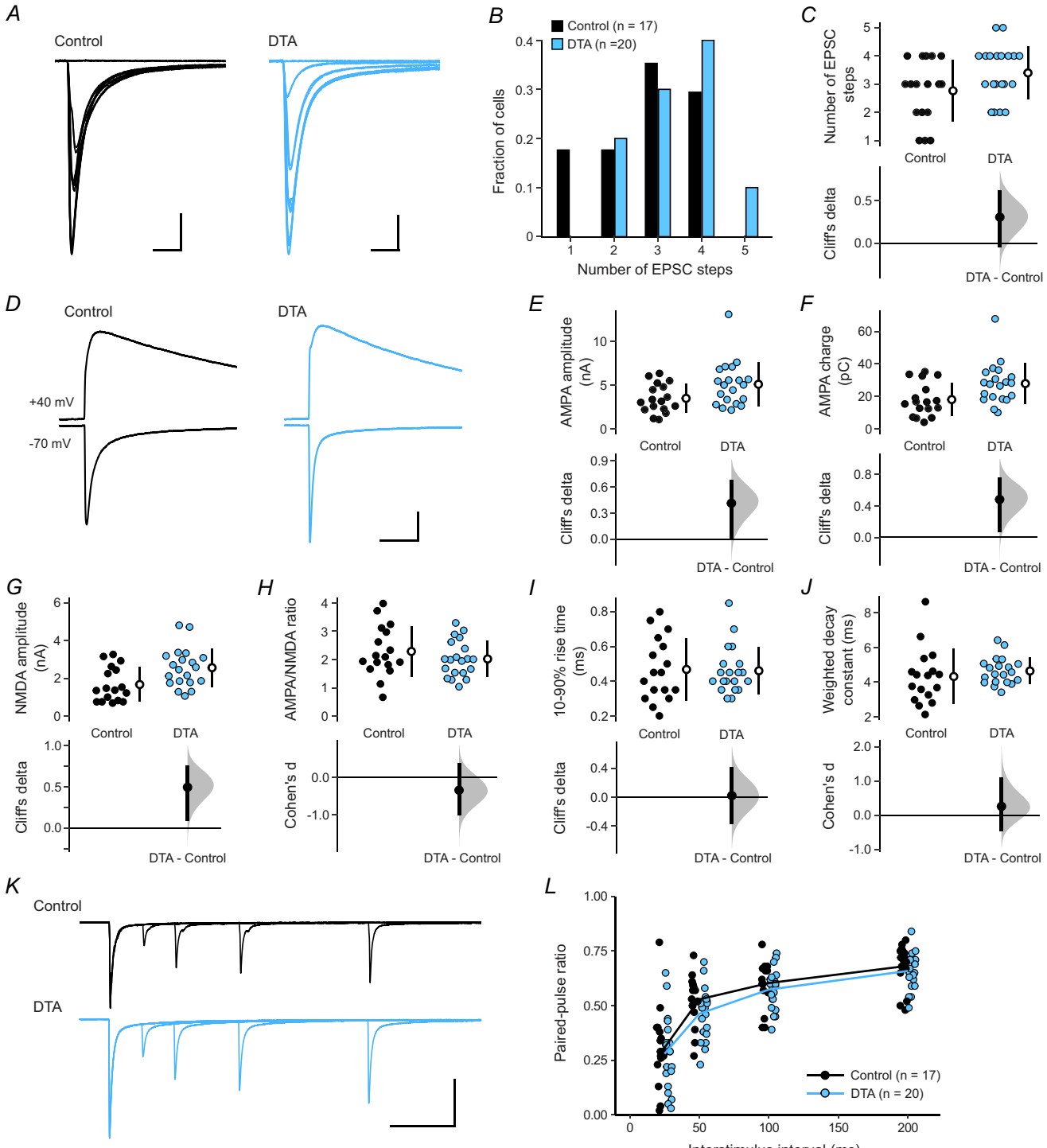

**Figure 4. Physiological properties of cerebellothalamic synapses in PC ablation mice**

*A*, AMPA-EPSCs evoked by gradually increased light power density. Scale bars = 1 nA and 5 ms. *B* and *C*, the number of EPSC steps was compared between the control and PC ablation (DTA) mice. The effect size (Cliff's delta in this case) was estimated by bootstrap resampling (DTA – Control in *C*). The statistical significance was assessed as described in Fig. 2. *D*, EPSCs recorded from the same cells at −70 mV and +40 mV. Scale bars = 1 nA and 20 ms. *E–J*, the amplitude (*E*) and charge transfer (*F*) of AMPA-EPSCs, the amplitude of NMDA-EPSCs (*G*), the AMPA/NMDA ratio (*H*), the 10–90% rise time (*I*) and the weighted decay constant (*J*) of AMPA-EPSCs were compared between the control and DTA mice by bootstrap resampling. *K*, paired-pulse depression with the interpulse interval of 25, 50, 100 and 200 ms. Scale bars = 2 nA and 50 ms. *L*, the paired-pulse ratio was compared between the groups. [Colour figure can be viewed at wileyonlinelibrary.com]

**Table 3. The summary of VL current-clamp recordings in the control and DTA mice**

|  | Control, *n* = 20 Mean (SD) | DTA, *n* = 16 Mean (SD) | Effect size (DTA − Control)* Observed [95% CI] |
|---|---|---|---|
| Resting potential (mV) | −69.45 (3.96) | −71.89 (3.45) | *−0.651 [−1.31, 0.0286]* |
| Input resistance (MΩ) | 156.15 (35.79) | 172.97 (56.64) | *0.364 [−0.351, 1.07]* |
| Max AP number | 4.45 (2.33) | 6.25 (2.57) | *0.739 [0.0156, 1.41]* |
| AP threshold (mV) | −44.71 (4.83) | −45.43 (3.86) | *−0.162 [−0.803, 0.552]* |
| AP amplitude (mV) | 68.11 (5.97) | 70.37 (6.72) | 0.178 [−0.231. 0.562] |
| AP half-width (ms) | 0.57 (0.07) | 0.55 (0.13) | −0.312 [−0.655, 0.116] |

*The regular and italic effect sizes represent Cliff's delta and Cohen's *d*, respectively.
CI, confidence interval.

## Purkinje cell ablation enhanced synaptically-evoked thalamic APs

Cerebellar activities are relayed to the cerebral cortex as suprathreshold activities of thalamic neurons. Although the loss of Purkinje cells strengthens the cerebellothalamic inputs, the extent to which it alters thalamic outputs is yet to be determined. Therefore, we compared intrinsic membrane properties and excitabilities of VL neurons between the control and DTA mice (Fig. 5 and Table 3).

Under whole-cell current-clamp mode, each VL neuron was excited by two different methods: synaptic stimulation by optogenetics (Fig. 5*A* and *B*) and somatic injection of depolarizing current (Fig. 5*F* and *G*). Although both methods evoked APs in almost all VL neurons, synaptic stimulation elicited steep EPSPs, often making the AP threshold detection unreliable (Fig. 5*A* inset). Therefore, we measured the AP threshold, AP amplitude and AP half-width by somatic current injection (Fig. 5*H–J*).

The resting membrane potential was measured in both methods before synaptic stimulation or current injection and then the two values were averaged (Fig. 5*C*). The input resistance was measured by injecting hyperpolarizing current (Fig. 5*D*). The maximum number of synaptically evoked APs was measured by optogenetic stimulation using the highest light power density (∼40 mW mm$^{-2}$). As shown in Fig. 3, this light power density always evoked maximum EPSCs in our experimental condition, ensuring that all ChR2-expressing cerebellothalamic inputs were stimulated.

The observed effect sizes were negligible to modest compared to their distribution ranges for all the measured properties. In particular, the 95% confidence intervals of the estimated effect sizes clearly included zero for the input resistance (Fig. 5*D*), AP threshold (Fig. 5*H*), AP amplitude (Fig. 5*I*) and AP half-width (Fig. 5*J*). Therefore, losing Purkinje cells probably does not change the subthreshold or suprathreshold excitability of VL neurons.

On the other hand, the effect size distribution was substantially biased toward negative for the resting potential (Fig. 5*C*). Although the 95% confidence interval marginally included zero (Table 3), it does not certainly rule out the possibility that the resting potential is slightly hyperpolarized in the DTA mice.

Previous studies showed that VL neurons discharge more APs when they are more hyperpolarized before being stimulated (Contreras & Steriade, 1995; Llinás & Steriade, 2006; Schäfer et al., 2021); thus, the resting potential influences the AP firing pattern. The maximum number of synaptically-evoked APs was 4.45 ± 2.33 (*n* = 20 cells) in the control mice and 6.25 ± 2.57 (*n* = 16 cells) in the DTA mice (Fig. 5*E* and Table 3). The effect size distribution was substantially biased toward positive, and the 95% confidence interval marginally excluded zero. Therefore, the difference is statistically significant, suggesting that synaptic stimulation probably evokes more APs in the DTA mice than in the control mice. It remains unclear whether the increased AP number in the DTA mice is primarily the result of increased AMPA currents (Fig. 4*E* and *F*) or the possible difference in the membrane potential (Fig. 5*C*). Nevertheless, these results suggest that the loss of Purkinje cells strengthens both thalamic input and outputs.

## Purkinje cell-specific homozygous deletion of *Tsc1* strengthened cerebellothalamic synapses

Although the Purkinje cell ablation mouse is a powerful model for studying the cerebellar contribution to thalamic development, the degree of cell loss does not mimic common neurological conditions. To perturb the developing cerebellum in a manner that more closely models human diseases, we used the *Pcp2^{Cre/+}* mouse to delete the tuberous sclerosis (TSC) complex subunit 1 (*Tsc1*) gene in Purkinje cells, which is an established mouse model for autism (Tsai et al., 2012). Because TSC1 is a negative regulator of mammalian target of rapamycin (mTOR), the lack of TSC1 can be rescued by systemic injection of mTOR inhibitor rapamycin. Administration of rapamycin from P7 to P35 rescued impaired social behaviours of *Tsc1* deletion mice (Gibson et al., 2022),

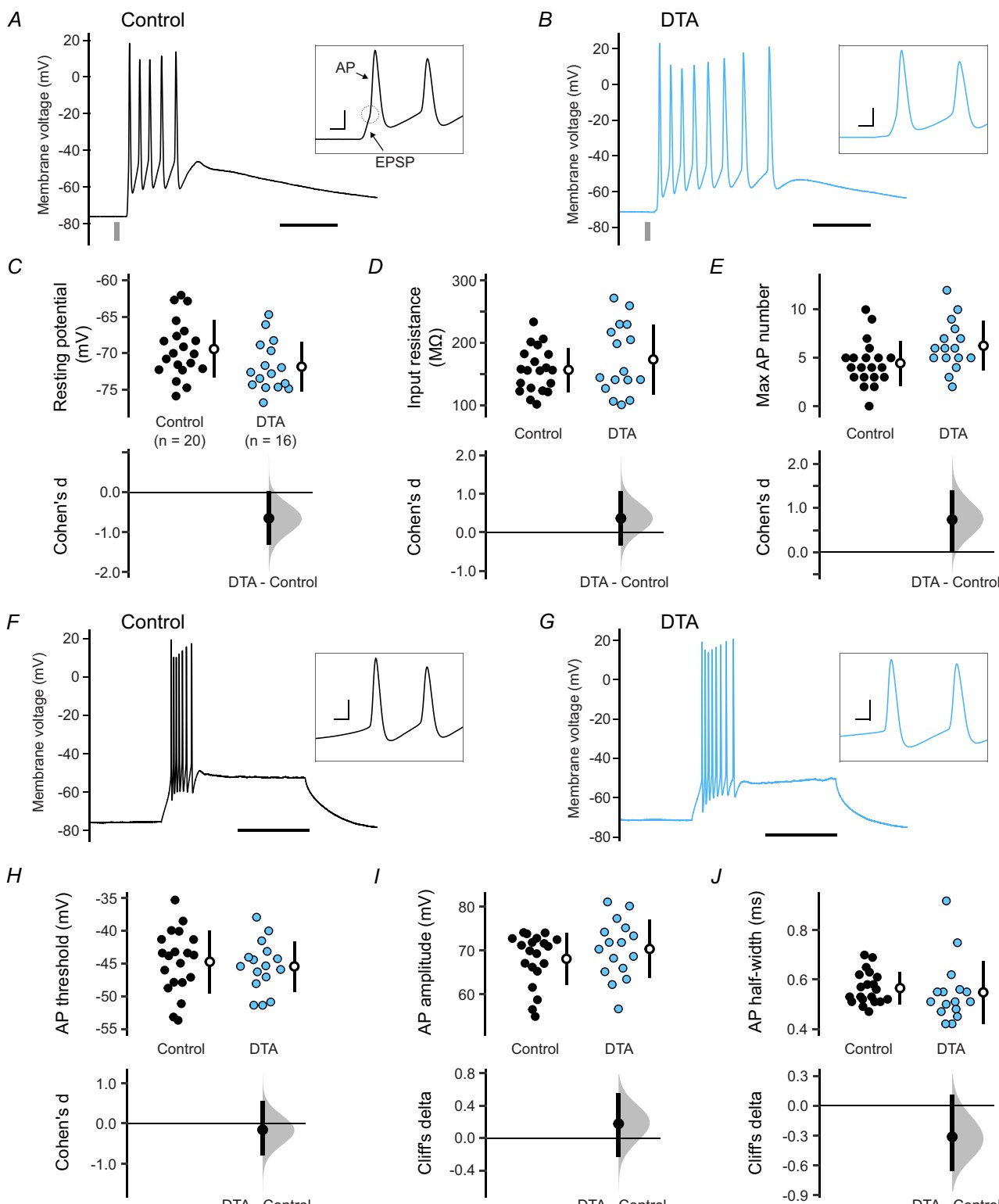

**Figure 5. Membrane properties and excitability of VL neurons in PC ablation mice**
*A* and *B*, APs evoked in VL neurons by photostimulation of cerebellothalamic axons in the control (*A*) and PC ablation (DTA) mice (*B*). The grey rectangles indicate the timing of photostimulation. Scale bar = 20 ms. Insets: the first two APs are magnified along the time scale. Scale bars = 20 mV and 1 ms. Note that the first AP was evoked on the rising phase of EPSP, making the AP threshold detection unreliable (dashed circle). *C–E*, the resting membrane potential (*C*), the input resistance (*D*) and the maximum number of synaptically evoked APs (*E*) were compared between the control and DTA mice by bootstrap resampling. *F* and *G*, APs evoked in VL neurons by somatic

current injection in the control (*F*) and DTA mice (*G*). Scale bar = 100 ms. Insets: the first two APs are magnified along the time scale. Scale bars = 20 mV and 1 ms. *H–J*, the AP threshold (*H*), amplitude (*I*) and half-width (*J*) were compared between the control and DTA mice by bootstrap resampling. [Colour figure can be viewed at wileyonlinelibrary.com]

**Table 4. The summary of VL voltage-clamp recordings in the control and TSC mice**

| | Control (Cont) *n* = 34 Mean (SD) | TSC +/– (Hemi) *n* = 16 Mean (SD) | TSC –/– (Null) *n* = 18 Mean (SD) | Effect size (TSC – Control) * Observed [97.5% CI] | |
|---|---|---|---|---|---|
| | | | | Hemi – Cont | Null – Cont |
| Series resistance (MΩ) | 7.28 (1.39) | 6.25 (1.45) | 6.83 (0.91) | *−0.734* [*−1.41, 0.11*] | *−0.364* [*−0.924, 0.239*] |
| Number of EPSC steps | 3.09 (1.16) | 3.00 (0.97) | 3.28 (1.13) | *−0.0551* [*−0.404, 0.318*] | *0.0948* [*−0.278, 0.448*] |
| AMPA amplitude (nA) | 3.92 (1.67) | 4.21 (1.34) | 4.93 (1.59) | *0.187* [*−0.468, 0.81*] | *0.618* [*−0.0669, 1.28*] |
| AMPA charge (pC) | 19.93 (9.38) | 18.56 (5.63) | 27.09 (10.09) | *−0.164* [*−0.763, 0.423*] | *0.743* [*0.032, 1.41*] |
| ˄ NMDA amplitude (nA) | 1.78 (0.88) | 1.93 (0.72) | 2.52 (0.68) | *0.180* [*−0.461, 0.843*] | *0.911* [*0.177, 1.58*] |
| ˄ AMPA/NMDA ratio | 2.44 (0.77) | 2.25 (0.74) | 1.97 (0.66) | *−0.261* [*−0.916, 0.478*] | *−0.64* [*−1.26, 0.0627*] |
| 10–90% rise time (ms) | 0.40 (0.10) | 0.46 (0.14) | 0.41 (0.11) | *0.276* [*−0.145, 0.64*] | *0.0784* [*−0.286, 0.436*] |
| Weighted decay constant (ms) | 4.31 (1.62) | 3.74 (1.18) | 4.78 (1.52) | *−0.199* [*−0.576, 0.196*] | *0.301* [*−0.127, 0.641*] |

*The regular and italic effect sizes represent Cliff's delta and Cohen's *d*, respectively.
˄ Control (*n* = 33), TSC +/– (*n* = 15), TSC –/– (*n* = 17) for these two measurements.
CI, confidence interval.

suggesting that TSC1 in Purkinje cells is crucial for a developmental period. This makes the *Tsc1* deletion model particularly suitable for the present study.

We used the single transgenic mice (*Pcp2*^*Cre/+*^, *Tsc1*^*flox/+*^ and *Tsc1*^*flox/flox*^) as a control group and compared them with their double transgenic littermates for hemizygous (*Pcp2*^*Cre/+*^ *Tsc1*^*flox/+*^) and homozygous (*Pcp2*^*Cre/+*^ *Tsc1*^*flox/flox*^) *Tsc1* deletion. Previous studies showed that hemizygous deletion of *Tsc1* reduces the firing rate of Purkinje cells, and the mutant animals exhibit impaired social behaviours and associative sensory learning (Gibson et al., 2022; Kloth et al., 2015; Tsai et al., 2012, 2018). Homozygous deletion causes more robust phenotypes, including motor deficits (Gibson et al., 2022; Tsai et al., 2012, 2018). Although Purkinje cell loss does not occur with hemizygous *Tsc1* deletion at least until 4 months old, the cell loss begins after 1.5 months old with homozygous deletion (Tsai et al., 2012).

We recorded EPSCs from the VL neurons in the control and *Tsc1* deletion mice using whole-cell patch-clamp recordings and optogenetic stimulation of cerebellothalamic axons (Fig. 6 and Table 4). The synaptic properties were compared between the control *vs. Tsc1* hemizygous deletion mice (Cont *vs.* Hemi) and

the control *vs. Tsc1* homozygous deletion mice (Cont *vs.* Null). The observed effect sizes between the groups (Hemi – Cont and Null – Cont) were negligible to modest compared to their distribution ranges for all measured properties except for the amplitude of NMDA-EPSCs (Fig. 6). In particular, the 97.5% confidence intervals clearly included zero for the number of discrete EPSC steps (Fig. 6*C*) and the kinetics of AMPA-EPSCs (Fig. 6*I* and *J*), suggesting that these synaptic properties were unaffected by *Tsc1* deletion.

On the other hand, the 97.5% confidence interval (Null – Cont) marginally included zero for the AMPA/NMDA ratio (Fig. 6*H* and Table 4). Although the effect missed the statistical significance, it does not rule out the possibility that the AMPA/NMDA ratio is lower in the *Tsc1* homozygous deletion mice. It is worth noting that the effect size distribution was also biased toward lower AMPA/NMDA ratios in the DTA mice, although to a lesser extent (Fig. 4*H*).

The paired-pulse ratio was not affected (Fig. 6*K* and *L*). While the ratio was dependent on the inter-stimulus intervals (mixed ANOVA, $F_{1.47,95.28} = 338.773$, $P < 0.001$), there was no significant difference between the control and *Tsc1* deletion mice (mixed ANOVA,

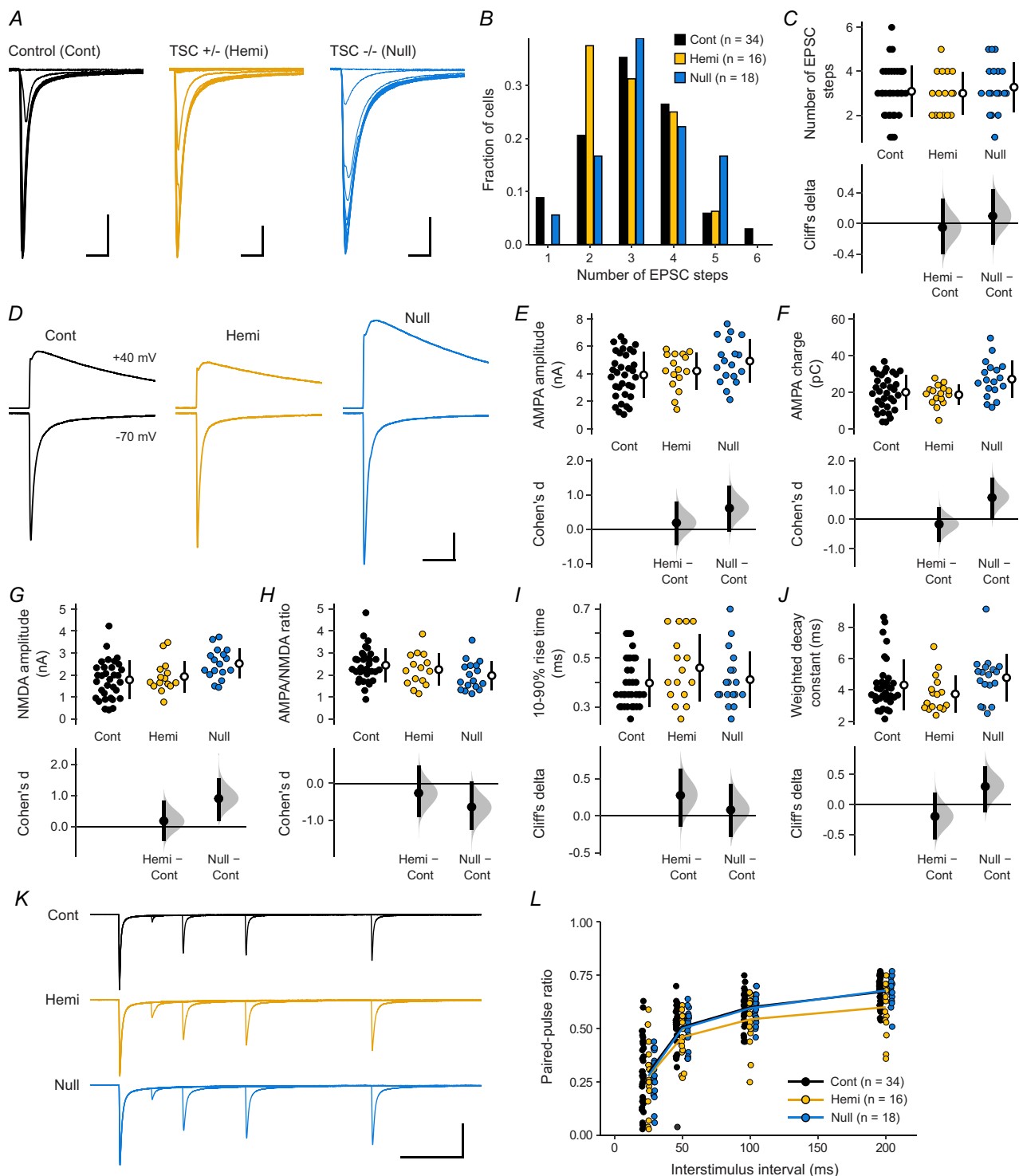

**Figure 6. Physiological properties of cerebellothalamic synapses in *Tsc1* deletion mice**

*A*, AMPA-EPSCs evoked by gradually increased light power density in the control mice (Cont), PC-specific hemizygous *Tsc1* deletion mice (TSC +/–, Hemi) and PC-specific homozygous *Tsc1* deletion mice (TSC –/–, Null). Scale bars = 1 nA and 5 ms. *B* and *C*, the number of EPSC steps was compared between Cont, Hemi, and Null. The effect size (Cliff's delta in this case) was estimated between Cont *vs*. Hemi (Hemi – Cont) and Cont *vs*. Null (Null – Cont) by bootstrap resampling (*C*). Unlike the previous figures, the thick vertical lines on the grey curves indicate the 97.5% confidence interval. *D*, EPSCs recorded from the same cells at –70 mV and +40 mV. Scale bars = 1 nA and 20 ms. *E–J*, the amplitude (*E*) and charge transfer (*F*) of AMPA-EPSCs, the amplitude of NMDA-EPSCs (*G*), the AMPA/NMDA ratio (*H*), the 10–90% rise time (*I*) and the weighted decay constant (*J*) of AMPA-EPSCs were

compared between Cont *vs*. Hemi and Cont *vs*. Null by bootstrap resampling. Note that one cell in each group was lost before NMDA-EPSCs were recorded. Therefore, the measurements requiring NMDA-EPSCs (*G* and *H*) have one less cell in each group. *K*, paired-pulse depression with the interpulse interval of 25, 50, 100 and 200 ms. Scale bars = 2 nA and 50 ms. *L*, the paired-pulse ratio was compared between the groups. [Colour figure can be viewed at wileyonlinelibrary.com]

**Table 5. The summary of VL current-clamp recordings in the control and TSC mice**

| | Control (Cont) $n = 31$ Mean (SD) | TSC +/− (Hemi) $n = 19$ Mean (SD) | TSC −/− (Null) $n = 19$ Mean (SD) | Effect size (TSC − Control) * Observed [97.5% CI] | |
| --- | --- | --- | --- | --- | --- |
| | | | | Hemi − Cont | Null − Cont |
| Resting potential (mV) | −68.03 (3.47) | −69.93 (2.33) | −69.73 (2.53) | −0.328 [−0.637, 0.0832] | −0.243 [−0.565, 0.161] |
| Input resistance (MΩ) | 187.90 (72.07) | 166.85 (53.79) | 152.65 (33.91) | −0.171 [−0.535, 0.212] | −0.263 [−0.582, 0.11] |
| Max AP number | 5.06 (2.21) | 5.74 (2.10) | 6.32 (2.14) | *0.310 [−0.38, 0.99]* | *0.574 [−0.121, 1.2]* |
| AP threshold (mV) | −43.56 (4.63) | −45.68 (4.93) | −43.33 (4.34) | −0.389 [−0.684, −0.0017] | 0.00849 [−0.368, 0.365] |
| AP amplitude (mV) | 71.21 (5.42) | 71.70 (6.93) | 66.92 (6.99) | 0.222 [−0.185, 0.555] | −0.343 [−0.667, 0.0404] |
| AP half-width (ms) | 0.53 (0.09) | 0.57 (0.10) | 0.54 (0.06) | *0.431 [−0.269, 1.2]* | *0.238 [−0.434, 0.93]* |

*The regular and italic effect sizes represent Cliff's delta and Cohen's *d*, respectively.
CI, confidence interval.

$F_{2,65} = 1.633$, $P = 0.203$), nor any significant interaction between the mouse groups and the interstimulus intervals ($F_{2.93,95.28} = 0.953$, $P = 0.417$). Therefore, similar to Purkinje cell ablation, *Tsc1* deletion does not alter presynaptic release probability.

Regarding the synaptic strength indicators (AMPA-EPSCs amplitude, AMPA-EPSCs charge transfer and NMDA-EPSCs amplitude), the Null − Cont effect size distributions were substantially biased toward positive (Fig. 6*E–G*). Although the 97.5% confidence interval for the AMPA-EPSCs amplitude marginally included zero, the confidence intervals for the AMPA-EPSCs charge transfer and the NMDA-EPSCs amplitude excluded zero (Table 4). In particular, the NMDA-EPSCs amplitude showed a prominent effect (Fig. 6*G*). These effects were not seen in the hemizygous deletion mice (Fig. 6*E–G*, Hemi − Cont). Considering that the charge transfer of AMPA-EPSCs more accurately represents the total synaptic currents than the EPSC amplitude (Fig. 3*K–M*), these results suggest that homozygous, not hemizygous, deletion of *Tsc1* strengthens cerebellothalamic synapses.

### Purkinje cell-specific deletion of *Tsc1* did not significantly alter thalamic excitability

To examine the extent to which *Tsc1* deletion alters thalamic outputs, we performed the same whole-cell current-clamp recordings used to analyse the DTA mice. The intrinsic membrane properties and excitabilities of

VL neurons were compared between the control and *Tsc1* deletion mice (Fig. 7 and Table 5).

The observed effect sizes for all the measured properties were negligible to modest compared to their distribution ranges. The effect was statistically significant only for the AP threshold between the control and the *Tsc1* hemizygous deletion mice (Fig. 7*J*, Hemi − Cont). However, the confidence interval was the closest to zero among all the statistically significant cases in this study (Table 6). Furthermore, the DTA mice and the *Tsc1* homozygous deletion mice showed a negligible, non-significant effect for the AP threshold (Fig. 5*H* and Fig. 7*J*). Although this result is difficult to interpret, we speculate that the effect was probably the result of an accidentally occurring sampling bias.

Overall, our data suggest that *Tsc1* deletion does not significantly alter the intrinsic membrane properties or excitabilities of VL neurons. Still, consistent with the DTA mice, the effect size distributions were largely biased toward lower resting potentials (Fig. 7*D*) and more synaptically-evoked APs (Fig. 7*F*) for both *Tsc1* hemizygous and homozygous deletion mice.

### Stronger cerebellothalamic inputs evoke more APs in the VL

Purkinje cell ablation and Purkinje cell-specific homozygous deletion of *Tsc1* both strengthened cerebellothalamic synapses. However, the enhanced

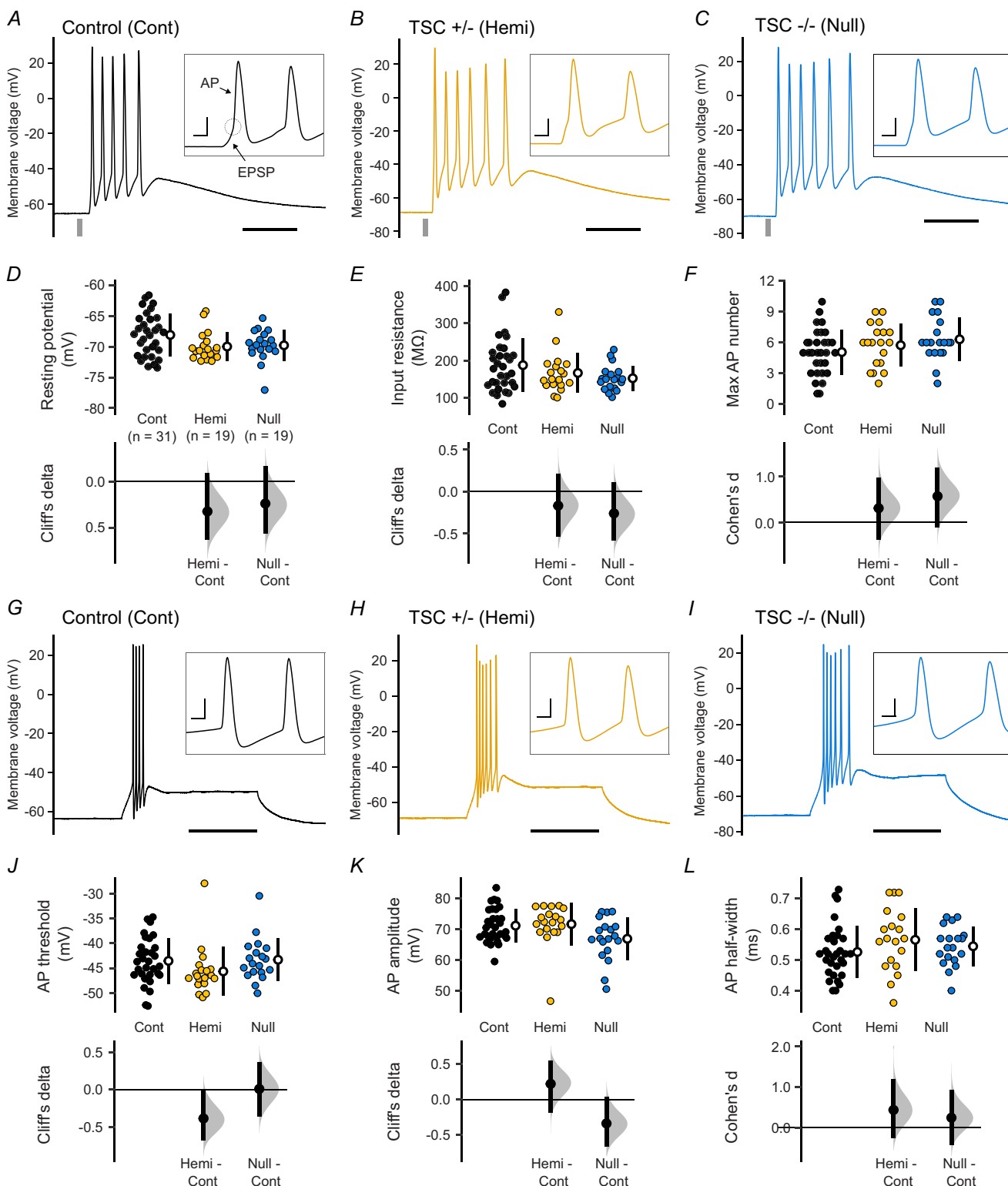

**Figure 7. Membrane properties and excitability of VL neurons in *Tsc1* deletion mice**
*A–C*, APs evoked in VL neurons by photostimulation of cerebellothalamic axons in the control mice (*A*, Cont), PC-specific hemizygous *Tsc1* deletion mice (*B*, TSC +/–, Hemi) and PC-specific homozygous *Tsc1* deletion mice (*C*, TSC –/–, Null). The grey rectangles indicate the timing of photostimulation. Scale bar = 20 ms. Insets: the first two APs are magnified along the time scale. Scale bars = 20 mV and 1 ms. Note that the first AP was evoked on the rising phase of EPSP, making the AP threshold detection unreliable (dashed circle). *D–F*, the resting membrane

potential (*D*), the input resistance (*E*) and the maximum number of synaptically evoked APs (*F*) were compared between Cont *vs.* Hemi and Cont *vs.* Null by bootstrap resampling. The thick vertical lines on the grey curves indicate the 97.5% confidence interval. *G–I*, APs evoked in VL neurons by somatic current injection in Cont (*G*), Hemi (*H*) and Null (*I*). Scale bar = 100 ms. Insets: the first two APs are magnified along the time scale. Scale bars = 20 mV and 1 ms. *J–L*, the AP threshold (*J*), amplitude (*K*) and half-width (*L*) were compared between Cont *vs.* Hemi and Cont *vs.* Null by bootstrap resampling. [Colour figure can be viewed at wileyonlinelibrary.com]

**Table 6. Comparison with commonly used statistical tests**

**The effect size CI clearly excluded zero**

| Comparison | Figure | Effect size* Observed [CI] | Statistical test and *p* value |
|---|---|---|---|
| Control *vs.* DTA, AMPA charge | Fig. 4*F* | 0.482 [0.0706, 0.765] | *P* = 0.012 (Wilcoxon rank sum test) |
| Control *vs.* DTA, NMDA amplitude | Fig. 4*G* | 0.494 [0.0882. 0.759] | *P* = 0.011 (Wilcoxon rank sum test) |
| Control *vs.* TSC null, NMDA amplitude | Fig. 6*G* | *0.911 [0.177, 1.58]* | *P* = 0.010 (one-way ANOVA) |
| | | | *P* = 0.817 for Cont *vs.* Hemi (Tukey's HSD) |
| | | | *P* = 0.007 for Cont *vs.* Null (Tukey's HSD) |
| | | | *P* = 0.097 for Hemi *vs.* Null (Tukey's HSD) |

**The effect size CI marginally excluded zero**

| Comparison | Figure | Effect size * Observed [CI] | Statistical test and *p* value |
|---|---|---|---|
| Control *vs.* DTA, AMPA amplitude | Fig. 4*E* | 0.412 [0.00588, 0.682] | *P* = 0.033 (Wilcoxon rank sum test) |
| Control *vs.* DTA, Max AP number | Fig. 5*E* | *0.739 [0.0156, 1.41]* | *P* = 0.037 (Welch's *t* test) |
| Control *vs.* TSC null, AMPA charge | Fig. 6*F* | *0.743 [0.032, 1.41]* | *P* = 0.010 (one-way ANOVA) |
| | | | *P* = 0.866 for Cont *vs.* Hemi (Tukey's HSD) |
| | | | *P* = 0.020 for Cont *vs.* Null (Tukey's HSD) |
| | | | *P* = 0.018 for Hemi *vs.* Null (Tukey's HSD) |
| Control *vs.* TSC Hemi, AP threshold | Fig. 7*J* | −0.389 [−0.684, −0.0017] | *P* = 0.027 (Kruskal–Wallis test) |
| | | | *P* = 0.030 for Cont *vs.* Hemi (Dunn's test) |
| | | | *P* = 0.862 for Cont *vs.* Null (Dunn's test) |
| | | | *P* = 0.030 for Hemi *vs.* Null (Dunn's test) |

**The effect size CI marginally included zero**

| Comparison | Figure | Effect size * Observed [CI] | Statistical test and *p* value |
|---|---|---|---|
| Control *vs.* DTA, DCN AP frequency (Juvenile) | Fig. 2*C* | −0.261 [−0.498, 0.0143] | *P* = 0.056 (Wilcoxon rank sum test) |
| Control *vs.* DTA, Resting potential | Fig. 5*C* | *−0.651 [−1.31, 0.0286]* | *P* = 0.057 (Welch's *t* test) |
| Control *vs.* TSC Null, AMPA amplitude | Fig. 6*E* | *0.618 [−0.0669, 1.28]* | *P* = 0.095 (one-way ANOVA) |
| Control *vs.* TSC Null, AMPA/NMDA ratio | Fig. 6*H* | *−0.64 [−1.26, 0.0627]* | *P* = 0.108 (one-way ANOVA) |
| Control *vs.* TSC Null, AP amplitude | Fig. 7*K* | −0.343 [−0.667, 0.0404] | *P* = 0.017 (Kruskal–Wallis test)^ |
| | | | *P* = 0.223 for Cont *vs.* Hemi (Dunn's test) |
| | | | *P* = 0.081 for Cont *vs.* Null (Dunn's test) |
| | | | *P* = 0.014 for Hemi *vs.* Null (Dunn's test) |

*The regular and italic effect sizes represent Cliff's delta and Cohen's *d*, respectively.
^ Our bootstrap resampling only compared the control *vs.* TSC Hemi and control *vs.* TSC Null, which were non-significant.
CI, confidence interval; HS, honestly significant difference.

synaptic inputs evoked more APs in the post-synaptic VL neurons only in the Purkinje cell ablation mice, at least at a statistically significant level. This result raises a question regarding how important the strength of cerebellothalamic synapses is in the cerebello-thalama-cortical circuit.

Previous studies showed that VL neurons evoke more APs when more hyperpolarized (Contreras & Steriade, 1995; Llinás & Steriade, 2006; Schäfer et al., 2021). On the other hand, how the strength of cerebellothalamic synapses affects postsynaptic AP firing is largely unknown. To address this point, we pooled

all 51 cells used to analyse the membrane properties and excitability of VL neurons in the control mice (Figs 5 and 7). Consistent with the previous studies, the maximum number of synaptically evoked APs in individual VL neurons negatively correlated with their resting membrane potential (Fig. 8*A*, Pearson's correlation coefficient, $r_{49} = -0.325$, $P = 0.020$).

Although these cells were recorded primarily under the current-clamp mode, voltage-clamp recordings were also performed to measure the size of the maximum AMPA-EPSCs. We excluded six cells that exhibited voltage-clamp failure. The remaining 45 control cells showed a significant positive correlation between the maximum number of synaptically evoked APs and the charge transfer of the maximum AMPA-EPSCs they received (Fig. 8*B*, $r_{43} = 0.421$, $P = 0.004$).

If VL neurons receiving stronger synaptic inputs were coincidentally more hyperpolarized in our data set, the maximum AP number indirectly correlates with the synaptic strength. To test this improbable scenario, we examined the correlation between the resting membrane potential and the charge transfer of the maximum AMPA-EPSCs. The correlation coefficient was near zero, indicating that these two values were uncorrelated (Fig. 8*C*, $r_{43} = 0.026$, $P = 0.865$). Thus, the number of postsynaptic APs directly correlates with the synaptic strength, suggesting that stronger cerebellothalamic inputs can evoke more APs in VL neurons.

Still, the synaptic strength is just one factor affecting thalamic AP firing. Although the charge transfers of the maximum AMPA-EPSCs were similar between the DTA mice ($27.77 \pm 12.72$ pC, $n = 20$ cells) and the *Tsc1* homozygous deletion mice ($27.09 \pm 10.09$ pC, $n = 19$ cells), the enhancement of synaptically-evoked thalamic APs was significant only in the DTA mice. This might suggest that

the thalamic resting potential is slightly hyperpolarized in the DTA mice (Fig. 5*C*).

## Comparison with more routinely used statistical tests

We used bootstrap resampling throughout the analysis to compute the observed difference (effect size) and estimate its distribution. Although this approach allows us to infer the certainty and magnitude of the effect, it is not a routine practice in similar studies. Therefore, we performed more routinely used null-hypothesis testing for the following comparisons: the confidence interval clearly excluded zero, marginally excluded zero, and marginally included zero (Table 6). In all these comparisons, whether the *P* value of the null hypothesis testing is below or above 0.05 is consistent with whether the confidence interval excluded or included zero. Thus, our approach is comparable to commonly used approaches regarding statistical significance testing.

## Discussion

In the present study, we examined how cerebellar perturbation during development affects the formation of cerebellothalamic circuits in the VL. We found that a relatively small number of cerebellothalamic axons provide a powerful excitatory drive to individual VL neurons. This synaptic organization is similar to sensory inputs innervating the primary sensory thalamus, which is known to be refined by neuronal activity during development. Furthermore, Purkinje cell ablation and Purkinje cell-specific homozygous deletion of *Tsc1* strengthened cerebellothalamic synapses, suggesting that DCN activity affects the formation of cerebellothalamic circuits. Because the organization of the cerebello-thalamo-cortical circuit is similar between

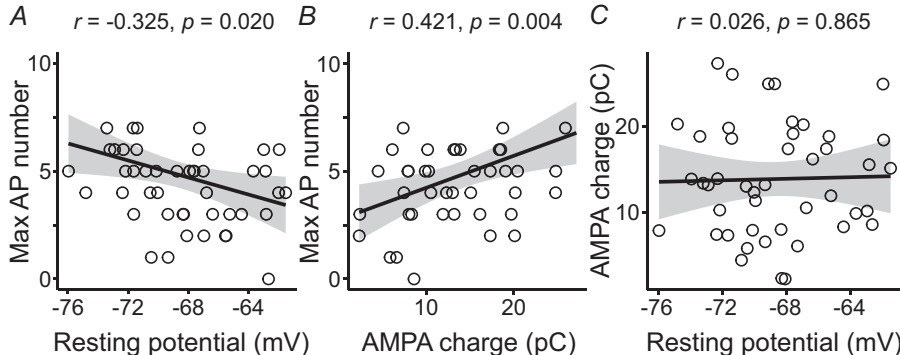

**Figure 8. Factors affecting synaptic outputs of VL neurons**
*A*, correlation between individual neurons' resting membrane potential and the maximum number of synaptically evoked APs. *B*, correlation between individual neurons' synaptic charge transfer by the maximum AMPA-EPSCs and the maximum number of synaptically evoked APs. *C*, correlation between individual neurons' resting membrane potential and synaptic charge transfer by the maximum AMPA-EPSCs. In all plots, the solid lines indicate linear regression lines, and the grey shades indicate their 95% confidence intervals.

motor and non-motor systems, what we learn from the VL may also be applicable to non-motor systems.

## Effects of Purkinje cell ablation on the intrinsic excitability of DCN neurons

More than 90% of Purkinje cells in the DTA mice degenerate by 2 weeks old. To study the consequence of this massive Purkinje cell loss on the intrinsic excitability of DCN neurons, we performed *ex vivo* recordings of the DCN.

Previous studies showed that DCN does not require Purkinje cells to maintain spontaneous APs (Aizenman & Linden, 1999; Beekhof et al., 2021; Huang & Uusisaari, 2013; Raman et al., 2000; Uusisaari & Knöpfel, 2010). However, the DCN neurons in these studies were disconnected from Purkinje cells for only several hours by tissue dissection and/or blocking inhibitory synaptic transmission. How chronic Purkinje cell loss for days and weeks affects DCN intrinsic excitability remains unknown, which is a crucial point to address because prolonged excitation may substantially alter DCN neurons. Our data showed that this is not the case. Although the firing rate of spontaneous APs may be slightly lower in the DTA mice, the difference was not significant. Furthermore, DCN neurons in the DTA mice increased their firing rate normally in response to depolarization.

These results do not indicate whether the overall DCN activity is higher or lower in intact DTA mice. However, considering the lack of Purkinje cells, we assume that the DCN in the DTA mice is more sensitive to excitatory synaptic inputs and shows an abnormal activity pattern.

## The number of cerebellothalamic inputs per VL neuron

Our data showed that individual VL neurons receive one to six cerebellothalamic axons, and approximately three on average. This number is probably smaller than the actual number of inputs because not all the deep cerebellar nuclei neurons expressed ChR2. However, electrical stimulation is not a better alternative because it is non-selective, and a stimulation electrode must be placed 0.5–1 mm away from the recording site to avoid stimulating corticothalamic axons (Aumann et al., 2000). In this configuration, axons that run at an oblique angle to the tissue surface are prone to be severed by slicing and difficult to excite.

Previous studies used electrical stimulation to quantify the number of sensory driver inputs innervating individual neurons in the primary visual and somatosensory thalamus. These studies reported one to three sensory driver inputs, mostly one input

per neuron (Arsenault & Zhang, 2006; Chen & Regehr, 2000; Hooks & Chen, 2006; Takeuchi et al., 2014; Wang & Zhang, 2008). However, a recent study showed that optogenetically evoked EPSCs are significantly larger than electrically evoked EPSCs in the primary visual thalamus, indicating that electrical stimulation indeed fails to excite some inputs. It is currently estimated that ∼10 retinal ganglion cell axons innervate a single neuron in the primary visual thalamus, among which around three inputs are strong enough to drive postsynaptic AP firing (Litvina & Chen, 2017).

Although the number we reported herein was fewer than the above estimate, the overall synaptic structure in the VL (i.e. a few driver inputs innervating individual thalamic neurons) is similar to the primary sensory thalamus. This synaptic structure is achieved by eliminating redundant inputs in the primary sensory thalamus, but whether the same thing occurs in the VL remains unknown. Because AAV needs 2–3 weeks until the gene expression reaches a sufficient level, it is unsuitable for performing optogenetics in neonatal animals. However, the main goal of the present study was not to determine whether VL neurons initially receive more inputs from the cerebellum. Instead, we aimed to reveal the consequences of early cerebellar perturbation on the formation of thalamic circuits. Our data showed that, regardless of whether synapse elimination occurs, neither *Tsc1* deletion nor Purkinje cell ablation substantially changes the number of cerebellar inputs per neuron in mature VL.

## Purkinje cell ablation and Purkinje-cell specific *Tsc1* deletion

Most electrophysiological recordings in our data set showed substantial cell-to-cell variation, indicating the heterogeneity of cellular and synaptic properties. Although such variation is unavoidable, it makes statistical inference challenging. Therefore, we sought to evaluate the difference between the groups not only by the statistical significance but also by the magnitude and certainty of the difference. To this end, we estimated the effect size and its confidence interval using bootstrap resampling.

Reflecting large cell-to-cell variations, effect sizes in most comparisons were negligible to modest compared to their distribution ranges. The few exceptions (i.e. effect size distributions clearly excluded zero) were all related to synaptic strength measurements in the DTA mice and the homozygous *Tsc1* deletion mice (Table 6). Besides, the charge transfer of AMPA currents showed a statistically significant increase in both mice. These results led to the main finding of this study: Purkinje cell ablation and Purkinje cell-specific deletion

of *Tsc1* strengthen cerebellothalamic synapses in the motor thalamus. A previous study using juvenile rat brain slices showed that cerebellothalamic synapses express long-term potentiation (Aumann et al., 2000). Although we cannot directly compare our findings with the long-term potentiation induction *in vitro*, these results suggest that neuronal activity affects the strength of cerebellothalamic synapses.

Because a similar number of cerebellothalamic fibres innervate individual VL neurons across genotypes, it could be argued that the larger EPSCs observed in the DTA mice and homozygous *Tsc1* deletion mice are a result of the strengthening of individual cerebellothalamic fibres. However, a single fibre input can be reliably measured only for the first EPSC evoked by the minimum stimulus intensity. This reflects the response of the most light-sensitive fibre, which varies based on its location in the tissue or ChR2 expression level. Its amplitude ranges dramatically, from under 100 pA to several nA, making comparisons of single fibre inputs across genotypes challenging. Therefore, we clarify that this study demonstrates the strengthening of total synaptic input evoked by the maximum light intensity.

Cerebellothalamic synapses in non-motor thalamic nuclei may also be strengthened in the DTA mice and homozygous *Tsc1* deletion mice. Even so, such strengthening may not be the sole means by which social and cognitive deficits arise in *Tsc1* deletion mice because the synaptic strengthening was not observed in the hemizygous *Tsc1* deletion mice that are known to exhibit autism-like behaviours.

The stimulation of cerebellothalamic axons evoked more thalamic APs in the DTA mice, possibly because of the stronger synaptic input and potentially hyperpolarized thalamic resting potentials. Because the 95% confidence intervals marginally excluded and included zero for the AP numbers and the resting potentials, these results are somewhat uncertain regardless of the statistical significance. However, the effect size distributions for the AP numbers and resting potentials were also largely biased toward more APs and lower resting potentials in the hemizygous and homozygous *Tsc1* deletion mice. Although those effects were not statistically significant in the *Tsc1* deletion mice, the similarity between the DTA mice and *Tsc1* deletion mice may not be just a coincidence. It is possible to speculate that both Purkinje cell ablation and *Tsc1* deletion may alter thalamic resting potentials and synaptically-evoked thalamic APs, but to a lesser extent by *Tsc1* deletion. One caveat is that the regulation of thalamic AP firing in the VL is influenced by thalamic states, corticothalamic inputs, and most probably by the pattern of DCN activities *in vivo*. How Purkinje cell ablation and *Tsc1* deletion affect thalamic AP firing *in vivo* is yet to be determined.

## Potential effects of Purkinje cell ablation and *Tsc1* deletion on DCN activity *in vivo*

Because Purkinje cells are not directly connected to the thalamus, Purkinje cell ablation and *Tsc1* deletion should affect the thalamus through the DCN. Our data showed that DCN neurons are relatively robust against massive loss of Purkinje cells. Considering that *Tsc1* deletion is substantially milder than Purkinje cell loss, it probably does not alter the DCN excitability more than Purkinje cell ablation potentially does. Therefore, we presume that the primary factor affecting DCN activity in these mutant mice is how loss of Purkinje cells (DTA mice) or lower firing rates of Purkinje cells (*Tsc1* deletion mice) alter DCN activity.

Because the DCN in these mutant mice receives drastically reduced inhibition from Purkinje cells, it is straightforward to assume that DCN activity in these mice is higher. However, the activity regulation in intact animals is complicated, and several factors should be considered. First, DCN neurons in normal animals increase their firing rate by temporal reduction of Purkinje cell firing rates (De Zeeuw et al., 2011; Heiney et al., 2014; Ito, 1984). This disinhibition-driven upregulation mechanism should increase DCN activity temporally in the mutant mice when the DCN neurons start to lose their presynaptic Purkinje cells, but its long-term effect is unclear. Second, DCN neurons receive excitatory inputs from mossy fibre and climbing fibre collaterals (Boele et al., 2013; Medina & Mauk, 1999; Najac & Raman, 2017; Pugh & Raman, 2006, 2009; Zhang & Linden, 2006). Although these inputs do not potently excite the DCN under the tonic inhibition from Purkinje cells, they might elicit burst-type activity in the mutant mice. In other words, mossy fibre and climbing fibre excitation in the DTA mice and *Tsc1* deletion mice may be transmitted with reduced countering inhibition through the DCN to the thalamus and other brain regions. Third, the cerebellum consists of parasagittally oriented functional modules, such that Purkinje cells in different modules project to different DCN subregions, regulating their activity in a region-specific manner (Apps & Hawkes, 2009; Kebschull et al., 2023). Assume that most Purkinje cells are lost or their activity is weakened. This region-specific regulation mechanism is largely lost or weakened, and the DCN activity is expected to become more widespread and disorganized. Thus, the consequences of Purkinje cell ablation and *Tsc1* deletion are probably not limited to changes in overall firing rates. Sensory and behaviourally-driven activities and the spatiotemporal pattern of DCN activities should also be affected. Future studies need to address these points to understand the activity-dependent mechanisms underlying the strengthening of cerebellothalamic synapses.

Another important point is the difference between the *Tsc1* hemizygous and homozygous deletion mice. Although Purkinje cells in both mutant mice showed reduced firing rates by 4–6 weeks old (i.e. before the onset of our thalamic recordings), only the homozygous mutant mice showed strengthened cerebellothalamic synapses. This may be because the homozygous mutant mice showed a more significant reduction in Purkinje cell firing rates than the hemizygous mutant mice (Tsai et al., 2012). However, it is also possible to assume that the strengthening of cerebellothalamic synapses is mediated by the loss of Purkinje cells, which only occurs in the homozygous mutant mice. Approximately 75% of Purkinje cells are lost in the homozygous mutant mice between 1.5 and 4 months old (Tsai et al., 2012) and this period largely overlaps with our thalamic recordings. The similarity between the DTA mice and the *Tsc1* homozygous deletion mice might suggest that loss of Purkinje cells strengthens cerebellothalamic synapses, even if it starts during adolescence.

### Comparison with other cerebellar damage models

Compared to the previous studies on the primary sensory thalamus, Purkinje cell ablation (DTA mice) and Purkinje cell hypoactivity (*Tsc1* homozygous deletion mice) had relatively minor effects on the motor thalamus. One caveat is that the activity of sensory driver inputs can be blocked by sensory deprivation, whereas the manipulation of DCN activity is more complicated. Nevertheless, it is interesting to study cerebellothalamic synapses in disease models associated with Purkinje cell hyperactivity, such as fetal alcohol syndrome and the gene mutation associated with amyotrophic lateral sclerosis and frontotemporal dementia (Liu et al., 2022; Servais et al., 2007).

In human patients with developmental cerebellar lesions, isolated cerebellar damage reduces the volume of the contralateral cerebral cortex (Limperopoulos et al., 2010, 2014; Stoodley & Limperopoulos, 2016). However, the *Tsc1* homozygous deletion mice showed no volumetric changes outside the cerebellum (Tsai et al., 2018). Magnetic resonance imaging revealed that cerebellothalamic axons seem somewhat damaged in some patients after pediatric tumor surgery (Law et al., 2011; Morris et al., 2009). We speculate that the reduced cortical volume in human patients was caused by reduced thalamocortical activity because of the damage in the DCN or degeneration of cerebellothalamic axons. When one side of cerebellothalamic axons is transected in neonatal animals, new collaterals sprout from the spared side and reinnervate the deafferented thalamus (Molinari et al., 1986). A similar circuit reorganization might occur in pediatric cerebellar surgery in human patients and contribute to the pathogenesis of motor and non-motor impairments.

In summary, Purkinje cell abnormality, one of the most common cerebellar pathology in neurological conditions, affects cerebellar inputs to the thalamus. Although its functional significance in neurological disorders is yet to be elucidated, abnormal cerebellothalamic circuits probably alter how the cerebellum regulates motor and non-motor behaviours throughout life.

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

## Additional information

### Data availability statement

The data supporting the results of this study are included within the figures in the published article and are available from the corresponding author upon reasonable request.

### Competing interests

The authors declare that they have no competing interests.

### Author contributions

H.N. designed the research. H.N. and N.N. performed the experiments. HN and NN analysed data. BVZ contributed unpublished reagents/analytical tools. HN and BVZ wrote the original draft. HN, NN and BVZ reviewed, edited and approved the final version of the manuscript.

## Funding

National Institutes of Health Grant R21NS108252 (HN) and 1U01 NS099720 and 1U01 NS094330 NINDS (BVZ).

## Acknowledgements

We thank Daniel Johnston, Richard Grey and Darrin Brager for their helpful support in conducting electrophysiology experiments, as well as Shigeyoshi Itohara for kindly providing us with the *Eno2$^{fsDTA/+}$* transgenic mice.

## Keywords

cerebellum, development, optogenetics, purkinje cells, thalamus, whole-cell recording

## Supporting information

Additional supporting information can be found online in the Supporting Information section at the end of the HTML view of the article. Supporting information files available:

**Peer Review History**

