## [Peer Review History · The Journal of Physiology]

Purkinje cell ablation and Purkinje cell-specific deletion of Tsc1 in the developing cerebellum strengthen cerebellothalamic synapses

Hiroshi Nishiyama, Naoko Nishiyama, and Boris V Zemelman

DOI: 10.1113/JP285887

Corresponding author(s): Hiroshi Nishiyama (hnishiyama@mail.clm.utexas.edu)

The following individual(s) involved in review of this submission have agreed to reveal their identity: Sean Williams (Referee #3)

Review Timeline:

Submission Date:	02-Nov-2023
Editorial Decision:	13-Dec-2023
Revision Received:	05-Sep-2024
Editorial Decision:	03-Oct-2024
Revision Received:	13-Oct-2024
Accepted:	22-Oct-2024

Senior Editor: Katalin Toth

Reviewing Editor: Nathan Schoppa

Transaction Report:

Dear Dr Nishiyama,

Re: JP-RP-2023-285887 "Loss of Purkinje cells in the developing cerebellum strengthens the cerebellothalamic synapses" by Hiroshi Nishiyama, Naoko Nishiyama, and Boris V Zemelman

Thank you for submitting your manuscript to The Journal of Physiology. It has been assessed by a Reviewing Editor and by 3 expert referees and we are pleased to tell you that it is potentially acceptable for publication following satisfactory major revision.

LANGUAGE EDITING AND SUPPORT FOR PUBLICATION: If you would like help with English language editing, or other article preparation support, Wiley Editing Services offers expert help, including English Language Editing, as well as translation, manuscript formatting, and figure formatting at www.wileyauthors.com/eoo/preparation. You can also find resources for Preparing Your Article for general guidance about writing and preparing your manuscript at www.wileyauthors.com/eoo/prepresources.

REVISION CHECKLIST:

We look forward to receiving your revised submission.

Yours sincerely,

Katalin Toth
Senior Editor
The Journal of Physiology

REQUIRED ITEMS

- Author photo and profile. First or joint first authors are asked to provide a short biography (no more than 100 words for one author or 150 words in total for joint first authors) and a portrait photograph. These should be uploaded and clearly labelled together in a Word document with the revised version of the manuscript. See Information for Authors for further details.
- You must start the Methods section with a paragraph headed Ethical Approval. A detailed explanation of journal policy and regulations on animal experimentation is given in Principles and standards for reporting animal experiments in The Journal of Physiology and Experimental Physiology by David Grundy J Physiol, 593: 2547-2549. doi:10.1113/JP270818). A checklist outlining these requirements and detailing the information that must be provided in the paper can be found at: <https://physoc.onlinelibrary.wiley.com/hub/animal-experiments>. Authors should confirm in their Methods section that their experiments were carried out according to the guidelines laid down by their institution's animal welfare committee, and conform to the principles and regulations as described in the Editorial by Grundy (2015), including an ethics approval reference number. The Methods section must contain a statement about access to food, water and housing, details of the anaesthetic regime: anaesthetic used, dose and route of administration, and method of killing the experimental animals.
- The Journal of Physiology funds authors of provisionally accepted papers to use the premium BioRender site to create high resolution schematic figures. Follow this link and enter your details and the manuscript number to create and download figures. Upload these as the figure files for your revised submission. If you choose not to take up this offer, we require figures to be of similar quality and resolution. If you are opting out of this service to authors, state this in the Comments section on the Detailed Information page of the submission form. The link provided should only be used for the purposes of this submission. Authors will be charged for figures created on this premium BioRender account if they are not related to this manuscript submission.
- Please upload separate high-quality figure files via the submission form.
- Please ensure that any tables are editable and in Word format, and wherever possible, embedded in the article file itself.
- Please ensure that the Article File you upload is a Word file.
- Papers must comply with the Statistics Policy: https://jp.msubmit.net/cgi-bin/main.plex?form_type=display_requirements#statistics.

In summary:

- If $n \leq 30$, all data points must be plotted in the figure in a way that reveals their range and distribution. A bar graph with data points overlaid, a box and whisker plot or a violin plot (preferably with data points included) are acceptable formats.
- If $n > 30$, then the entire raw dataset must be made available either as supporting information, or hosted on a not-for-profit repository, e.g. FigShare, with access details provided in the manuscript.
- 'n' clearly defined (e.g. x cells from y slices in z animals) in the Methods. Authors should be mindful of pseudoreplication.
- All relevant 'n' values must be clearly stated in the main text, figures and tables.
- The most appropriate summary statistic (e.g. mean or median and standard deviation) must be used. Standard Error of the Mean (SEM) alone is not permitted.

- Exact p values must be stated. Authors must not use 'greater than' or 'less than'. Exact p values must be stated to three significant figures even when 'no statistical significance' is claimed.

- Please include an Abstract Figure file, as well as the Figure Legend text within the main article file. The Abstract Figure is a piece of artwork designed to give readers an immediate understanding of the research and should summarise the main conclusions. If possible, the image should be easily 'readable' from left to right or top to bottom. It should show the physiological relevance of the manuscript so readers can assess the importance and content of its findings. Abstract Figures should not merely recapitulate other figures in the manuscript. Please try to keep the diagram as simple as possible and without superfluous information that may distract from the main conclusion(s). Abstract Figures must be provided by authors no later than the revised manuscript stage and should be uploaded as a separate file during online submission labelled as File Type 'Abstract Figure'. Please also ensure that you include the figure legend in the main article file. All Abstract Figures should be created using BioRender. Authors should use The Journal's premium BioRender account to export high-resolution images. Details on how to use and access the premium account are included as part of this email.

- Please include a full title page as part of your main article (Word) file, which should contain the following: title, authors, affiliations, corresponding author name and contact details, keywords, and running title.

- The corresponding author must provide an institutional email address (not a personal address) for their author account. We encourage ALL co-authors to also provide institutional email addresses. If this cannot be provided (as corresponding author), then a stamped letter must be provided from the institution which confirms their role and employment there (please upload this with the revised submission).

- Please ensure that all figures and tables have a title and legend, and that they have been cited within the main article text.

EDITOR COMMENTS

Reviewing Editor:

This study examines the effect of cerebellar Purkinje cell perturbations on the formation and function of the cerebello-thalamic circuit. The authors use two Purkinje cell manipulations, Purkinje cell-specific hemizygous deletion of tuberous sclerosis complex-1 (Tsc-1), which is a well-established model of autism spectrum disorder (ASD), and a more severe manipulation involving cell ablation using cre-specific diphtheria toxin (DTA) targeted to Purkinje cells. Effects on cerebellothalamic connections are evaluated using electrophysiological recordings in slices that contain the ventrolateral thalamus (VL) prepared from 1.5-3-month-old mice. Results are presented supporting that the more severe DTA ablation strengthens cerebellothalamic connections onto VL neurons and also increases the number of synaptically-evoked action potentials. These effects are however not observed in mice with the milder hemizygous Tsc-1 deletion. The manuscript has been reviewed by two expert reviewers, who both felt that the study provided novel and important mechanistic insights into how cerebellar dysfunction early in development can influence cortical circuits. One of the novel aspects of the study was to examine connections in the thalamus, through which signals originating in the cerebellum must pass prior to reaching cortex. Both reviewers however raised a number of concerns, which will need to be addressed. The most important of these included:

1. Reviewer 2 raises an important point around the choice of severe perturbation involving DTA-mediated ablation of Purkinje cells. As an alternative, the authors could have used homozygous Tsc-1 deletion mice. Studies in these latter mice would have been more useful for comparisons with prior literature that used both hemi- and homozygous Tsc-1 deletion as mouse models of ASD.

The study would be more complete if the authors were to add new data from homozygous Tsc-1 deletion mice, and, if the authors have these data, they should be included. The authors may also discuss their results in DTA ablation mice in the context of the points raised by Reviewer 2. Such a discussion should include more elaboration on their choice for a severe Purkinje cell perturbation. The authors may justify the use of DTA-mediated ablation as a model on the grounds that it enabled them to distinguish the effects on cerebellothalamic connectivity due to changes in Purkinje cell number versus changes in Purkinje spiking rate, since DTA likely only alters cell number. Homozygous Tsc-1 deletion in contrast causes large reductions in both Purkinje cell number and spike rate (Tsai et al., 2012).

2. As recommended by Reviewer 2 (see end of their Major Point 2), the authors should quantify the effect of the DTA manipulation on Purkinje cell number. This control will enable them to evaluate the effectiveness/severity of the DTA manipulation.

3. Both reviewers discuss developmental issues that may impact the interpretation of the circuit effects (or non-effects) observed for DTA ablation and hemizygous Tsc-1 deletion. The authors should address these concerns with discussion in the text. It would seem that some of the concerns could be addressed by the fact that the studies here were conducted in relatively old (1.5-3 month) mice.

4. As discussed by Reviewer 2, the authors assume that DTA ablation of Purkinje cells will result in an enhancement of

activity in cerebellar nuclear cells (due to disinhibition). However, it is possible that the effect of Purkinje cell loss will be so severe that there will be depolarization block of spiking. The authors should provide evidence supporting that their manipulation likely increased cerebellar nuclear cell activity.

5. The authors use bootstrapping methods to assess statistical significance, but there are more standard non-parametric statistical methods (e.g., Mann-Whitney U test) that could have been used. The authors need not reanalyze all of their results, but it would be useful to know whether more standard statistical methods result in significant effects for a few of the most important results. Another set of data that would be interesting to analyze using a more standard statistical test are those shown in Fig. 6E, where it appears that there is not a significant effect of hemizygous Tsc-1 deletion on action potential number in VL neurons (but see Comment from Reviewer 1, Major Point 1).

6. The authors should be clear in the statistics section of the Methods that standard deviations (rather than standard errors) are being reported in the text. That SDs are being used appears in the first instance where data are provided in the Results section, but having this information in the Methods is helpful for readers.

7. The authors need to add information about the animals' access to food and water.

REFeree COMMENTS

Referee #1:

In this study, Nishiyama et al. examine the synaptic basis by which cerebellar disruption may lead to distal functional changes. Their focus is the developmental disruption of Purkinje cell function, a condition that has been shown to lead to cognitive and social deficits by Tsai/Sahin in 2012. They also examine an extreme Purkinje cell ablation, done by expressing diphtheria toxin under the Pcp2 (Purkinje cell) promoter. They report that the diphtheria toxin condition leads to cerebellothalamic unitary synaptic strength of 1.5-fold the size of the unperturbed condition. The hemizygous Tsc1 knockout does not show this effect, suggesting that the effect is dose-dependent.

This is interesting and important work that extends our understanding of how cerebellar disruption can have distant effects. The experiments look well done. However, there are some deficiencies of analysis and interpretation that need to be corrected. In addition, it would be useful to have more experimental data describing exactly when the Tsc1 knockout starts to affect Purkinje cell firing.

My major and minor concerns are expressed as follows.

Major points

1. Figures 6E (hemizygous Tsc1 deletion in Purkinje cells) and 7E (diphtheria toxin killing of Purkinje cells) both show an increase in the number of APs evoked by stimulation. Therefore the two perturbations cause a shared functional change. However, this is not emphasized in the text.

This similarity should be mentioned in Results and incorporated in Discussion. Overall it seems the picture is (a) both perturbations increase the synaptically evoked AP output, and (b) the diphtheria toxin perturbation additionally increases unitary EPSC size. Together these findings would suggest increasing intensity of phenotype with earlier and larger knockout of Purkinje cells. They also show that the Pcp2-Tsc1 condition is sufficient to alter the functional properties of thalamus, which is an important and novel finding.

2. The age at which Purkinje cells begin to be disrupted is not totally defined. Pcp2 starts to be expressed around P7. Nonetheless, the two experimental conditions may affect Purkinje cell output at different ages. The Pcp2Cre/+ Eno2fsDTA/+ cross generates diphtheria toxin which would kill Purkinje cells within a day or two. However, correct me if I am wrong, but I think it is not known how long it would take hemizygous knockout of Tsc1 to affect Purkinje cell firing output. Therefore the difference in result for this case may be because of (a) smaller reductions in Purkinje cell firing or (b) delayed effects on Purkinje cell firing. The evidence presented does not resolve which is the case. This should be acknowledged.

3. A partial resolution could be achieved by measuring Purkinje cell firing frequency vs current step curves (i.e. an f-I curve) in the hemizygous knockout condition at different ages (for instance, P12, P18, P24) to see when it starts to change.

Minor points

4. For comparing the number of discrete EPSC steps, the bootstrap comparison shown in Figure 4C (and described on page 15 second full paragraph) seems overcomplicated. It would be good to also show something simpler: plot the cumulative distributions for the two conditions shown in Figure 4B, and then do a Kolmogorov-Smirnov test. This will give a negative result too.

5. Figure 7 shows differences (or lack of differences) in resting membrane potential and other intrinsic properties of VL neurons. These measurements are used to consider the question of whether any changes would be likely to change the

effective firing output of VL neurons. However, VL neurons in brain slices are in a different activity environment than in awake animals. Here it would be better to avoid reaching a conclusion about in vivo function, or to state that the difference pertains to VL neurons in a quiescent environment.

6. In Discussion paragraph 4 (page 20), a comparison is made between the authors' observed synaptic strength (1 nA) and sensory driver inputs to primary visual thalamus. This comparison may not necessarily be appropriate, since it compares cerebellothalamic and retinogeniculate synapses. This paragraph should be modified to clarify the type of comparison that is being made.

Referee #2 (please see attachment):

In the manuscript "Loss of Purkinje cells in the developing cerebellum strengthens the cerebellothalamic synapses" the authors examine the effect of Purkinje cell dysfunction/ablation on the formation and function of the cerebello-thalamic circuit. The authors use two manipulations of cerebellar Purkinje cells to address whether Purkinje cell dysfunction/loss impacts cerebellothalamic circuits. First they examine the effect of Purkinje cell specific hemizygous deletion of TSC1, which is a well-established model of autism spectrum disorder. As a second, more severe manipulation, they ablate Purkinje cells using a cre-specific DTA manipulation targeted to Purkinje cells. To examine cerebellothalamic function, they use elegant slice electrophysiological techniques to probe circuit function, including synaptic strength and intrinsic properties. Although little effects are found using following hemizygous deletion of TSC1, the authors demonstrate that loss of Purkinje cells via DTA ablation strengthens the cerebellothalamic circuit. The manuscript is exceedingly well written and clear and provides novel insight into how cerebellar dysfunction alters synaptic connectivity in the cerebellothalamic circuit, which is the primary (if not only) means by which the cerebellum influences cortical circuits. The results are a substantial step forward in understanding how cerebellar dysfunction, especially early in development, may alter activity in the cortex, which has been hypothesized as an important element in the pathophysiology and etiology of autism spectrum disorders. Despite these important advances, the enthusiasm for the manuscript is dampened by the use of such a severe manipulation of cerebellar activity as the ablation of all Purkinje cells. Because this manipulation is so severe, it is difficult to reconcile how more subtle dysregulation of activity in ASD models may contribute to cortical dysfunction. To remedy these concerns, the authors should examine dysregulation of the cerebellothalamic circuit in Purkinje-cell specific homozygous TSC1 knockout mice. Previous work has shown strong gene-dosage effects of Purkinje-cell specific TSC1 knockouts: hemizygous deletion results in rather subtle changes in Purkinje cell activity, whereas homozygous knockouts results in more robust decreases in Purkinje cell spiking as well as a moderate loss of Purkinje cell density, which matches the observed pathology in ASD human models (Tsai et al., 2012). In fact, the authors argue against using the TSC1 homozygous knockout specifically because of the loss of Purkinje cells, and then go on to completely ablate Purkinje cells, which is counterintuitive.

Major Concerns (summarized):

1) My primary concern with the manuscript is the use and justification of the DTA ablation of nearly all Purkinje cells in the cerebellar cortex. This manipulation is extreme and not well justified for studying how neuropsychiatric disorders such as ASD alter the cerebellothalamic circuit. As the authors correctly state, numerous neuropsychiatric disorders, including autism spectrum disorders, are associated with Purkinje cell loss and hypofunction (i.e. Tsai et al., 2012; Rogers et al., 2013; Courchesne et al., 1994 and others). However, the cell loss that is typically observed is exceedingly mild compared to the complete ablation of Purkinje cells. Therefore, the utility of examining complete Purkinje cell ablation as a model for understanding how ASD alters cerebellothalamic circuit development is minimal at best. This criticism is tempered by the fact that the authors make the argument that the primary goal of the paper is basic biology, namely examining how the cerebellum contributes to thalamic circuit function. Therefore, the use of the DTA ablation model does provide insight into the most extreme possible outcome. Recommended Action: To remedy these concerns, the authors should repeat similar experiments in the homozygous Pkj-cell specific TSC1 knockout, which shows both Purkinje cell hypofunction and moderate Purkinje cell loss, which more accurately recapitulates the pathology observed in ASD. The DTA ablation can then serve as an 'upper bound' of the deficits observed in mice with disrupted cerebellar function.

2) The authors use the pcp2-cre line to target Purkinje cells, and although this is the standard line used in the field, it does present some issues for studying synaptic development. The pcp2 gene is turned on at around PND6, however, full penetrance is not observed until 2-3 weeks of age (Barski et al., 2000). Therefore, ablation of Purkinje cells using the pcp-2 line may not be fully observed until 3-4 weeks of age. Recommended Action: To more accurately examine the timecourse of Purkinje cell loss, the authors should show a timecourse of Purkinje cell loss across development in the DTA-ablated animals. This point should also be more thoroughly addressed in the text/discussion. If the timecourse of Purkinje cell loss occurs after normal circuit development, then the effects observed may underestimate the effects of cerebellar dysfunction during critical periods. Figure 1 would also benefit from a more thorough quantification of cell density/morphology in all the genetic models used.

3) The authors make the assumption that Purkinje cell loss in the DTA-treated mice will increase cerebellar nuclear cell activity, as Purkinje cells provide tonic inhibition of the cerebellar nuclei. This assumption, however, must be validated to properly interpret their results. Because cerebellar nuclear cells sit above threshold (Raman et al., 2000), they may be particularly prone to depolarization block in the absence of any synaptic inhibition. Therefore, Purkinje cell ablation may increase nuclear cell activity, or may reduce it if the cells are in depolarization block due to sodium channel inactivation. Recommended Action: The authors should explicitly record from cerebellar nuclear cells in control and DTA treated animals

to assess spontaneous action potential firing in the DTA treated animals.

Minor Comments:

- 1) The title should either read "strengthens the cerebellothalamic synapse" or "strengthens cerebellothalamic synapses".
- 2) Figure 2 and 3 do not have any group data. The relative control data is replicated and further quantified in Figure 4/5 (i.e. paired pulse ratio and AMPA/NMDA ratio). Therefore, figure 2/3 should either be incorporated into figure 4/5 or perhaps figure 2 and 3 can be combined as a methodological figure. Alternatively, the authors can add group data for AMPA/NMDA ratio and the distribution of cells with different number of presynaptic partners to figures 2 and 3.
- 3) Clarify how NMDA measurements were made. Generally, NMDA current amplitude is measured at a latency of ~40 ms to prevent contamination by AMPA receptor-mediated current. It is not clear if that method was used here.
- 4) The choice of cut off for amplitudes greater than 1 nA seems quite large, despite being driver-type synapses. Can this be further explained or justified in the text (i.e. how many cells were excluded due to this criteria?)
- 5) Pg 12 paragraph 3 - The authors mention that cre-mediated recombination in the PCP2 line starts at PND6, although true, it does not complete until 3-4 weeks of age (Barski et al., 2000). This should be mentioned in the text.
- 6) Pg 12 last paragraph - "the degree of cell loss exceeds the conditions found in most neurological disorders" is a vast understatement and must be reworded. To my knowledge there are no common neurological disorders in which >90% of Purkinje cells are lost.
- 7) Pg 13 paragraph 1 - the authors should clarify in the results what cerebellar nucleus was targeted.
- 8) Figure 6/7 - scale bars and an absolute voltage measurement should be included in panels A, B, F, and G. Scale bars are added to the inset graphs, but not the main graphs. Additionally, for current clamp recordings, the graph must include an absolute voltage reference (i.e. dashed line at 0 mV) to assess actual membrane voltages.

Referee #3:

The bootstrapping method used in this manuscript is indeed a robust way to analyse these data. It avoids making assumptions about the underlying distributions and allows robust 95% CIs to be constructed. It's great to see the authors have included the 95% CIs throughout their results to enable to reader to view the uncertainty associated with each outcome.

The other analyses undertaken (mixed ANOVA and Pearsons r correlations) are also appropriate for the given research questions, though it would be important to confirm that the relevant assumptions for these tests have been checked and adjusted for, namely:

- That variables used for the Pearson correlation were normally distributed.
- The degree of asphericity associated with the paired factor (interstimulus interval?) used in the mixed ANOVA. If a substantial degree of asphericity was present (GG epsilon < .75), the Greenhouse-Geisser correction should be used to adjust the degrees of freedom and correct the F-ratio. Otherwise, the Huynh-Feldt correction should be used if the GG epsilon was > .75. Please see the link below for more information: <https://statistics.laerd.com/statistical-guides/sphericity-statistical-guide-2.php>.

END OF COMMENTS

Confidential Review

02-Nov-2023

Review: Loss of Purkinje cells in the developing cerebellum strengthens the cerebellothalamic synapses

Confidential Assessment:

The manuscript “Loss of Purkinje cells in the developing cerebellum strengthens the cerebellothalamic synapses” is an overall well-written manuscript that provides moderate value in understanding how cerebellar dysfunction impacts the formation of cerebellothalamic circuits. This is an important topic as emerging evidence suggests that early cerebellar dysfunction may significantly contribute to the etiology and pathophysiology of the cerebral cortex, especially during early sensitive periods. The enthusiasm and relevance of the manuscript, however, is significantly dampened by the curious choice to use a hemizygous TSC1 knockout model and a complete Purkinje cell ablation model. Previous evidence strongly indicates a gene dosage effect of TSC1 knockout, with the homozygous knockout resulting in Purkinje cell hypofunction and moderate Purkinje cell loss, which closely mimics observed pathophysiology in clinical cases. However, instead of using the TSC1 homozygous knockout mice, the authors instead ablate >90% of Purkinje cells using a cre dependent diphtheria toxin, using the argument that the TSC1 homozygous knockout results in a loss of Purkinje cells. The use of the TSC1 homozygous knockout animals is an essential revision – as it will provide valuable information about circuit dysfunction in a more clinically relevant disease model.

It is also worth noting that I am slightly uncomfortable with their statistical choices. I am no expert in statistics, and they use bootstrap resampling rather than traditional statistical tests to examine significance throughout the manuscript. The justification for this decision was unclear, and due to these decisions making a clear judgement on the statistics is difficult. I elected not to include this in the comments to the authors, as I am not an expert, but I did want to bring it to the editors attention.

Recommendation on Acceptability: Overall, the manuscript can be accepted with significant revisions - particularly the use of the homozygous TSC1 knockout animals. Without those essential revisions, the relevance and influence of the paper is severely diminished.

Comment on Influence: Ranking 2

As stated above, the influence of the manuscript as written is relatively low without the use of the homozygous TSC1 knockout. If those experiments were added, the influence would be significantly increased, as it would provide compelling evidence to demonstrate how cerebellar dysfunction impacts thalamic function, and by extension cerebral cortex.

Comments to the authors:

In the manuscript “Loss of Purkinje cells in the developing cerebellum strengthens the cerebellothalamic synapses” the authors examine the effect of Purkinje cell dysfunction/ablation on the formation and function of the cerebello-thalamic circuit. The authors use two manipulations of cerebellar Purkinje cells to address whether Purkinje cell dysfunction/loss impacts cerebellothalamic circuits. First they examine the effect of Purkinje cell specific hemizygous deletion of TSC1, which is a well-established model of autism spectrum disorder. As a second, more severe manipulation, they ablate Purkinje cells using a cre-specific DTA manipulation targeted to Purkinje cells. To examine cerebellothalamic function, they use elegant slice electrophysiological techniques to probe circuit function, including synaptic strength and intrinsic properties. Although little effects are found using following hemizygous deletion of TSC1, the authors demonstrate that loss of Purkinje cells via DTA ablation strengthens the cerebellothalamic circuit. The manuscript is exceedingly well written and clear and provides novel insight into how cerebellar dysfunction alters synaptic connectivity in the cerebellothalamic circuit, which is the primary (if not only) means by which the cerebellum influences cortical circuits. The results are a substantial step forward in understanding how cerebellar dysfunction, especially early in development, may alter activity in the cortex, which has been hypothesized as an important element in the pathophysiology and etiology of autism spectrum disorders. Despite these important advances, the enthusiasm for the manuscript is dampened by the use of such a severe manipulation of cerebellar activity as the ablation of all Purkinje cells. Because this manipulation is so severe, it is difficult to reconcile how more subtle dysregulation of activity in ASD models may contribute to cortical dysfunction. To remedy these concerns, the authors should examine dysregulation of the cerebellothalamic circuit in Purkinje-cell specific homozygous TSC1 knockout mice. Previous work has shown strong gene-dosage effects of Purkinje-cell specific TSC1 knockouts: hemizygous deletion results in rather subtle changes in Purkinje cell activity, whereas homozygous knockouts results in more robust decreases in Purkinje cell spiking as well as a moderate loss of Purkinje cell density, which matches the observed pathology in ASD human models (Tsai et al., 2012). In fact, the authors argue against using the TSC1 homozygous knockout specifically because of the loss of Purkinje cells, and then go on to completely ablate Purkinje cells, which is counterintuitive.

Major Concerns (summarized):

- 1) My primary concern with the manuscript is the use and justification of the DTA ablation of nearly all Purkinje cells in the cerebellar cortex. This manipulation is extreme and not well justified for studying how neuropsychiatric disorders such as ASD alter the cerebellothalamic circuit. As the authors correctly state, numerous neuropsychiatric disorders, including autism spectrum disorders, are associated with Purkinje cell loss and hypofunction (i.e. Tsai et al., 2012; Rogers et al., 2013; Courchesne et al., 1994 and others). However, the cell loss that is typically observed is exceedingly mild compared to the complete ablation of Purkinje cells. Therefore, the utility of examining complete Purkinje cell ablation as a model for understanding how ASD alters cerebellothalamic circuit development is minimal at best. This criticism is tempered by the fact that the authors make the argument that the primary goal of the paper is basic biology, namely examining how the cerebellum contributes to thalamic circuit function. Therefore, the use of the DTA ablation model does provide insight into the most extreme possible outcome.

Recommended Action: To remedy these concerns, the authors should repeat similar

experiments in the homozygous Pkj-cell specific TSC1 knockout, which shows both Purkinje cell hypofunction and moderate Purkinje cell loss, which more accurately recapitulates the pathology observed in ASD. The DTA ablation can then serve as an 'upper bound' of the deficits observed in mice with disrupted cerebellar function.

- 2) The authors use the pcp2-cre line to target Purkinje cells, and although this is the standard line used in the field, it does present some issues for studying synaptic development. The pcp2 gene is turned on at around PND6, however, full penetrance is not observed until 2-3 weeks of age (Barski et al., 2000). Therefore, ablation of Purkinje cells using the pcp-2 line may not be fully observed until 3-4 weeks of age. **Recommended Action:** To more accurately examine the timecourse of Purkinje cell loss, the authors should show a timecourse of Purkinje cell loss across development in the DTA-ablated animals. This point should also be more thoroughly addressed in the text/discussion. If the timecourse of Purkinje cell loss occurs after normal circuit development, then the effects observed may underestimate the effects of cerebellar dysfunction during critical periods. Figure 1 would also benefit from a more thorough quantification of cell density/morphology in all the genetic models used.
- 3) The authors make the assumption that Purkinje cell loss in the DTA-treated mice will increase cerebellar nuclear cell activity, as Purkinje cells provide tonic inhibition of the cerebellar nuclei. This assumption, however, must be validated to properly interpret their results. Because cerebellar nuclear cells sit above threshold (Raman et al., 2000), they may be particularly prone to depolarization block in the absence of any synaptic inhibition. Therefore, Purkinje cell ablation may increase nuclear cell activity, or may reduce it if the cells are in depolarization block due to sodium channel inactivation. **Recommended Action:** The authors should explicitly record from cerebellar nuclear cells in control and DTA treated animals to assess spontaneous action potential firing in the DTA treated animals.

Minor Comments:

- 1) The title should either read "strengthens the cerebellothalamic synapse" or "strengthens cerebellothalamic synapses"
- 2) Figure 2 and 3 do not have any group data. The relative control data is replicated and further quantified in Figure 4/5 (i.e. paired pulse ratio and AMPA/NMDA ratio). Therefore, figure 2/3 should either be incorporated into figure 4/5 or perhaps figure 2 and 3 can be combined as a methodological figure. Alternatively, the authors can add group data for AMPA/NMDA ratio and the distribution of cells with different number of presynaptic partners to figures 2 and 3.
- 3) Clarify how NMDA measurements were made. Generally, NMDA current amplitude is measured at a latency of ~40 ms to prevent contamination by AMPA receptor-mediated current. It is not clear if that method was used here.
- 4) The choice of cut off for amplitudes greater than 1 nA seems quite large, despite being driver-type synapses. Can this be further explained or justified in the text (i.e. how many cells were excluded due to this criteria?)

- 5) Pg 12 paragraph 3 – The authors mention that cre-mediated recombination in the PCP2 line starts at PND6, although true, it does not complete until 3-4 weeks of age (Barski et al., 2000). This should be mentioned in the text.
- 6) Pg 12 last paragraph – “the degree of cell loss exceeds the conditions found in most neurological disorders” is a vast understatement and must be reworded. To my knowledge there are no common neurological disorders in which >90% of Purkinje cells are lost.
- 7) Pg 13 paragraph 1 – the authors should clarify in the results what cerebellar nucleus was targeted.
- 8) Figure 6/7 – scale bars and an absolute voltage measurement should be included in panels A, B, F, and G. Scale bars are added to the inset graphs, but not the main graphs. Additionally, for current clamp recordings, the graph must include an absolute voltage reference (i.e. dashed line at 0 mV) to assess actual membrane voltages.

EDITOR COMMENTS

Reviewing Editor:

This study examines the effect of cerebellar Purkinje cell perturbations on the formation and function of the cerebello-thalamic circuit. The authors use two Purkinje cell manipulations, Purkinje cell-specific hemizygous deletion of tuberous sclerosis complex-1 (Tsc-1), which is a well-established model of autism spectrum disorder (ASD), and a more severe manipulation involving cell ablation using cre-specific diphtheria toxin (DTA) targeted to Purkinje cells. Effects on cerebellothalamic connections are evaluated using electrophysiological recordings in slices that contain the ventrolateral thalamus (VL) prepared from 1.5-3-month-old mice. Results are presented supporting that the more severe DTA ablation strengthens cerebellothalamic connections onto VL neurons and also increases the number of synaptically-evoked action potentials. These effects are however not observed in mice with the milder hemizygous Tsc-1 deletion. The manuscript has been reviewed by two expert reviewers, who both felt that the study provided novel and important mechanistic insights into how cerebellar dysfunction early in development can influence cortical circuits. One of the novel aspects of the study was to examine connections in the thalamus, through which signals originating in the cerebellum must pass prior to reaching cortex. Both reviewers however raised a number of concerns, which will need to be addressed. The most important of these included:

We are thankful for the encouraging reviews and constructive feedback. As detailed below, we fully addressed the editor's and referees' comments and followed the suggestions as much as possible. We believe that this revision represents a substantial improvement from the original manuscript. Our responses are shown using blue font color.

1. Reviewer 2 raises an important point around the choice of severe perturbation involving DTA-mediated ablation of Purkinje cells. As an alternative, the authors could have used homozygous Tsc-1 deletion mice. Studies in these latter mice would have been more useful for comparisons with prior literature that used both hemi- and homozygous Tsc-1 deletion as mouse models of ASD.

The study would be more complete if the authors were to add new data from homozygous Tsc-1 deletion mice, and, if the authors have these data, they should be included. The authors may also discuss their results in DTA ablation mice in the context of the points raised by Reviewer 2. Such a discussion should include more elaboration on their choice for a severe Purkinje cell perturbation. The authors may justify the use of DTA-mediated ablation as a model on the grounds that it enabled them to distinguish the effects on cerebellothalamic connectivity due to changes in Purkinje cell number versus changes in Purkinje spiking rate, since DTA likely only alters cell number. Homozygous Tsc-1 deletion in contrast causes large reductions in both Purkinje cell number and spike rate (Tsai et al., 2012).

We performed the same thalamic recordings using the homozygous Tsc-1 deletion mice (Fig. 6 and 7). The homozygous Tsc-1 deletion mice showed similar phenotypes to the Purkinje cell ablation mice (DTA mice) but to a lesser extent. We compared the new results with the DTA mice and the hemizygous Tsc-1 deletion mice and discussed what we can infer from those

comparisons. We believe that keeping the DTA mice data is beneficial because the similarity between the DTA mice and the homozygous Tsc-1 deletion mice strengthens the main conclusion of this study.

2. As recommended by Reviewer 2 (see end of their Major Point 2), the authors should quantify the effect of the DTA manipulation on Purkinje cell number. This control will enable them to evaluate the effectiveness/severity of the DTA manipulation.

We quantified the time course of Purkinje cell degeneration in the DTA mice (Fig. 1). Consistent with Barski et al. (2000), near-complete Purkinje cell loss was achieved at 3-4 weeks of age. However, more than 90% of Purkinje cells had degenerated by the end of the second postnatal week, and the change afterward was modest. Since cerebellar cortical circuits are established by the end of the third postnatal week in mice, the DTA mice lose more than 90% of Purkinje cells during the developmental phase of the cerebellar cortex. We described these points in the Results section (page 19, lines 472-476).

3. Both reviewers discuss developmental issues that may impact the interpretation of the circuit effects (or non-effects) observed for DTA ablation and hemizygous Tsc-1 deletion. The authors should address these concerns with discussion in the text. It would seem that some of the concerns could be addressed by the fact that the studies here were conducted in relatively old (1.5-3 month) mice.

The new data showed that (1) Cre-mediated recombination occurs in more than 90% of Purkinje cells by the age of two weeks (Fig. 1), and (2) strengthening of cerebellothalamic synapses also occurs in the homozygous Tsc-1 deletion mice (Fig. 6). We believe these new results addressed some of the concerns. Furthermore, we discussed the possibility of a relatively late developmental effect, suggested by the differences between the hemizygous vs. homozygous Tsc-1 deletion mice in the Discussion section (pages 35-36, lines 905-916).

4. As discussed by Reviewer 2, the authors assume that DTA ablation of Purkinje cells will result in an enhancement of activity in cerebellar nuclear cells (due to disinhibition). However, it is possible that the effect of Purkinje cell loss will be so severe that there will be depolarization block of spiking. The authors should provide evidence supporting that their manipulation likely increased cerebellar nuclear cell activity.

We agree with the editor and referee that we need to show how Purkinje cell loss affects the excitability of DCN neurons, which is currently unknown. Therefore, we performed patch-clamp recordings from DCN neurons in acutely prepared cerebellar slices (Fig. 2). The main findings are that (1) DCN neurons in the DTA mice maintain their ability to spontaneously discharge APs even several months after losing their presynaptic Purkinje cells, and (2) they respond to depolarizing stimuli similarly to the control mice.

However, these results do not indicate whether the overall DCN activity in vivo is higher or lower in the DTA mice. We realized that the potential effects of Purkinje cell loss on DCN activity and their contributions to synaptic strengthening are more complicated than we previously assumed. As detailed in the Discussion section (pages 34-35, lines 876-904), we need to consider not

only spontaneous activity but also sensory and behaviorally-driven DCN activities, including their spatiotemporal pattern, to understand how cerebellothalamic synapses are strengthened in the DTA mice and homozygous Tsc-1 deletion mice. Besides, it is currently unclear whether the strengthening results from chronic changes in DCN activity for weeks and months or whether temporal changes that occur during Purkinje cell degeneration are sufficient to induce the strengthening. Addressing these points requires in vivo recordings of DCN neurons from awake animals at multiple ages, including pre-weaned mice. These experiments are substantial work and technically challenging, and we believe they deserve to be reported in a future study.

In this revision, we withdrew the previous argument that Purkinje cell loss increases the DCN activity. Instead, we thoroughly described the potential effects of Purkinje cell loss on DCN activity in the Discussion section (pages 34-35, lines 876-904).

The main finding of this study, the strengthening of cerebellothalamic synapses by Purkinje cell ablation and Purkinje cell-specific homozygous deletion of Tsc-1, still provides novel insight into the roles of the cerebellum in forebrain development. We can reasonably wait for subsequent studies to investigate underlying activity-dependent mechanisms.

5. The authors use bootstrapping methods to assess statistical significance, but there are more standard non-parametric statistical methods (e.g., Mann-Whitney U test) that could have been used. The authors need not reanalyze all of their results, but it would be useful to know whether more standard statistical methods result in significant effects for a few of the most important results. Another set of data that would be interesting to analyze using a more standard statistical test are those shown in Fig. 6E, where it appears that there is not a significant effect of hemizygous Tsc-1 deletion on action potential number in VL neurons (but see Comment from Reviewer 1, Major Point 1).

We performed more commonly used statistical tests for the following comparisons. The confidence intervals of the estimated effect sizes (1) clearly excluded zero (significant difference), (2) marginally excluded zero (significant difference), and (3) marginally included zero (non-significant difference). In other words, these comparisons include all the statistically significant cases and those that slightly missed the significance. The results are summarized in a new table (Table 6), showing the consistency between the bootstrapping methods and the commonly used methods regarding the statistical significance.

6. The authors should be clear in the statistics section of the Methods that standard deviations (rather than standard errors) are being reported in the text. That SDs are being used appears in the first instance where data are provided in the Results section, but having this information in the Methods is helpful for readers.

At the beginning of the Statistical Analysis section, we now state that all measured values are reported as the mean \pm SD.

7. The authors need to add information about the animals' access to food and water.

The information is added to the "Ethical approval" section in the Methods section.

REFEREE COMMENTS

Referee #1:

In this study, Nishiyama et al. examine the synaptic basis by which cerebellar disruption may lead to distal functional changes. Their focus is the developmental disruption of Purkinje cell function, a condition that has been shown to lead to cognitive and social deficits by Tsai/Sahin in 2012. They also examine an extreme Purkinje cell ablation, done by expressing diphtheria toxin under the *Pcp2* (Purkinje cell) promoter. They report that the diphtheria toxin condition leads to cerebellothalamic unitary synaptic strength of 1.5-fold the size of the unperturbed condition. The hemizygous *Tsc1* knockout does not show this effect, suggesting that the effect is dose-dependent.

This is interesting and important work that extends our understanding of how cerebellar disruption can have distant effects. The experiments look well done. However, there are some deficiencies of analysis and interpretation that need to be corrected. In addition, it would be useful to have more experimental data describing exactly when the *Tsc1* knockout starts to affect Purkinje cell firing.

My major and minor concerns are expressed as follows.

We are thankful for the supportive review and constructive suggestions. As explained below, we addressed the referee's major points by adding new data and improving the analysis.

Major points

1. Figures 6E (hemizygous *Tsc1* deletion in Purkinje cells) and 7E (diphtheria toxin killing of Purkinje cells) both show an increase in the number of APs evoked by stimulation. Therefore the two perturbations cause a shared functional change. However, this is not emphasized in the text.

This similarity should be mentioned in Results and incorporated in discussion. Overall it seems the picture is (a) both perturbations increase the synaptically evoked AP output, and (b) the diphtheria toxin perturbation additionally increases unitary EPSC size. Together these findings would suggest increasing intensity of phenotype with earlier and larger knockout of Purkinje cells. They also show that the *Pcp2-Tsc1* condition is sufficient to alter the functional properties of thalamus, which is an important and novel finding.

We addressed this crucial point by (1) improving statistical analysis and (2) adding new data obtained from the homozygous *Tsc1* deletion mice.

We compared the mean difference between the control vs. mutant mice in the original manuscript. However, the mean difference does not inform us about the magnitude of the

difference, i.e., how substantially the distribution differs between the groups. Therefore, the meaning of the difference is difficult to infer when the statistical significance is missed. To overcome this problem, we computed effect sizes, the measure of magnitude. For details, please see the “Statistical analysis” section of the Methods section.

The newly added data showed that cerebellothalamic synapses are significantly strengthened in the homozygous *Tsc1* deletion mice (Fig. 6E-G). Notably, the effect size of synaptically-evoked thalamic AP number was substantially biased toward more APs in the homozygous *Tsc1* deletion mice (Fig. 7F). Although it slightly missed the statistical significance, we cannot certainly rule out the possibility that more APs are evoked in the homozygous *Tsc1* deletion mice. In short, Purkinje cell ablation and homozygous *Tsc1* deletion showed similar effects in synaptic strength, thalamic resting potentials, and synaptically-evoked AP number. We described this important consistency in the Results section (page 29, lines 731-734) and Discussion section (pages 33-34, lines 847-871).

2. The age at which Purkinje cells begin to be disrupted is not totally defined. *Pcp2* starts to be expressed around P7. Nonetheless, the two experimental conditions may affect Purkinje cell output at different ages. The *Pcp2Cre/+ Eno2fsDTA/+* cross generates diphtheria toxin which would kill Purkinje cells within a day or two. However, correct me if I am wrong, but I think it is not known how long it would take hemizygous knockout of *Tsc1* to affect Purkinje cell firing output. Therefore the difference in result for this case may be because of (a) smaller reductions in Purkinje cell firing or (b) delayed effects on Purkinje cell firing. The evidence presented does not resolve which is the case. This should be acknowledged.

We quantified the timing and extent of Purkinje cell loss in the DTA mice (Fig .1). Although the data show the time course of Cre-mediated recombination in our mutant mice, DTA expression and *Tsc1* deletion may start to affect Purkinje cells at different timing, as the reviewer pointed out.

Tsai et al. (2012) examined the electrophysiological properties of *Tsc1* deletion mice using 4-6-week-old animals; hence, it remains unclear whether Purkinje cells start to reduce their firing rate earlier. However, the homozygous deletion of *Tsc1* strengthened cerebellothalamic synapses like Purkinje cell ablation, indicating that the onset of the Purkinje cell abnormality in the *Tsc1* deletion mice is not too late to affect cerebellothalamic synapses.

Still, it remains unclear why hemizygous deletion of *Tsc1* does not strengthen cerebellothalamic synapses. It may be because of more subtle reductions in Purkinje cell firing rates or the lack of cell death in the hemizygous mutant. We acknowledged this uncertainty in the Discussion section (pages 35-36, lines 905-916).

3. A partial resolution could be achieved by measuring Purkinje cell firing frequency vs current step curves (i.e. an f-I curve) in the hemizygous knockout condition at different ages (for instance, P12, P18, P24) to see when it starts to change.

The referee suggested this experiment because a primary factor differentiating between the DTA mice and the hemizygous *Tsc1* deletion mice may be the age at which Purkinje cells start

degenerating (DTA mice) and reducing their firing rates (Tsc1 deletion mice). However, comparing the DTA mice and the hemizygous Tsc1 deletion mice is tricky, although we only had these two mutant mice in the original manuscript.

Now, we added the homozygous Tsc1 deletion mice, allowing more appropriate comparisons between (1) the DTA mice vs. the homozygous Tsc1 deletion mice and (2) the homozygous Tsc1 deletion mice vs. the hemizygous Tsc1 deletion mice. As described in our response to the referee's major comment #2, these comparisons suggest that age is not a primary factor contributing to no detectable effects of hemizygous Tsc1 deletion. Therefore, we did not perform the suggested recordings. Besides, even if the age at which Purkinje cells start reducing their firing rates differs between the homozygous vs. hemizygous Tsc1 deletion mice, it is still difficult to argue that age is a primary factor.

Perhaps the primary factors differentiating between the homozygous vs. hemizygous Tsc1 deletion mice are (1) the degree of reduction in Purkinje cell firing rates and (2) Purkinje cell loss. Addressing these points requires experimentally manipulating Purkinje cell firing rates and cell death, for which we can reasonably wait for future studies.

Minor points

4. For comparing the number of discrete EPSC steps, the bootstrap comparison shown in Figure 4C (and described on page 15 second full paragraph) seems overcomplicated. It would be good to also show something simpler: plot the cumulative distributions for the two conditions shown in Figure 4B, and then do a Kolmogorov-Smirnov test. This will give a negative result too.

Figure 4C in the original manuscript was the first data set with which we explained the bootstrap comparisons. Since it clearly showed no difference between the control and the hemizygous Tsc1 deletion mice (the mean of each group was the same), our analysis might appear overcomplicated. However, we would like to keep the analysis not only for the specific data set but also for the entire analysis in this study. We now explain the reason in the Discussion section (page 33, lines 841-846). Below is the excerpt.

"Most electrophysiological recordings in our data set showed substantial cell-to-cell variation, indicating the heterogeneity of cellular and synaptic properties. Although such variation is unavoidable, it makes statistical inference challenging. Therefore, we sought to evaluate the difference between the groups not only by the statistical significance but also by the magnitude and certainty of the difference. To this end, we estimated the effect size and its confidence interval using bootstrap resampling."

Since this approach is not often used in similar studies, we provided sufficient details in the Methods section. We also thoroughly explained how to interpret the data when they first appeared in the paper (Fig. 2C and D). Furthermore, we compared our method with commonly used statistical tests, which showed complete consistency regarding the statistical significance testing (Table 6).

5. Figure 7 shows differences (or lack of differences) in resting membrane potential and other

intrinsic properties of VL neurons. These measurements are used to consider the question of whether any changes would be likely to change the effective firing output of VL neurons. However, VL neurons in brain slices are in a different activity environment than in awake animals. Here it would be better to avoid reaching a conclusion about in vivo function, or to state that the difference pertains to VL neurons in a quiescent environment.

We agree with the referee and added the following statement in the Discussion section (page 34, lines 871-874)

“One caveat is that the regulation of thalamic AP firing in the VL is influenced by thalamic states, corticothalamic inputs, and most likely by the pattern of DCN activities in vivo. How Purkinje cell ablation and Tsc1 deletion affect thalamic AP firing in vivo is yet to be determined.”

6. In Discussion paragraph 4 (page 20), a comparison is made between the authors' observed synaptic strength (1 nA) and sensory driver inputs to primary visual thalamus. This comparison may not necessarily be appropriate, since it compares cerebellothalamic and retinogeniculate synapses. This paragraph should be modified to clarify the type of comparison that is being made.

We agree with the referee that the argument in the original manuscript was an excessive comparison of two different circuits. In this revision, we just pointed out the overall similarity between the primary sensory thalamus and cerebellothalamic synapses (pages 32-33, lines 828-838).

Referee #2 (please see attachment):

In the manuscript “Loss of Purkinje cells in the developing cerebellum strengthens the cerebellothalamic synapses” the authors examine the effect of Purkinje cell dysfunction/ablation on the formation and function of the cerebello-thalamic circuit. The authors use two manipulations of cerebellar Purkinje cells to address whether Purkinje cell dysfunction/loss impacts cerebellothalamic circuits. First they examine the effect of Purkinje cell specific hemizygous deletion of TSC1, which is a well-established model of autism spectrum disorder. As a second, more severe manipulation, they ablate Purkinje cells using a cre-specific DTA manipulation targeted to Purkinje cells. To examine cerebellothalamic function, they use elegant slice electrophysiological techniques to probe circuit function, including synaptic strength and intrinsic properties. Although little effects are found using following hemizygous deletion of TSC1, the authors demonstrate that loss of Purkinje cells via DTA ablation strengthens the cerebellothalamic circuit. The manuscript is exceedingly well written and clear and provides novel insight into how cerebellar dysfunction alters synaptic connectivity in the cerebellothalamic circuit, which is the primary (if not only) means by which the cerebellum influences cortical circuits. The results are a substantial step forward in understanding how cerebellar dysfunction, especially early in development, may alter activity in the cortex, which has been hypothesized as an important element in the pathophysiology and etiology of autism spectrum disorders. Despite these important advances, the enthusiasm for the manuscript is dampened by the use

of such a severe manipulation of cerebellar activity as the ablation of all Purkinje cells. Because this manipulation is so severe, it is difficult to reconcile how more subtle dysregulation of activity in ASD models may contribute to cortical dysfunction. To remedy these concerns, the authors should examine dysregulation of the cerebellothalamic circuit in Purkinje-cell specific homozygous TSC1 knockout mice. Previous work has shown strong gene-dosage effects of Purkinje-cell specific TSC1 knockouts: hemizygous deletion results in rather subtle changes in Purkinje cell activity, whereas homozygous knockouts results in more robust decreases in Purkinje cell spiking as well as a moderate loss of Purkinje cell density, which matches the observed pathology in ASD human models (Tsai et al., 2012). In fact, the authors argue against using the TSC1 homozygous knockout specifically because of the loss of Purkinje cells, and then go on to completely ablate Purkinje cells, which is counterintuitive.

We are thankful for the positive, encouraging review and crucial constructive feedback. We sincerely tried to address the referee's concerns and recommended action. Since all major concerns are summarized in the editor's comments above, please see our responses there.

Major Concerns (summarized):

1) My primary concern with the manuscript is the use and justification of the DTA ablation of nearly all Purkinje cells in the cerebellar cortex. This manipulation is extreme and not well justified for studying how neuropsychiatric disorders such as ASD alter the cerebellothalamic circuit. As the authors correctly state, numerous neuropsychiatric disorders, including autism spectrum disorders, are associated with Purkinje cell loss and hypofunction (i.e. Tsai et al., 2012; Rogers et al., 2013; Courchesne et al., 1994 and others). However, the cell loss that is typically observed is exceedingly mild compared to the complete ablation of Purkinje cells. Therefore, the utility of examining complete Purkinje cell ablation as a model for understanding how ASD alters cerebellothalamic circuit development is minimal at best. This criticism is tempered by the fact that the authors make the argument that the primary goal of the paper is basic biology, namely examining how the cerebellum contributes to thalamic circuit function. Therefore, the use of the DTA ablation model does provide insight into the most extreme possible outcome. Recommended Action: To remedy these concerns, the authors should repeat similar experiments in the homozygous Pkj-cell specific TSC1 knockout, which shows both Purkinje cell hypofunction and moderate Purkinje cell loss, which more accurately recapitulates the pathology observed in ASD. The DTA ablation can then serve as an 'upper bound' of the deficits observed in mice with disrupted cerebellar function.

Please see our response to the editor's comment #1.

2) The authors use the pcp2-cre line to target Purkinje cells, and although this is the standard line used in the field, it does present some issues for studying synaptic development. The pcp2 gene is turned on at around PND6, however, full penetrance is not observed until 2-3 weeks of age (Barski et al., 2000). Therefore, ablation of Purkinje cells using the pcp-2 line may not be fully observed until 3-4 weeks of age. Recommended Action: To more accurately examine the timecourse of Purkinje cell loss, the authors should show a timecourse of Purkinje cell loss across development in the DTA-ablated animals. This point should also be more thoroughly addressed in the text/discussion. If the timecourse of Purkinje cell loss occurs after normal

circuit development, then the effects observed may underestimate the effects of cerebellar dysfunction during critical periods. Figure 1 would also benefit from a more thorough quantification of cell density/morphology in all the genetic models used.

Please see our response to the editor's comment #2.

Since the time course of Purkinje cell loss in the hemizygous and homozygous Tsc-1 deletion was quantified by Tsai et al. (2012), we did not repeat the analysis. However, we described their findings in the Results and Discussion sections (page 27, lines 683-685, and page 36, lines 910-916).

3) The authors make the assumption that Purkinje cell loss in the DTA-treated mice will increase cerebellar nuclear cell activity, as Purkinje cells provide tonic inhibition of the cerebellar nuclei. This assumption, however, must be validated to properly interpret their results. Because cerebellar nuclear cells sit above threshold (Raman et al., 2000), they may be particularly prone to depolarization block in the absence of any synaptic inhibition. Therefore, Purkinje cell ablation may increase nuclear cell activity, or may reduce it if the cells are in depolarization block due to sodium channel inactivation. Recommended Action: The authors should explicitly record from cerebellar nuclear cells in control and DTA treated animals to assess spontaneous action potential firing in the DTA treated animals.

Please see our response to the editor's comment #4.

Minor Comments:

1) The title should either read "strengthens the cerebellothalamic synapse" or "strengthens cerebellothalamic synapses".

We changed the statement in the title to "strengthen cerebellothalamic synapses." The verb was changed from "strengthens" to "strengthen" because we also changed the subject.

2) Figure 2 and 3 do not have any group data. The relative control data is replicated and further quantified in Figure 4/5 (i.e. paired pulse ratio and AMPA/NMDA ratio). Therefore, figure 2/3 should either be incorporated into figure 4/5 or perhaps figure 2 and 3 can be combined as a methodological figure. Alternatively, the authors can add group data for AMPA/NMDA ratio and the distribution of cells with different number of presynaptic partners to figures 2 and 3.

We combined Figures 2 and 3 in the original manuscript into a single methodological figure (new Figure 3).

3) Clarify how NMDA measurements were made. Generally, NMDA current amplitude is measured at a latency of ~40 ms to prevent contamination by AMPA receptor-mediated current. It is not clear if that method was used here.

We reanalyzed NMDA measurements because we measured the peak amplitude of EPSCs at +40 mV in the original manuscript. However, even slowly decaying EPSCs +40 mV substantially

decay after 40 msec. Therefore, we sought the optimal timing at which AMPA currents mostly decay, whereas NMDA currents are still near the peak. Based on the decay constants of AMPA currents obtained from double exponential curve fitting, we calculated that approximately 92% of AMPA currents decay 10 msec after the peak (approximately 13 msec after the photostimulation). At this time, EPSCs at +40 mV are only a few milliseconds after the peak and decay only slightly. Therefore, we measured the amplitude of EPSCs at +40 mV 13 msec after the photostimulation. This information is described in the “Voltage-clamp recordings in the VL” section of the Methods section (page 14, lines 341-349).

4) The choice of cut off for amplitudes greater than 1 nA seems quite large, despite being driver-type synapses. Can this be further explained or justified in the text (i.e. how many cells were excluded due to this criteria?)

Unfortunately, we cannot accurately tell the number of excluded cells because we did not record EPSCs when the photostimulation elicited no or tiny currents. However, those cells—their driver inputs were not labeled by ChR2 or were severely damaged by slicing—were relatively minor. Most VL neurons (approximately 70-80%) we patch-clamped showed synaptic currents larger than 1 nA at -70 mV. Among all the recorded cells, we excluded only 4% of cells by this cutting-off criterion.

5) Pg 12 paragraph 3 - The authors mention that cre-mediated recombination in the PCP2 line starts at PND6, although true, it does not complete until 3-4 weeks of age (Barski et al., 2000). This should be mentioned in the text.

As explained in our response to the editor’s comment #2, we quantified the time course of Purkinje cell degeneration in the DTA mice. We believe that the new data and our explanation about them efficiently address this comment #5.

6) Pg 12 last paragraph - “the degree of cell loss exceeds the conditions found in most neurological disorders” is a vast understatement and must be reworded. To my knowledge there are no common neurological disorders in which >90% of Purkinje cells are lost.

We reworded the statement to “the degree of cell loss does not mimic common neurological conditions.” (page 26, lines 668-669).

7) Pg 13 paragraph 1 - the authors should clarify in the results what cerebellar nucleus was targeted.

Our injection coordinates primarily target the interposed nucleus, but the virus most likely spreads into the lateral and medial nucleus. This information is described in the “Animal injection” section of Materials and Methods (page 8, lines 188-189).

8) Figure 6/7 - scale bars and an absolute voltage measurement should be included in panels A, B, F, and G. Scale bars are added to the inset graphs, but not the main graphs. Additionally, for current clamp recordings, the graph must include an absolute voltage reference (i.e. dashed line at 0 mV) to assess actual membrane voltages.

The Figures 6/7 in the original manuscript are now Figures 5/7. We added the scale bars and an absolute voltage measurement in the main graphs.

Referee #3:

The bootstrapping method used in this manuscript is indeed a robust way to analyse these data. It avoids making assumptions about the underlying distributions and allows robust 95% CIs to be constructed. It's great to see the authors have included the 95% CIs throughout their results to enable to reader to view the uncertainty associated with each outcome.

We are thankful for the referee's strong support for our bootstrapping method. We keep it in this revision but with some improvements to make the analysis more reliable and informative.

We computed the estimated mean difference between the two groups in the original manuscript. However, the absolute mean difference does not inform the magnitude of the difference, i.e., how substantially the distribution differs between the groups. Therefore, we computed the effect sizes in this revision.

For normally distributed data, we computed the standardized mean difference known as Cohen's d , i.e., the mean difference divided by the pooled SD. Although it is one of the most commonly used parametric effect sizes, it is not appropriate for data not normally distributed. Therefore, we computed Cliff's delta as the non-parametric effect size for data not normally distributed. The details about the Cliff's delta are described in the "Statistical analysis" section of the Methods section (page 16, lines 401-411).

These modifications have two advantages. First, outliers and skewed data distributions potentially introduce bias even with bootstrapping. Computing Cliff's delta effect size allows us to avoid the risk. Second, by showing the effect sizes, we can quantify the magnitude of the effect (difference), allowing readers to see not only whether differences are statistically significant but also the magnitude and certainty of the differences.

The other analyses undertaken (mixed ANOVA and Pearsons r correlations) are also appropriate for the given research questions, though it would be important to confirm that the relevant assumptions for these tests have been checked and adjusted for, namely:

- That variables used for the Pearson correlation were normally distributed.

We confirmed the normality of the data distribution by the Shapiro-Wilk test. It is now described in the "Statistical analysis" section of the Methods section (page 17, line 435).

- The degree of asphericity associated with the paired factor (interstimulus interval?) used in the mixed ANOVA. If a substantial degree of asphericity was present (GG epsilon < .75), the Greenhouse-Geisser correction should be used to adjust the degrees of freedom and correct

the F-ratio. Otherwise, the Huynh-Feldt correction should be used if the GG epsilon was $> .75$. Please see the link below for more information: <https://statistics.laerd.com/statistical-guides/sphericity-statistical-guide-2.php>.

The degree of freedom was adjusted by the Greenhouse-Geisser correction in the original manuscript, but we did not state the information. This revision includes the information in the “Statistical analysis” section of the Methods section (page 17, lines 429-431).

END OF COMMENTS

Dear Dr Nishiyama,

Re: JP-RP-2024-285887R1 "Purkinje cell ablation and Purkinje cell-specific deletion of Tsc1 in the developing cerebellum strengthen cerebellothalamic synapses" by Hiroshi Nishiyama, Naoko Nishiyama, and Boris V Zemelman

Thank you for submitting your manuscript to The Journal of Physiology. It has been assessed by a Reviewing Editor and by 3 expert referees and we are pleased to tell you that it is acceptable for publication following satisfactory revision.

REVISION CHECKLIST:

We look forward to receiving your revised submission.

Yours sincerely,

Katalin Toth
Senior Editor
The Journal of Physiology

REQUIRED ITEMS

(1) You must start the Methods section with a paragraph headed Ethical approval (https://jp.msubmit.net/cgi-bin/main.plex?form_type=display_requirements#methods).

Research must comply with The Journal's policies regarding animal experiments (<https://physoc.onlinelibrary.wiley.com/hub/animal-experiments>) and adherence to these policies must be stated in the manuscript.

Authors should confirm in their Methods section that their experiments were carried out according to the guidelines laid down by their institution's animal welfare committee, ***including an ethics approval reference number***. The Methods section must contain a statement about access to food, water and housing, details of the anaesthetic regime: anaesthetic used, dose and route of administration, and method of killing the experimental animals.

(2) Please include an Abstract Figure file, ***as well as the Figure Legend text within the main article file***. The Abstract Figure is a piece of artwork designed to give readers an immediate understanding of the research and should summarise the main conclusions. If possible, the image should be easily 'readable' from left to right or top to bottom. It should show the physiological relevance of the manuscript so readers can assess the importance and content of its findings. Abstract Figures should not merely recapitulate other figures in the manuscript. Please try to keep the diagram as simple as possible and without superfluous information that may distract from the main conclusion(s). Abstract Figures must be provided by authors no later than the revised manuscript stage and should be uploaded as a separate file during online submission labelled as File Type 'Abstract Figure'. Please also ensure that you include the figure legend in the main article file. All Abstract Figures should be created using BioRender. Authors should use The Journal's premium BioRender account to export high-resolution images. Details on how to use and access the premium account are included as part of this email.

EDITOR COMMENTS

Reviewing Editor:

Your revised manuscript has been reviewed by two referees of your original submission. They were generally satisfied with how you addressed their prior concerns. The authors have also addressed my prior concerns around animal use and minor point about being clear that SDs were being used. Reviewer 1 did however have a few additional points that will require your further attention. They specifically were concerned about overstatements in conclusions made based on negative results in the hemizygous Tsc1 mice, given limitations in sample size, and overclaiming of the significance of the overall results for human autism. The Reviewer had several specific suggestions for changes in wording in the text and also requests power calculations for experiments in hemizygous Tsc1 mice (described in Comment 1A) that could shed light on the strength of the conclusion that can be made based on the negative results.

Senior Editor:

Please also see 'Required Items' above. At present, your manuscript is missing (a) an ethics committee approval reference number in the Methods, and (b) a legend to accompany your Abstract Figure. These two items must be provided in a revised version.

REFeree COMMENTS

Referee #1:

This manuscript is improved from the previous submission. The principal finding is that in mice, disruption of Purkinje cell output to the cerebellum leads to a specific increase in the unitary strength of connections from cerebellar nuclei to motor thalamus, and that these connections are individually quite strong, as they are in retinothalamic synapses. The authors do not detect changes in hemizygous knockout of the autism risk gene *Tsc1*, and suggest that therefore this pathway is insufficient to account for autism behavioral and social phenotypes in humans.

The specific findings are presented accurately and cautiously, and in enough detail that a reader can evaluate them. However, there are two major remaining problems that must be corrected in the manuscript: (1) underclaiming the possible role for cerebellothalamic (i.e. nucleothalamic) synaptic strength in driving developmental change, and (2) overclaiming the significance for human autism.

Major comments

Comment 1, sufficiency/insufficiency of observed changes to account for behavioral alteration. Ordinarily I would commend the authors for their close attention to statistical analysis, bootstrapping, and emphasizing when results are not significant. However, there is an issue here that needs to be addressed.

1A. The authors wish to claim that a lack of change in cerebellothalamic connections in hemizygous *Tsc1* mice means that those connections cannot be the cause of behavioral differences. However, the data do not support such a strong conclusion. Instead, it may simply be that the differences are smaller, and hard to detect.

The changes in AMPA EPSC amplitude, NMDA EPSC amplitude, resting potential, input resistance, and maximum AP number are all in the same direction for hemizygous and homozygous mice. See Tables 3 and 4. Although statistical significance was not reached for a number of these measurements, the shared direction for these five parameters cannot be ignored, and must be cited/discussed.

1B. The authors repeatedly call attention to effects that are near statistical significance, but this emphasis may be misplaced. A common problem in neuroscience and psychology is excessive reliance on p-values and insufficient attention to power and the risk of false positives/negatives. The abstract, results, and discussion must be modified to allow for the possibility that the authors' observed changes might indeed be causative of behavioral consequences.

In the methods and/or discussion, provide a power calculation to clarify the probability of detecting an effect given the N of observations. Report in a form such as "For N cells, the probability of detecting an effect size of $d=0.2$ is 0.xx, and of an effect size of $d=0.5$ is 0.xx." Also, the authors may be interested in the problems of underpowered studies (Button et al. 2013 Nat Rev Neurosci <https://doi.org/10.1038/nrn3475>; Colquhoun 2017 Proc Roy Soc <https://doi.org/10.1098/rsos.171085>)

1C. Page 3 lines 80-81: change "These results suggest that autistic behaviors are not necessarily linked to thalamic abnormality, but the cerebellothalamic circuit is vulnerable to substantial disturbances in the developing cerebellum." to "These results therefore suggest that although the nucleothalamic projection is vulnerable to disturbances in the developing cerebellar cortex, other changes may also drive the behavioral consequences observed."

1D. Page 36 line 912: please evaluate the possibility that your study was underpowered, and that hemizygous mice have differences in cerebellothalamic strength, but you could not detect it.

1E. Page 36 lines 929-930: "This phenomenon is unlikely to be explained by the enhanced cerebellothalamic synapses." This statement is unsupported. I suggest deleting it and change "Indeed," to "However,"

Comment 2, specificity of observed changes. It would be good to clarify that the observed differences are those in unitary synaptic strength. The following line edits may help.

2A. Page 24 line 613: "the synaptic strength was higher" perhaps you mean "the strength of putative unitary cerebellothalamic connections"? Better to clarify.

2B. Page 31 line 787: change "strengthened cerebellothalamic synapses" to "strengthened UNITARY cerebellothalamic synapses" to clarify that individual connections were strengthened

2C. Page 33 line 849: change "related to synaptic strength measurements" to "related to measurements of unitary synaptic connections"

Comment 3, autism in mice. Claims that the authors are studying autism go beyond what is generally claimed for animal models. Similarly, cerebellar disruption is not necessarily frequent in autism but rather, when it is disrupted, autism often results.

3A. Page 3 line 79: change "cause autistic behaviors" to "cause AUTISM-LIKE behaviors" and again throughout manuscript, replacing all instances of "autistic"

3B. Page 3 line 84 change "autism spectrum disorders" to "autism spectrum disorder" and change this throughout, i.e. ASD instead of ASDs

3C. Page 3 lines 83-84 change "A cerebellar abnormality is one of the most prominent characteristics of brain pathology in" to "Cerebellar abnormality occurs in a variety of"

3D. Page 4 line 95: cerebellar injury at birth is not a large risk factor for ASD. Instead say "cerebellar injury at birth has one of the highest known risk ratios for ASD"

3E. Page 34 line line 859: change "unlikely related to social and cognitive deficits in autism" to "may not be the sole means by which social and cognitive deficits in Tsc1 model mice arise" Two conceptual errors must be corrected here, even if we concede that hemizygous mutation does not lead to changes in unitary strength. First, if cerebellothalamic unitary synaptic strength and behavioral deficits are both consequences of disrupted Purkinje cell output, increases in unitary connection

strength may still reflect a plasticity process in this pathway or in other pathways from cerebellum to the rest of the brain. Second, mice do not have autism - they serve only as a model for the consequences of perturbations that cause autism in humans. It is incorrect to extrapolate from mouse behavior to human autism in such a direct manner.

Minor comments

I suggest the following line edits.

Page 3 line 68: change "thalamus is the direct target of the cerebellum, sending cerebellar outputs" to "thalamus is A MAJOR direct target of the cerebellAR NUCLEI, CONVEYING cerebellar INFLUENCE"

Page 3 line 72: consider changing "cerebellothalamic" to "NUCLEOthalamic" here and throughout the entire manuscript

Page 6 line 143: spell out what DTA stands for

Page 7 lines 162-164: currently does not state what protein expression was driven by the AAV

Page 16 line 411: change "affected" to "distorted"

Page 20 line 520: "essentially the same" - more accurate might be "similar" since I believe Lorente de No observed that some regions of cerebral cortex had spatially dispersed thalamocortical projections, whereas sensory/motor regions had more focal projections

Page 23 line 592: change "that lose" to "which lose" since you mean that on average they all do it; "that" is restrictive and "which" is nonrestrictive

Page 26 line 660: change "in the DTA mice" to "in brain slices from the DTA mice" to properly convey that these experiments were not done under in vivo conditions. Do this here and elsewhere (line 661...)

Page 26 line 671: delete "complex" in "Tuberous sclerosis complex-1" since it is not part of the definition of Tsc1 (sc="sclerosis")

Page 34 line 862: change "due to the stronger synaptic input" to "due to stronger unitary synaptic strength"

Page 35 line 895: the authors may wish to insert an explanatory phrase or sentence such as "Thus in DTA mice and Tsc1 mutant mice, mossy and climbing fiber excitation are transmitted with reduced countering inhibition through the DCN and to the thalamus and other brain regions." This helps the reader understand that the mice studied do not only have increased tonic DCN output, but may transmit inputs to the cerebellum more strongly to the rest of the brain.

Page 37 Lines 933-945 the text may be more persuasive and have more impact if you changed "Therefore, it is reasonable

to assume that" to "We speculate that"

Page 37 Line 941 "affects the thalamus" to "affects inputs to the thalamus".

Referee #2:

The authors have adequately addressed all my concerns. I congratulate them on a fantastic study.

Referee #3:

I am happy with the statistical analysis approach used and the description provided.

END OF COMMENTS

1st Confidential Review

05-Sep-2024

This manuscript is improved from the previous submission. The principal finding is that in mice, disruption of Purkinje cell output to the cerebellum leads to a specific increase in the unitary strength of connections from cerebellar nuclei to motor thalamus, and that these connections are individually quite strong, as they are in retinothalamic synapses. The authors do not detect changes in hemizygous knockout of the autism risk gene *Tsc1*, and suggest that therefore this pathway is insufficient to account for autism behavioral and social phenotypes in humans.

The specific findings are presented accurately and cautiously, and in enough detail that a reader can evaluate them. However, there are two major remaining problems that must be corrected in the manuscript: (1) underclaiming the possible role for cerebellothalamic (i.e. nucleothalamic) synaptic strength in driving developmental change, and (2) overclaiming the significance for human autism.

Major comments

Comment 1, sufficiency/insufficiency of observed changes to account for behavioral alteration. Ordinarily I would commend the authors for their close attention to statistical analysis, bootstrapping, and emphasizing when results are not significant. However, there is an issue here that needs to be addressed.

1A. The authors wish to claim that a lack of change in cerebellothalamic connections in hemizygous *Tsc1* mice means that those connections cannot be the cause of behavioral differences. However, the data do not support such a strong conclusion. Instead, it may simply be that the differences are smaller, and hard to detect.

The changes in AMPA EPSC amplitude, NMDA EPSC amplitude, resting potential, input resistance, and maximum AP number are all in the same direction for hemizygous and homozygous mice. See Tables 3 and 4. Although statistical significance was not reached for a number of these measurements, the shared direction for these five parameters cannot be ignored, and must be cited/discussed.

1B. The authors repeatedly call attention to effects that are near statistical significance, but this emphasis may be misplaced. A common problem in neuroscience and psychology is excessive reliance on p-values and insufficient attention to power and the risk of false positives/negatives. The abstract, results, and discussion must be modified to allow for the possibility that the authors' observed changes might indeed be causative of behavioral consequences.

In the methods and/or discussion, provide a power calculation to clarify the probability of detecting an effect given the N of observations. Report in a form such as "For N cells, the probability of detecting an effect size of $d=0.2$ is 0.xx, and of an effect size of $d=0.5$ is 0.xx." Also, the authors may be interested in the problems of underpowered studies (Button et al. 2013 *Nat Rev Neurosci* <https://doi.org/10.1038/nrn3475>; Colquhoun 2017 *Proc Roy Soc* <https://doi.org/10.1098/rsos.171085>)

1C. Page 3 lines 80-81: change "These results suggest that autistic behaviors are not necessarily linked to thalamic abnormality, but the cerebellothalamic circuit is vulnerable to substantial disturbances in the developing cerebellum." to "These results therefore suggest that

although the nucleothalamic projection is vulnerable to disturbances in the developing cerebellar cortex, other changes may also drive the behavioral consequences observed.”

1D. Page 36 line 912: please evaluate the possibility that your study was underpowered, and that hemizygous mice have differences in cerebellothalamic strength, but you could not detect it.

1E. Page 36 lines 929-930: “This phenomenon is unlikely to be explained by the enhanced cerebellothalamic synapses.” This statement is unsupported. I suggest deleting it and change “Indeed,” to “However,”

Comment 2, specificity of observed changes. It would be good to clarify that the observed differences are those in unitary synaptic strength. The following line edits may help.

2A. Page 24 line 613: “the synaptic strength was higher” perhaps you mean “the strength of putative unitary cerebellothalamic connections”? Better to clarify.

2B. Page 31 line 787: change “strengthened cerebellothalamic synapses” to “strengthened UNITARY cerebellothalamic synapses” to clarify that individual connections were strengthened

2C. Page 33 line 849: change “related to synaptic strength measurements” to “related to measurements of unitary synaptic connections”

Comment 3, autism in mice. Claims that the authors are studying autism go beyond what is generally claimed for animal models. Similarly, cerebellar disruption is not necessarily frequent in autism but rather, when it is disrupted, autism often results.

3A. Page 3 line 79: change “cause autistic behaviors” to “cause AUTISM-LIKE behaviors” and again throughout manuscript, replacing all instances of “autistic”

3B. Page 3 line 84 change “autism spectrum disorders” to “autism spectrum disorder” and change this throughout, i.e. ASD instead of ASDs

3C. Page 3 lines 83-84 change “A cerebellar abnormality is one of the most prominent characteristics of brain pathology in” to “Cerebellar abnormality occurs in a variety of”

3D. Page 4 line 95: cerebellar injury at birth is not a large risk factor for ASD. Instead say “cerebellar injury at birth has one of the highest known risk ratios for ASD”

3E. Page 34 line line 859: change “unlikely related to social and cognitive deficits in autism” to “may not be the sole means by which social and cognitive deficits in Tsc1 model mice arise” Two conceptual errors must be corrected here, even if we concede that hemizygous mutation does not lead to changes in unitary strength. First, if cerebellothalamic unitary synaptic strength and behavioral deficits are both consequences of disrupted Purkinje cell output, increases in unitary connection strength may still reflect a plasticity process in this pathway or in other pathways from cerebellum to the rest of the brain. Second, mice do not have autism – they serve only as a model for the consequences of perturbations that cause autism in humans. It is incorrect to extrapolate from mouse behavior to human autism in such a direct manner.

Minor comments

I suggest the following line edits.

Page 3 line 68: change “thalamus is the direct target of the cerebellum, sending cerebellar outputs” to “thalamus is A MAJOR direct target of the cerebellAR NUCLEI, CONVEYING cerebellar INFLUENCE”

Page 3 line 72: consider changing “cerebellothalamic” to “NUCLEOthalamic” here and throughout the entire manuscript

Page 6 line 143: spell out what DTA stands for

Page 7 lines 162-164: currently does not state what protein expression was driven by the AAV

Page 16 line 411: change “affected” to “distorted”

Page 20 line 520: “essentially the same” – more accurate might be “similar” since I believe Lorente de No observed that some regions of cerebral cortex had spatially dispersed thalamocortical projections, whereas sensory/motor regions had more focal projections

Page 23 line 592: change “that lose” to “which lose” since you mean that on average they all do it; “that” is restrictive and “which” is nonrestrictive

Page 26 line 660: change “in the DTA mice” to “in brain slices from the DTA mice” to properly convey that these experiments were not done under in vivo conditions. Do this here and elsewhere (line 661...)

Page 26 line 671: delete “complex” in “Tuberous sclerosis complex-1” since it is not part of the definition of Tsc1 (sc=“sclerosis”)

Page 34 line 862: change “due to the stronger synaptic input” to “due to stronger unitary synaptic strength”

Page 35 line 895: the authors may wish to insert an explanatory phrase or sentence such as “Thus in DTA mice and Tsc1 mutant mice, mossy and climbing fiber excitation are transmitted with reduced countering inhibition through the DCN and to the thalamus and other brain regions.” This helps the reader understand that the mice studied do not only have increased tonic DCN output, but may transmit inputs to the cerebellum more strongly to the rest of the brain.

Page 37 Lines 933-945 the text may be more persuasive and have more impact if you changed “Therefore, it is reasonable to assume that” to “We speculate that”

Page 37 Line 941 “affects the thalamus” to “affects inputs to the thalamus”.

This manuscript is improved from the previous submission. The principal finding is that in mice, disruption of Purkinje cell output to the cerebellum leads to a specific increase in the unitary strength of connections from cerebellar nuclei to motor thalamus, and that these connections are individually quite strong, as they are in retinothalamic synapses. The authors do not detect changes in hemizygous knockout of the autism risk gene *Tsc1*, and suggest that therefore this pathway is insufficient to account for autism behavioral and social phenotypes in humans.

The specific findings are presented accurately and cautiously, and in enough detail that a reader can evaluate them. However, there are two major remaining problems that must be corrected in the manuscript: (1) underclaiming the possible role for cerebellothalamic (i.e. nucleothalamic) synaptic strength in driving developmental change, and (2) overclaiming the significance for human autism.

Major comments

Comment 1, sufficiency/insufficiency of observed changes to account for behavioral alteration. Ordinarily I would commend the authors for their close attention to statistical analysis, bootstrapping, and emphasizing when results are not significant. However, there is an issue here that needs to be addressed.

1A. The authors wish to claim that a lack of change in cerebellothalamic connections in hemizygous *Tsc1* mice means that those connections cannot be the cause of behavioral differences. However, the data do not support such a strong conclusion. Instead, it may simply be that the differences are smaller, and hard to detect.

The changes in AMPA EPSC amplitude, NMDA EPSC amplitude, resting potential, input resistance, and maximum AP number are all in the same direction for hemizygous and homozygous mice. See Tables 3 and 4. Although statistical significance was not reached for a number of these measurements, the shared direction for these five parameters cannot be ignored, and must be cited/discussed.

1B. The authors repeatedly call attention to effects that are near statistical significance, but this emphasis may be misplaced. A common problem in neuroscience and psychology is excessive reliance on p-values and insufficient attention to power and the risk of false positives/negatives. The abstract, results, and discussion must be modified to allow for the possibility that the authors' observed changes might indeed be causative of behavioral consequences.

In the methods and/or discussion, provide a power calculation to clarify the probability of detecting an effect given the N of observations. Report in a form such as "For N cells, the probability of detecting an effect size of $d=0.2$ is 0.xx, and of an effect size of $d=0.5$ is 0.xx." Also, the authors may be interested in the problems of underpowered studies (Button et al. 2013 *Nat Rev Neurosci* <https://doi.org/10.1038/nrn3475>; Colquhoun 2017 *Proc Roy Soc* <https://doi.org/10.1098/rsos.171085>)

1C. Page 3 lines 80-81: change "These results suggest that autistic behaviors are not necessarily linked to thalamic abnormality, but the cerebellothalamic circuit is vulnerable to substantial disturbances in the developing cerebellum." to "These results therefore suggest that

although the nucleothalamic projection is vulnerable to disturbances in the developing cerebellar cortex, other changes may also drive the behavioral consequences observed.”

1D. Page 36 line 912: please evaluate the possibility that your study was underpowered, and that hemizygous mice have differences in cerebellothalamic strength, but you could not detect it.

1E. Page 36 lines 929-930: “This phenomenon is unlikely to be explained by the enhanced cerebellothalamic synapses.” This statement is unsupported. I suggest deleting it and change “Indeed,” to “However,”

Comment 2, specificity of observed changes. It would be good to clarify that the observed differences are those in unitary synaptic strength. The following line edits may help.

2A. Page 24 line 613: “the synaptic strength was higher” perhaps you mean “the strength of putative unitary cerebellothalamic connections”? Better to clarify.

2B. Page 31 line 787: change “strengthened cerebellothalamic synapses” to “strengthened UNITARY cerebellothalamic synapses” to clarify that individual connections were strengthened

2C. Page 33 line 849: change “related to synaptic strength measurements” to “related to measurements of unitary synaptic connections”

Comment 3, autism in mice. Claims that the authors are studying autism go beyond what is generally claimed for animal models. Similarly, cerebellar disruption is not necessarily frequent in autism but rather, when it is disrupted, autism often results.

3A. Page 3 line 79: change “cause autistic behaviors” to “cause AUTISM-LIKE behaviors” and again throughout manuscript, replacing all instances of “autistic”

3B. Page 3 line 84 change “autism spectrum disorders” to “autism spectrum disorder” and change this throughout, i.e. ASD instead of ASDs

3C. Page 3 lines 83-84 change “A cerebellar abnormality is one of the most prominent characteristics of brain pathology in” to “Cerebellar abnormality occurs in a variety of”

3D. Page 4 line 95: cerebellar injury at birth is not a large risk factor for ASD. Instead say “cerebellar injury at birth has one of the highest known risk ratios for ASD”

3E. Page 34 line line 859: change “unlikely related to social and cognitive deficits in autism” to “may not be the sole means by which social and cognitive deficits in Tsc1 model mice arise” Two conceptual errors must be corrected here, even if we concede that hemizygous mutation does not lead to changes in unitary strength. First, if cerebellothalamic unitary synaptic strength and behavioral deficits are both consequences of disrupted Purkinje cell output, increases in unitary connection strength may still reflect a plasticity process in this pathway or in other pathways from cerebellum to the rest of the brain. Second, mice do not have autism – they serve only as a model for the consequences of perturbations that cause autism in humans. It is incorrect to extrapolate from mouse behavior to human autism in such a direct manner.

Minor comments

I suggest the following line edits.

Page 3 line 68: change “thalamus is the direct target of the cerebellum, sending cerebellar outputs” to “thalamus is A MAJOR direct target of the cerebellAR NUCLEI, CONVEYING cerebellar INFLUENCE”

Page 3 line 72: consider changing “cerebellothalamic” to “NUCLEOthalamic” here and throughout the entire manuscript

Page 6 line 143: spell out what DTA stands for

Page 7 lines 162-164: currently does not state what protein expression was driven by the AAV

Page 16 line 411: change “affected” to “distorted”

Page 20 line 520: “essentially the same” – more accurate might be “similar” since I believe Lorente de No observed that some regions of cerebral cortex had spatially dispersed thalamocortical projections, whereas sensory/motor regions had more focal projections

Page 23 line 592: change “that lose” to “which lose” since you mean that on average they all do it; “that” is restrictive and “which” is nonrestrictive

Page 26 line 660: change “in the DTA mice” to “in brain slices from the DTA mice” to properly convey that these experiments were not done under in vivo conditions. Do this here and elsewhere (line 661...)

Page 26 line 671: delete “complex” in “Tuberous sclerosis complex-1” since it is not part of the definition of Tsc1 (sc=“sclerosis”)

Page 34 line 862: change “due to the stronger synaptic input” to “due to stronger unitary synaptic strength”

Page 35 line 895: the authors may wish to insert an explanatory phrase or sentence such as “Thus in DTA mice and Tsc1 mutant mice, mossy and climbing fiber excitation are transmitted with reduced countering inhibition through the DCN and to the thalamus and other brain regions.” This helps the reader understand that the mice studied do not only have increased tonic DCN output, but may transmit inputs to the cerebellum more strongly to the rest of the brain.

Page 37 Lines 933-945 the text may be more persuasive and have more impact if you changed “Therefore, it is reasonable to assume that” to “We speculate that”

Page 37 Line 941 “affects the thalamus” to “affects inputs to the thalamus”.

EDITOR COMMENTS

Reviewing Editor:

Your revised manuscript has been reviewed by two referees of your original submission. They were generally satisfied with how you addressed their prior concerns. The authors have also addressed my prior concerns around animal use and minor point about being clear that SDs were being used. Reviewer 1 did however have a few additional points that will require your further attention. They specifically were concerned about overstatements in conclusions made based on negative results in the hemizygous Tsc1 mice, given limitations in sample size, and overclaiming of the significance of the overall results for human autism. The Reviewer had several specific suggestions for changes in wording in the text and also requests power calculations for experiments in hemizygous Tsc1 mice (described in Comment 1A) that could shed light on the strength of the conclusion that can be made based on the negative results.

We are delighted to hear that all Referees are generally satisfied with our revised manuscript and that it is now acceptable for publication following satisfactory revisions. As detailed below, we have fully addressed the Referees' comments and followed their suggestions to the greatest extent possible. On a few points where our views differ from those of the Referee, we have explained the reasons for our disagreement. Overall, the referees' feedback was invaluable in helping us present our findings accurately.

Senior Editor:

Please also see 'Required Items' above. At present, your manuscript is missing (a) an ethics committee approval reference number in the Methods, and (b) a legend to accompany your Abstract Figure. These two items must be provided in a revised version.

Thank you for the clarification. We included these required items in this revision. Since we renewed our animal protocol once during this study, we listed the number of expired and current protocols (page 6, lines 139-140).

REFEREE COMMENTS

Referee #1:

This manuscript is improved from the previous submission. The principal finding is that in mice, disruption of Purkinje cell output to the cerebellum leads to a specific increase in the unitary strength of connections from cerebellar nuclei to motor thalamus, and that these connections are individually quite strong, as they are in retinthalamic synapses. The authors do not detect changes in hemizygous knockout of the autism risk gene Tsc1, and suggest that therefore this pathway is insufficient to account for autism behavioral and social phenotypes in humans.

The specific findings are presented accurately and cautiously, and in enough detail that a reader

can evaluate them. However, there are two major remaining problems that must be corrected in the manuscript: (1) underclaiming the possible role for cerebellothalamic (i.e. nucleothalamic) synaptic strength in driving developmental change, and (2) overclaiming the significance for human autism.

We sincerely appreciate the Referee's positive review and thoughtful suggestions for improving our manuscript. We have incorporated most of the Referee's suggestions. In the few instances where our perspective differs, we have explained our differing views.

Major comments

Comment 1, sufficiency/insufficiency of observed changes to account for behavioral alteration. Ordinarily I would commend the authors for their close attention to statistical analysis, bootstrapping, and emphasizing when results are not significant. However, there is an issue here that needs to be addressed.

1A. The authors wish to claim that a lack of change in cerebellothalamic connections in hemizygous *Tsc1* mice means that those connections cannot be the cause of behavioral differences. However, the data do not support such a strong conclusion. Instead, it may simply be that the differences are smaller, and hard to detect.

The changes in AMPA EPSC amplitude, NMDA EPSC amplitude, resting potential, input resistance, and maximum AP number are all in the same direction for hemizygous and homozygous mice. See Tables 3 and 4. Although statistical significance was not reached for a number of these measurements, the shared direction for these five parameters cannot be ignored, and must be cited/discussed.

This comment highlights two main concerns: one regarding how we present the negative results of hemizygous *Tsc1* deletion mice and the other on how we contextualize these negative results more broadly. We address each concern in turn.

We share the Referee's view on the over-reliance on p-values in our research field (Comment 1B). Accordingly, our primary analysis does not rely on conventional null hypothesis testing. Instead, we calculated effect sizes, which provide standardized measures of the magnitude of differences. Unlike p-values and statistical power in null hypothesis testing, effect sizes are independent of sample sizes.

We also used bootstrap resampling to estimate the effect size distribution and its 95% or 97.5% confidence intervals. This approach allows readers to assess our results based on the magnitude and certainty of the effects rather than relying solely on statistical significance.

The effect sizes (Cohen's *d*) for the hemizygous *Tsc1* deletion mice are below 0.2 for AMPA amplitude, AMPA charge, and NMDA amplitude (Table 4). Cohen classifies $d = 0.2$ as a small effect, and $d < 0.2$ is typically considered negligible, even if statistical significance is achieved.

We are not claiming that synaptic strengthening does not occur in the hemizygous Tsc1 deletion mice purely because of statistically non-significant results. Rather, our conclusion is primarily based on the negligible effect sizes for these synaptic strength measures and the fact that their distributions are centered near zero (Fig. 6E-G). These data suggest that a meaningful difference would unlikely emerge even with larger sample sizes, making it inappropriate to speculate otherwise. Please also refer to our response to Comment 1B for further clarification.

That said, we agree with the Referee that we must be cautious in interpreting the results. While it is necessary to relate our findings to previous studies on the Tsc1 deletion mice behaviors, we acknowledge that extrapolating from mouse behavior to human autism is not appropriate. Though we did not intend such extrapolation, certain sentences in the previous manuscript may have implied this. In this revision, we have revised the wording as per the Referee's suggestions in Comment 3 to avoid any overstatements.

1B. The authors repeatedly call attention to effects that are near statistical significance, but this emphasis may be misplaced. A common problem in neuroscience and psychology is excessive reliance on p-values and insufficient attention to power and the risk of false positives/negatives. The abstract, results, and discussion must be modified to allow for the possibility that the authors' observed changes might indeed be causative of behavioral consequences.

In the methods and/or discussion, provide a power calculation to clarify the probability of detecting an effect given the N of observations. Report in a form such as "For N cells, the probability of detecting an effect size of $d=0.2$ is 0.xx, and of an effect size of $d=0.5$ is 0.xx." Also, the authors may be interested in the problems of underpowered studies (Button et al. 2013 Nat Rev Neurosci <https://doi.org/10.1038/nrn3475>; Colquhoun 2017 Proc Roy Soc <https://doi.org/10.1098/rsos.171085>)

This comment addresses how we present the negative results of the hemizygous Tsc1 deletion mice. Since this largely overlaps with Comment 1A, please also see our response to that comment. Here, we focus on the Referee's request for a power analysis.

If we aim for a statistical power of 0.8 (i.e., a 20% chance of a false negative) to detect a small effect (Cohen's $d = 0.2$) using a two-sample t-test ($p < 0.05$), we would need approximately 400 samples per group. Such a sample size is impractical for whole-cell patch-clamp recordings. Even if we reduce the power to 0.5 (i.e., a 50% chance of a false negative), we would still require around 200 samples per group.

We believe the appropriate interpretation here is not that the sample size is insufficient but rather that the potential difference is too small to detect within a feasible experimental design. Therefore, we respectfully disagree with the suggestion to conduct a power analysis and speculate that a difference might exist.

While most similar studies only report p-values, we report effect sizes and their estimated distributions with a confidence interval. As discussed in our response to Comment 1A, these data suggest that a potential enhancement of synaptic strength in the hemizygous Tsc1 deletion mice is likely negligible, if any.

1C. Page 3 lines 80-81: change “These results suggest that autistic behaviors are not necessarily linked to thalamic abnormality, but the cerebellothalamic circuit is vulnerable to substantial disturbances in the developing cerebellum.” to “These results therefore suggest that although the nucleothalamic projection is vulnerable to disturbances in the developing cerebellar cortex, other changes may also drive the behavioral consequences observed.”

We revised the sentence accordingly with a few minor modifications (page 3, lines 80-82). First, we removed “therefore” from the Referee’s suggestion. Second, we changed “behavioral consequences observed” to “behavioral consequences of early cerebellar perturbation.” Third, we retain the term “cerebellothalamic.” A recent PubMed search shows that while the keyword “cerebellothalamic” yields 195 results, “nucleothalamic” returns only one, indicating that “cerebellothalamic” is more commonly used. Moreover, “cerebellothalamic” clearly specifies that our study focuses on the projection from the cerebellum to the thalamus, which is not conveyed by the term “nucleothalamic.”

1D. Page 36 line 912: please evaluate the possibility that your study was underpowered, and that hemizygous mice have differences in cerebellothalamic strength, but you could not detect it.

Please see our responses to Comments 1A and 1B.

1E. Page 36 lines 929-930: “This phenomenon is unlikely to be explained by the enhanced cerebellothalamic synapses.” This statement is unsupported. I suggest deleting it and change “Indeed,” to “However,”

We revised the sentences accordingly (page 37, lines 936-937).

Comment 2, specificity of observed changes. It would be good to clarify that the observed differences are those in unitary synaptic strength. The following line edits may help.

We appreciate the Referee’s suggestions (Comments 2A-C).

Since a similar number of cerebellothalamic fibers innervate individual VL neurons across genotypes, one could argue that the larger EPSCs observed in the DTA mice and homozygous *Tsc1* deletion mice are due to the strengthening of unitary synaptic inputs (the inputs of single cerebellothalamic fibers). However, a single fiber input can be reliably measured only for the first EPSC evoked by the minimum stimulus intensity. This reflects the response of the most light-sensitive fiber, which varies based on its location in the tissue or ChR2 expression level. Its amplitude ranges dramatically, from under 100 pA to several nA, making comparisons of single fiber inputs across genotypes challenging. Therefore, we are hesitant to argue that unitary synaptic inputs are strengthened.

In response to the Referee’s request for clarification, we have revised the Discussion section to add the above explanation and clarify that this study demonstrates the strengthening of total synaptic input (pages 33-34, lines 852-860).

2A. Page 24 line 613: “the synaptic strength was higher” perhaps you mean “the strength of putative unitary cerebellothalamic connections”? Better to clarify.

Please see our response to Comment 2 above.

2B. Page 31 line 787: change “strengthened cerebellothalamic synapses” to “strengthened UNITARY cerebellothalamic synapses” to clarify that individual connections were strengthened

Please see our response to Comment 2 above.

2C. Page 33 line 849: change “related to synaptic strength measurements” to “related to measurements of unitary synaptic connections”

Please see our response to Comment 2 above.

Comment 3, autism in mice. Claims that the authors are studying autism go beyond what is generally claimed for animal models. Similarly, cerebellar disruption is not necessarily frequent in autism but rather, when it is disrupted, autism often results.

To address this concern, we revised the text as suggested by the Referee. Please see below for details.

3A. Page 3 line 79: change “cause autistic behaviors” to “cause AUTISM-LIKE behaviors” and again throughout manuscript, replacing all instances of “autistic”

We changed “autistic” to “autism-like” throughout the manuscript.

3B. Page 3 line 84 change “autism spectrum disorders” to “autism spectrum disorder” and change this throughout, i.e. ASD instead of ASDs

We changed the wording accordingly throughout the manuscript.

3C. Page 3 lines 83-84 change “A cerebellar abnormality is one of the most prominent characteristics of brain pathology in” to “Cerebellar abnormality occurs in a variety of”

We changed the wording accordingly (page 4, lines 84-87).

3D. Page 4 line 95: cerebellar injury at birth is not a large risk factor for ASD. Instead say “cerebellar injury at birth has one of the highest known risk ratios for ASD”

We changed the wording accordingly (page 4, line 95). Since Wang et al. (2014) reported ASD risk as the ratio compared to the general population, “risk ratio” more accurately represents the finding than “risk factor.”

3E. Page 34 line line 859: change “unlikely related to social and cognitive deficits in autism” to “may not be the sole means by which social and cognitive deficits in Tsc1 model mice arise” Two

conceptual errors must be corrected here, even if we concede that hemizygous mutation does not lead to changes in unitary strength. First, if cerebellothalamic unitary synaptic strength and behavioral deficits are both consequences of disrupted Purkinje cell output, increases in unitary connection strength may still reflect a plasticity process in this pathway or in other pathways from cerebellum to the rest of the brain. Second, mice do not have autism - they serve only as a model for the consequences of perturbations that cause autism in humans. It is incorrect to extrapolate from mouse behavior to human autism in such a direct manner.

We changed the wording accordingly with a slight modification (“deficits in Tsc1 model mice arise” to “deficits arise in Tsc1 deletion mice”) (page 34, lines 862-863). Although we did not intend to extrapolate from mouse behavior to human autism directly, our original sentence can be read in such a way. We are thankful for the Referee pointing it out and allowing us to correct it.

Minor comments

I suggest the following line edits.

Page 3 line 68: change “thalamus is the direct target of the cerebellum, sending cerebellar outputs” to “thalamus is A MAJOR direct target of the cerebellAR NUCLEI, CONVEYING cerebellar INFLUENCE”

We edited the wording accordingly with a minor modification (“cerebellar influence” to “cerebellar signals”) (page 3, line 68)

Page 3 line 72: consider changing “cerebellothalamic” to “NUCLEOthalamic” here and throughout the entire manuscript

As we explained in our response to the Referee’s major comment 1C, “cerebellothalamic” is more commonly used than “nucleothalamic” to specify the projection from the cerebellar nuclei to the thalamus. Therefore, we want to retain “cerebellothalamic.”

Page 6 line 143: spell out what DTA stands for

We changed “DTA” to “diphtheria toxin fragment A (DTA) (page 6, line 143).”

Page 7 lines 162-164: currently does not state what protein expression was driven by the AAV

We state “channelrhodopsin2-tdTomato” on page 7, line 163.

Page 16 line 411: change “affected” to “distorted”

We changed “affected” to “distorted.” (page 16, line 407)

Page 20 line 520: “essentially the same” - more accurate might be “similar” since I believe Lorente de No observed that some regions of cerebral cortex had spatially dispersed

thalamocortical projections, whereas sensory/motor regions had more focal projections

We changed “essentially the same” to “similar.” (page 20, line 516, and page 31, line 785)

Page 23 line 592: change “that lose” to “which lose” since you mean that on average they all do it; “that” is restrictive and “which” is nonrestrictive

We changed “that lose” to which lose.” (page 23, line 588)

Page 26 line 660: change “in the DTA mice” to “in brain slices from the DTA mice” to properly convey that these experiments were not done under in vivo conditions. Do this here and elsewhere (line 661...)

We respectfully argue that this change makes many sentences unnecessarily wordy and redundant. We do not present any in vivo data. Since all of our experiments were performed in brain slices, the potential risk of confusion between in vivo and slice recordings should be low. Therefore, we want to keep the current wording.

Page 26 line 671: delete “complex” in “Tuberous sclerosis complex-1” since it is not part of the definition of Tsc1 (sc= “sclerosis”)

We checked the official symbol and full name of the gene on an NIH website (<https://www.ncbi.nlm.nih.gov/gene/64930>). The official symbol is Tsc1, and the official full name is TSC complex subunit 1. Although “sc” stands for “sclerosis,” as the Referee explained, removing “complex” makes the name different from the official full name. Therefore, we changed “Tuberous sclerosis complex-1 (*Tsc1*) gene” to “Tuberous sclerosis (TSC) complex subunit 1 (*Tsc1*) gene.” We also applied the same change in the Abstract. (page 3, line 76, and page 26, lines 666-667)

Page 34 line 862: change “due to the stronger synaptic input” to “due to stronger unitary synaptic strength”

As we explained in our response to the Referee’s major comment 2, our data do not directly indicate that unitary synaptic inputs are stronger in the cerebellar mutant mice. Therefore, we want to keep the current wording.

Page 35 line 895: the authors may wish to insert an explanatory phrase or sentence such as “Thus in DTA mice and Tsc1 mutant mice, mossy and climbing fiber excitation are transmitted with reduced countering inhibition through the DCN and to the thalamus and other brain regions.” This helps the reader understand that the mice studied do not only have increased tonic DCN output, but may transmit inputs to the cerebellum more strongly to the rest of the brain.

We inserted the suggested explanatory sentence with a slight modification (page 35, lines 899-902).

Page 37 Lines 933-945 the text may be more persuasive and have more impact if you changed “Therefore, it is reasonable to assume that” to “We speculate that”

We changed “Therefore, it is reasonable to assume that” to “We speculate that.” (page 37, line 939)

Page 37 Line 941 “affects the thalamus” to “affects inputs to the thalamus”.

We changed “affects the thalamus” to “affects cerebellar inputs to the thalamus.” (page 37, line 947)

Referee #2:

The authors have adequately addressed all my concerns. I congratulate them on a fantastic study.

We are thankful for the Referee’s constructive critiques in the first round of review and the kind compliment for the revised manuscript.

Referee #3:

I am happy with the statistical analysis approach used and the description provided.

We are pleased to hear that the Referee, an expert in statistics, is satisfied with our statistical analysis approach.

END OF COMMENTS

Dear Dr Nishiyama,

Re: JP-RP-2024-285887R2 "Purkinje cell ablation and Purkinje cell-specific deletion of Tsc1 in the developing cerebellum strengthen cerebellothalamic synapses" by Hiroshi Nishiyama, Naoko Nishiyama, and Boris V Zemelman

We are pleased to tell you that your paper has been accepted for publication in The Journal of Physiology.

Yours sincerely,

Katalin Toth
Senior Editor
The Journal of Physiology

If you would like to receive our 'Research Roundup', a monthly newsletter highlighting the cutting-edge research published in The Physiological Society's family of journals (The Journal of Physiology, Experimental Physiology, Physiological Reports, The Journal of Nutritional Physiology and The Journal of Precision Medicine: Health and Disease), please click this link, fill in your name and email address and select 'Research Roundup':

<https://www.physoc.org/journals-and-media/membernews>

- You can help your research get the attention it deserves! Check out Wiley's free Promotion Guide for best-practice recommendations for promoting your work at: www.wileyauthors.com/eeo/guide. You can learn more about Wiley Editing Services which offers professional video, design, and writing services to create shareable video abstracts, infographics, conference posters, lay summaries, and research news stories for your research at: www.wileyauthors.com/eeo/promotion.

The Corresponding Author will receive an email from Wiley with details on how to register or log-in to Wiley Authors Services where you will be able to place an order

Reviewing Editor's comments:

The authors have done a nice job addressing the prior concerns. They have now toned down their discussion extrapolating mouse behavioral results to human autism, addressing one of the concerns of Reviewer 1. Around concerns about statistical analysis and conclusions made from the negative results in hemizygous Tsc1 deletion mice, Reviewer 1 is satisfied with the authors' response.

Referee #1:

Acceptable now.

Referee #2:

In my view, the authors have adequately addressed the concerns of Reviewer 1 regarding their statistical analysis. Although I agree in principal that data from the hemizygous mice may contribute to behavioral alterations, the authors careful statistical analysis and interpretation are well-justified and not over-stated in the manuscript. I have no other major concerns.

END OF COMMENTS

2nd Confidential Review

13-Oct-2024